# Chemically stable fluorescent proteins for advanced microscopy

Benjamin C. Campbell [1,2,5] ✉, Maria G. Paez-Segala[3], Loren L. Looger[3,4], Gregory A. Petsko [1,2,6] and Ce Feng Liu[1,2,7]

We report the rational engineering of a remarkably stable yellow fluorescent protein (YFP), 'hyperfolder YFP' (hfYFP), that withstands chaotropic conditions that denature most biological structures within seconds, including superfolder green fluorescent protein (GFP). hfYFP contains no cysteines, is chloride insensitive and tolerates aldehyde and osmium tetroxide fixation better than common fluorescent proteins, enabling its use in expansion and electron microscopies. We solved crystal structures of hfYFP (to 1.7-Å resolution), a monomeric variant, monomeric hyperfolder YFP (1.6 Å) and an mGreenLantern mutant (1.2 Å), and then rationally engineered highly stable 405-nm-excitable GFPs, large Stokes shift (LSS) monomeric GFP (LSSmGFP) and LSSA12 from these structures. Lastly, we directly exploited the chemical stability of hfYFP and LSSmGFP by devising a fluorescence-assisted protein purification strategy enabling all steps of denaturing affinity chromatography to be visualized using ultraviolet or blue light. hfYFP and LSSmGFP represent a new generation of robustly stable fluorescent proteins developed for advanced biotechnological applications.

Fluorescent proteins (FPs) have been used for nearly three decades to probe cell biology[1], but the technology is still catching up with striking advances in microscopy methods such as super-resolution imaging[2], expansion microscopy (ExM)[3] and correlative light and electron microscopy (CLEM)[4]. FPs can be adversely affected (sometimes dramatically) by even simple chemical fixation, and many do not function well in the hydrogel-enmeshed samples of ExM.

CLEM places even greater demands on FPs, as it usually involves secondary fixation and staining with caustic chemicals such as osmium tetroxide (OsO₄). Recently developed green-to-red photoconvertible FPs, mEos4b (ref. [4]) and mEosEM[5], showed impressive resistance to the quenching effects of aldehyde fixation and OsO₄. Diverse FPs—particularly constitutively fluorescent ones such as GFP—are required to expand the reach of such methods. Moreover, in addition to requirements of specialized modalities such as CLEM, FPs should have favorable properties for routine use, including fast and complete folding and maturation, high brightness and photostability, and low oligomerization when used in fusions.

We recently developed mGreenLantern (mGL)[6], a bright, monomeric GFP that is well suited for ExM and tissue clearing methods such as 3DISCO (ref. [7]), thereby facilitating neuronal imaging experiments including the tracing of supraspinal projections in a cleared, fully intact brain, while bypassing the laborious and time-consuming antibody enhancement steps[8]. The greater resistance of mGL to chemical and thermal denaturation compared with other common FPs, such as enhanced GFP (eGFP), mClover3 (ref. [9]) and mNeonGreen (mNG)[10], prompted us to expand our study of mGL to realize additional improvements to protein stability and thereby enhance functionality in advanced imaging modalities.

Here we generated well-folded cysteine-free FPs from mGL with stability characteristics eclipsing those of the renowned superfolder

[1]Helen and Robert Appel Alzheimer's Disease Research Institute, Weill Cornell Medicine, New York, NY, USA. [2] Feil Family Brain and Mind Research Institute, Weill Cornell Medicine, New York, NY, USA. [3]Janelia Research Campus, Howard Hughes Medical Institute, Ashburn, VA, USA. [4]Howard Hughes Medical Institute, Department of Neurosciences, University of California, San Diego, CA, USA. [5]Present address: Department of Neurosciences, University of California, San Diego, San Diego, CA, USA. [6]Present address: Ann Romney Institute for Neurologic Diseases, Department of Neurology, Brigham & Women's Hospital and Harvard Medical School, Boston, MA, USA. [7]Present address: MeiraGTx, New York, NY, USA. ✉e-mail: ben.campbell@protonmail.com

**Table 1 | Spectroscopic characterization of FPs**

| Protein | $\lambda_{ex}$ | $\lambda_{em}$ | $\phi$ | $\varepsilon$ (M$^{-1}$cm$^{-1}$) | Molecular brightness[a] | Cellular brightness ($\times$eGFP)[b] | p$K_a$ | Maturation (min) | $T_m$ (°C) | Oligomeric state[e] |
|---|---|---|---|---|---|---|---|---|---|---|
| eGFP | 488 | 507 | 0.71 | 53,400 | 38 | 1.0 | 6.0 | 28 | 80.3 | Weak dimer |
| sfGFP | 485 | 508 | 0.67 | 51,300 | 32 | 1.7 | 6.2 | 37 | 86.4 | Weak dimer |
| eYFP | 513 | 527 | 0.60 | 94,000 | 58 | 1.8 | 6.8 | 37[c] | 72.9 | ND |
| hfYFP | 514 | 529 | 0.60 | 119,500 | 72 | 2.6 | 5.6 | 21 | 94.2 | Weak dimer |
| mhYFP | 515 | 529 | 0.62 | 124,000 | 77 | 2.5 | 5.7 | 27 | 92.8 | Monomer |
| mT-Sapphire | 400 | 510 | 0.63 | 35,400 | 22 | ND[d] | 5.1 | ND | 86.3 | Monomer |
| mAmetrine | 412 | 527 | 0.48 | 33,400 | 16 | ND | 6.3 | ND | 74.8 | Monomer |
| LSSmGFP | 400 | 510 | 0.48 | 36,900 | 18 | ND | 4.7 | ND | 84.8 | Monomer |
| LSSA12 | 398 | 511 | 0.38 | 38,700 | 15 | ND | 4.6 | ND | 93.9 | Monomer |

[a]Molecular brightness=($\phi\times\varepsilon$)/10$^3$. [b]MFI of HeLa cells expressing the FPs relative to eGFP-P2A-mCherry (Methods). [c]eYFP reaches its 50% half-maximal value in 37 min, after which point it shows a slow phase not observed in the other FPs. [d]ND, not determined. [e]We define 'monomer' as OSER assay score >80%; 'weak dimer,' 50–79%. All values in this table were determined experimentally in our laboratory.

GFP (sfGFP)[11]. Our structure-guided screening efforts produced a YFP, hfYFP, whose stability in notoriously chaotropic conditions, including OsO$_4$, allowed it to survive CLEM sample preparation with fluorescence retention matching mEos4b's. hfYFP also showed strong resistance to aldehyde fixatives and performed well in ExM.

We solved the crystal structures of hfYFP (to 1.7-Å resolution), a monomeric variant, monomeric hyperfolder YFP (mhYFP) (1.6 Å), and a cysteine-free mGL variant called FOLD6 (1.2 Å), shedding light on structure–function relationships underpinning their brightness and stability. By applying this knowledge, we generated FPs LSSA12 and LSSmGFP which are exclusively excited by 405-nm excitation, in contrast to mT-Sapphire, which is inconveniently coexcited with eGFP under 488-nm illumination[12]. We then leveraged the high stability of hfYFP and LSSmGFP by developing a fluorescence-assisted protein purification strategy using hfYFP or LSSmGFP fusions to visualize all steps of native and even denaturing Ni-NTA chromatography (6 M guanidinium hydrochloride (GdnHCl)) using inexpensive ultraviolet (UV) or blue light-emitting diodes (LEDs) at the benchtop.

With its remarkable resilience, high solubility, low oligomerization propensity, chloride insensitivity and lack of cysteine residues, hfYFP is a versatile FP which overcomes most of enhanced YFP (eYFP)'s limitations and can replace it in many applications. hfYFP is compatible with protein-retention expansion microscopy (proExM) and CLEM, as well as diverse applications traditionally served by sfGFP. hfYFP and FOLD6 should be good starting points for engineering new biosensors, and the crystal structures can inform further studies. The hyperfolder FPs are ripe for biotechnological applications that have previously been unreachable.

## Results

### Characterization

The spectral properties and performance of hfYFP in cells (Table 1)—having similar excitation/emission maxima (Fig. 1a) but lacking eYFP's 405-nm absorbance peak (Fig. 1b)—make it an appealing replacement for eYFP. hfYFP exhibits faster chromophore maturation (Fig. 1c), fluorescence intensity in *Escherichia coli* equivalent to mGL's and greater than the other GFPs/YFPs tested (Fig. 1d), 33% greater brightness than eYFP and 2.4-fold greater brightness than eGFP in three human cell lines (Fig. 1e), greater pH stability than eYFP (Fig. 1f) and insensitivity to chloride (Fig. 1g). Its bright fluorescence in bacteria corresponded with high soluble protein production. Only 50–60% of the total eGFP, mNG, eYFP and sfGFP protein produced by *E. coli* occurred in the soluble fraction, compared with 68% of mGL and nearly 80% for hfYFP (Extended Data Fig. 1a,b). Thus, improved soluble

protein production and folding efficiency contribute to the superior brightness of hfYFP in bacteria.

hfYFP tolerated deleterious mutations that rendered eGFP and even sfGFP almost entirely nonfunctional, suggesting that it will be a good template for mutagenesis and sensor engineering (Extended Data Fig. 2). hfYFP is compatible with antibodies designed for eGFP (Supplementary Fig. 1), and cells transfected with nuclear localized hfYFP and mhYFP show healthy morphology (Extended Data Fig. 3a). With photostability under laser-scanning confocal illumination approximately equal to mGL's[6], hfYFP bleaches faster than Clover or eYFP, so care should be taken during intensive prolonged imaging (Extended Data Fig. 3b). Compared with human-codon-optimized moxGFP, a cysteine-free sfGFP mutant engineered for performance in the secretory pathway[13], hfYFP offers twofold greater molecular brightness, nearly threefold greater cellular brightness, improved acid resistance, much greater thermodynamic stability (melting temperature ($T_m$) = 94.2 °C versus 79.5 °C, respectively), twofold accelerated chromophore maturation rate (Extended Data Table 1) and faster refolding (Extended Data Fig. 4a,b). Similar to sfGFP, moxGFP denatured immediately upon exposure to GdnHCl, whereas hfYFP never denatured (Extended Data Fig. 4c).

hfYFP localized properly upon fusion or targeting to common intracellular targets, including actin, tubulin, clathrin, endoplasmic reticulum, mitochondria and nuclei (Fig. 1h–m), indicating that it should perform well in difficult fusions that demand monomeric FPs. mhYFP eluted as a pure monomer by gel filtration chromatography (Extended Data Fig. 3c) and scored as a monomer in the organized smooth endoplasmic reticulum (OSER) assay[14]. hfYFP exhibited weak dimer properties in the OSER assay, similar to Clover (Extended Data Fig. 3d,e) and eGFP[15], both of which perform well in common fusions[16]. Altogether, mhYFP and hfYFP offer many advantages over eYFP and moxGFP for routine imaging experiments.

### Chemical stability

We subjected hfYFP to a barrage of denaturing challenges and compared it with widely used FPs (eGFP, sfGFP, mClover3, mNG, eYFP and mGL) in each experiment. Consistent with earlier data[6], sfGFP, mClover3, mNG and eYFP proteins fully denatured within 10 min after dilution into buffered 6.3 M GdnHCl, whereas mGL remained above its half-initial fluorescence value until ~200 min. mGL fluorescence was still measurable after 12 h, and the signal was above background levels for 24 h (Fig. 2a). Intriguingly, instead of dimming, hfYFP grew 50% brighter in 6.3 M GdnHCl. When measured after 12, 24 and 48 h, hfYFP fluorescence was unchanged relative to its value at 2 h (Fig. 2a).

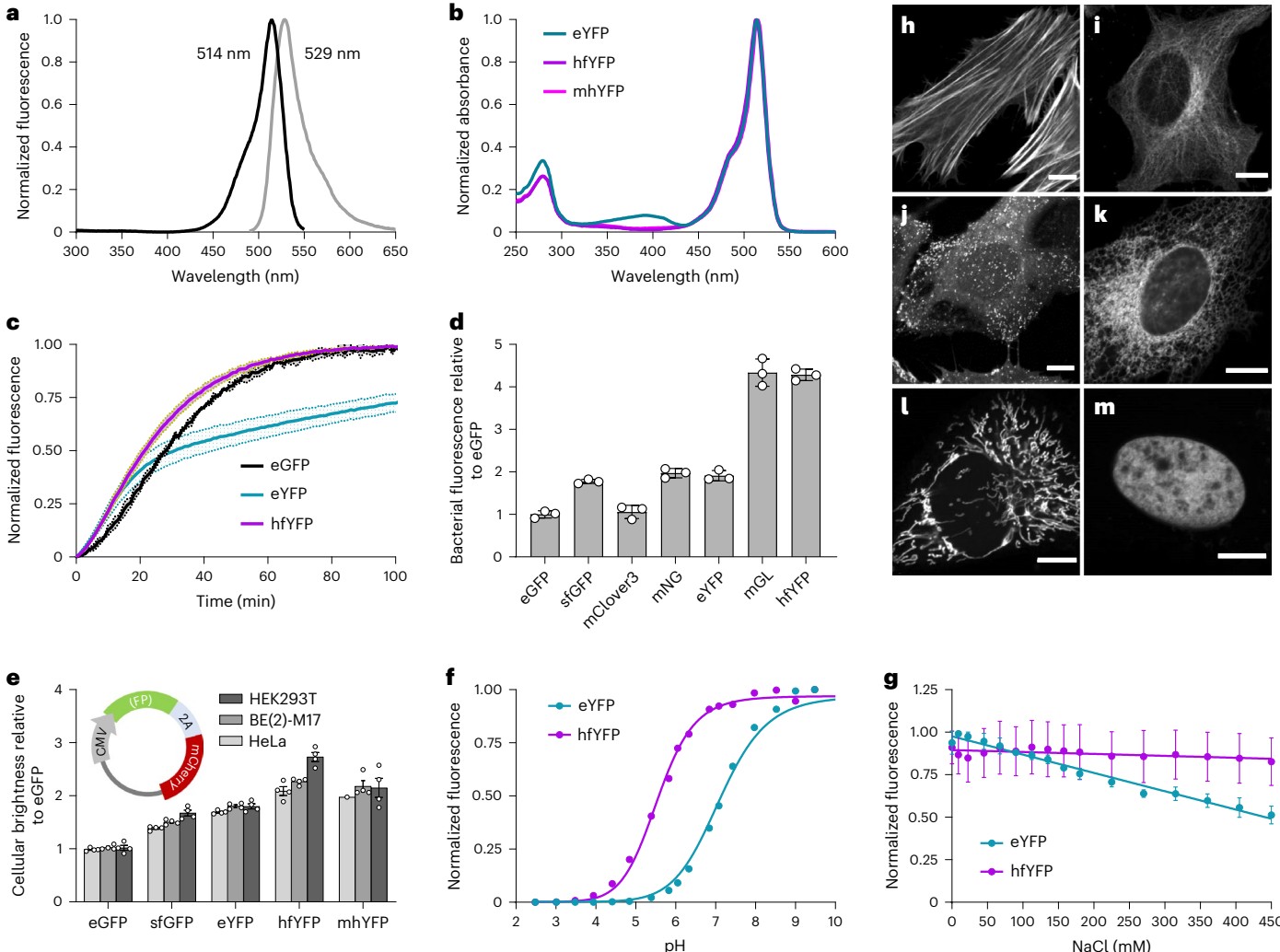

**Fig. 1 | Biochemical characterization of hfYFP and its performance in cells.**
**a**, Excitation and emission spectrum of hfYFP. **b**, Absorbance spectrum at pH 7.5. Arrow indicates the nonexcitable 390–405-nm band present in eYFP and absent in hfYFP and mhYFP. The latter two spectra overlap. **c**, Chromophore maturation in bacterial lysate. Mean ± s.d., $n = 3$ experiments. **d**, Fluorescence of FP-expressing bacteria after overnight growth at 37 °C. **e**, Fluorescence of HeLa, BE(2)-M17 and HEK293T cells 48 h after chemical transfection using a plasmid containing a P2A peptide to generate each FP and an mCherry reference in a roughly equimolar ratio (inset: plasmid). Data for each FP are normalized to the mCherry signal and plotted relative to eGFP. Mean ± s.e.m., $n = 4$ experiments,
each an average of 3 replicate transfections. **f**, pH titration of hfYFP and eYFP. $n = 1$ experiment with 3 technical replicates averaged. **g**, Chloride titration in solutions of HEPES-NaOH, pH 7.5. Data points are fit to a simple linear regression. Mean ± s.d., $n = 2$ experiments. **h**–**m**, Live HeLa cells imaged after overnight transfection using plasmids encoding: **h**, LifeAct-7aa-hfYFP; actin. **i**, hfYFP-6aa-tubulin; tubulin. **j**, hfYFP-15aa-clathrin; clathrin. **k**, pCytERM-hfYFP; endoplasmic reticulum. **l**, COX8A[×4]-4aa-hfYFP; mitochondria, here shown in BE(2)-M17 cells. **m**, H2B-6aa-hfYFP; nucleus, in HeLa cells. Scale bars, 10 μm. Localization images in **h**–**m** are representative of three experiments.

Whereas sfGFP denatured instantly in 7 M GdnHCl, hfYFP persisted >3 months in the same solution at room temperature (RT) (Fig. 2a, inset photograph).

Likewise, in guanidinium thiocyanate (GdnSCN), a stronger denaturant than GdnHCl which has been studied in the context of sfGFP[17], almost all FPs—including sfGFP—denatured instantly at 3.6 M concentration, while mGL and hfYFP did not (Fig. 2b, inset). Instead, mGL and hfYFP persisted for 1.6 and 9.3 min before reaching their half-maximal intensity values along double- and mono-exponential decay trajectories, respectively, with hfYFP persisting over 40 min before its fluorescence fell below background levels (Fig. 2b).

Equilibrium unfolding experiments using GdnHCl confirmed the behavior of hfYFP in 6.3 M GdnHCl and demonstrated dose-dependent fluorescence intensity responses directly proportional to GdnHCl concentration—in marked contrast to the other FPs, which denatured rapidly after surpassing a critical GdnHCl concentration. Consistent

with the kinetic unfolding data, hfYFP brightness in 6.3 M GdnHCl was approximately 50% greater than without GdnHCl (Fig. 2c). mGL and hfYFP also outperformed the other FPs in the more chaotropic GdnSCN solutions during equilibrium unfolding (Fig. 2d). The data provide further evidence that hfYFP is stabilized by GdnHCl even at concentrations that rapidly denature common FPs (and almost all known proteins)[18].

Next, we measured the length of time that FPs could remain fluorescent in Tris-buffered solutions maintained at specific high temperatures ('isothermal melting'). Mirroring the GdnSCN denaturation trends (Fig. 2b), fluorescence quenching of mGL and hfYFP followed double- and mono-exponential decay trajectories with half-maximal time values ($t_{1/2}$) of 17.4 and 40.2 min, respectively, in a thermal cycler programmed to maintain 87 °C temperature while fluorescence measurements were collected every minute (Fig. 2e). All other FPs fully denatured within 8 min, including sfGFP ($t_{1/2} = 1.5$ min) (Fig. 2e, inset).

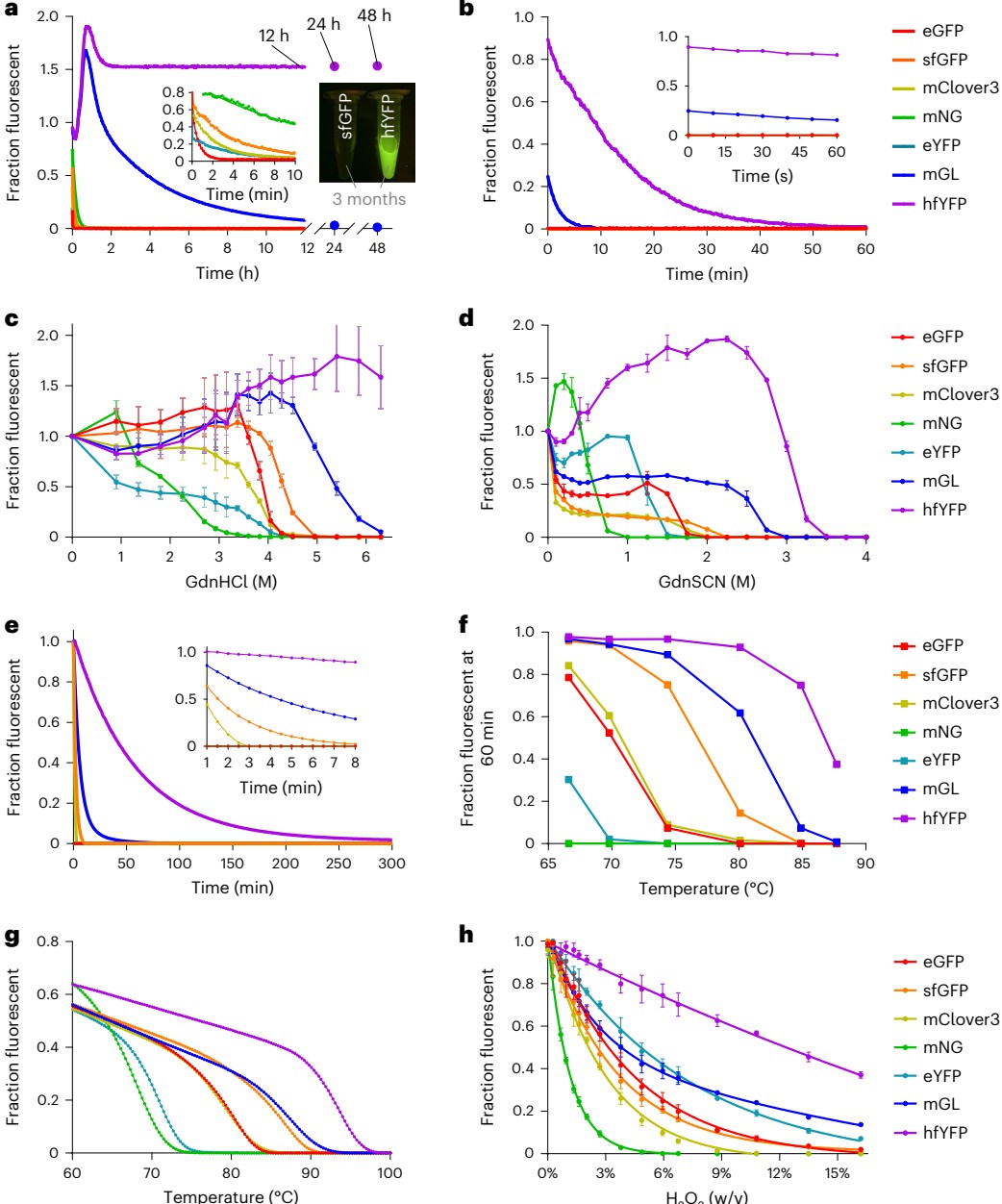

**Fig. 2 | Stability of purified FPs in chaotropic conditions. a**, Kinetic unfolding in 6.3 M buffered GdnHCl solution, pH 7.5. Inset figure: first 10 min of the same dataset. Mean for $n = 2$ experiments. Inset photograph: fluorescence of 1 μM purified sfGFP and hfYFP protein after 3 months in 7 M GdnHCl solution at RT, protected from light. **b**, Kinetic unfolding in 3.6 M buffered GdnSCN, pH 7.5. Inset: first 60 s. Every individual data point in **a** and **b** is normalized to native protein run in parallel under identical conditions in the same buffer without Gdn. Mean for $n = 2$ experiments. **c**,**d**, Equilibrium unfolding in GdnHCl (**c**) and in GdnSCN (**d**), at 24 h. Fluorescence is normalized to the intensity value for each FP at 0 M Gdn. Mean ± s.d., $n = 3$ experiments. **e**, Fluorescence intensity during isothermal

melting at 87 °C, relative to time zero, with normalization as described in **a** and **b**. Inset: first 8 min of the dataset. **f**, Isothermal melting after 60 min of exposure to various temperatures. Data are plotted relative to the intensity value for each FP at 25 °C. See Supplementary Fig. 2 for the complete dataset. **g**, Fluorescence during a 0.3 °C per min temperature ramp from 25 °C to 100 °C, with temperature range 60–100 °C displayed. Data are normalized to the intensity values at 25 °C for the individual FPs. **h**, Intensity versus $H_2O_2$ concentration in buffered solution after exactly 15 min of incubation at RT. Double-exponential curve fits are shown with data points. Mean ± s.d., $n = 3$ experiments. The same seven FPs are plotted in every panel in this figure, including those that denatured immediately ($y \approx 0$).

A summary of relative intensity values at the 60-min time point for each of the six temperatures tested is presented in Fig. 2f, again showing similar rank-ordered stability trends: hfYFP was the most stable, followed by mGL and then by sfGFP. eYFP was only marginally fluorescent after 1 h at just 67 °C, and mNG could not withstand any condition for more than several minutes. Similar to eYFP, mNG instantly denatured at $T \geq 80$ °C (Supplementary Fig. 2). Additional melting curve experiments with freshly purified proteins confirmed

the absence of batch-to-batch response variability and faithfully reproduced $T_m$ values recorded during the screening process (Fig. 2g). The melting temperature of hfYFP ($T_m = 94.2$ °C) was approximately 20 °C and 14 °C greater eYFP's and eGFP's values, respectively. hfYFP and mGL can tolerate higher temperatures for much greater lengths of time than the other FPs, including sfGFP.

When exposed to large molar excess quantities of hydrogen peroxide ($H_2O_2$), hfYFP withstood the greatest concentrations. mNG and

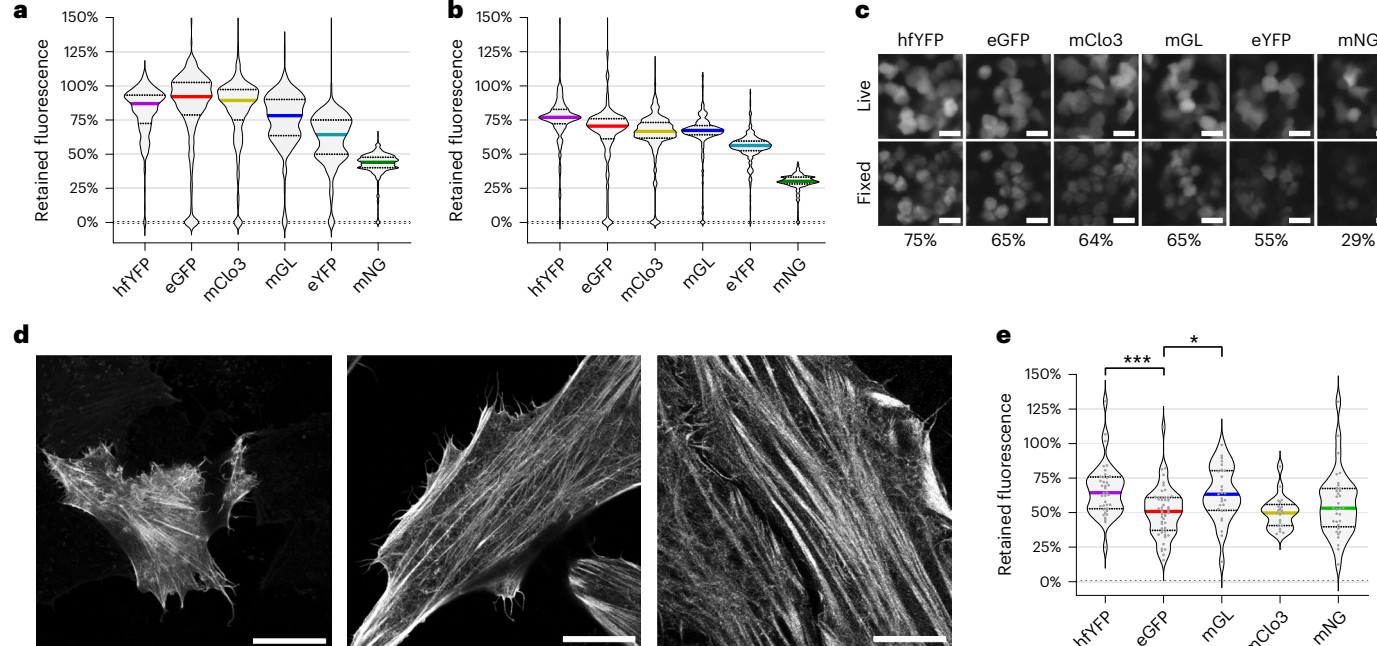

**Fig. 3 | Fluorescence retained by transfected human cells after ExM and fixation. a,b**, Fluorescence retained by HEK293T cells after fixation using RT 4% PFA in PBS, pH 7.4 (**a**), or 4% PFA + 5% Glut in PBS, pH 7.4 (**b**). Cytosolic expression, *n* > 227 cells for each condition per FP, centerline of plot indicates median value. **c**, Representative images of HEK293T cells captured on a widefield microscope using the same acquisition settings before and after fixation as in **b**. Retained fluorescence (%) is indicated below the fixed images in gray. Scale bars, 20 μm. Cytosolic expression. **d**, HeLa cells transfected with LifeAct-mhYFP were imaged on a confocal microscope before proExM (left); after partial expansion of the

hydrogel-enmeshed sample using PBS (middle); and after full expansion using dd-H$_2$O (right). Each image was acquired at ×63 magnification. Scale bars, 25 μm. **e**, Fluorescence retained by H2B-FP-transfected HeLa cells in hypertonic 'shrinking solution' after full expansion in proExM, relative to the same live cells (Supplementary Methods). One-way ANOVA with multiple comparisons, \*\*\**P* = 0.0007, \**P* = 0.0189. Complete statistics for **a**, **b** and **e** are available in Supplementary Fig. 8. Images in **c** and **d** are representative of at least two replicate experiments. mClo3, mClover3.

mClover3 were again the least stable, and sfGFP fared no better than eGFP (Fig. 2h). Altogether, the data demonstrate the superior stability of hfYFP against multiple chemical and thermal denaturing challenges.

We next targeted β-strand 7, the most structurally heterogeneous region of *Aequorea victoria* FPs (avFPs)[19,20], to engineer variants with further enhanced stability. Introducing the S147P mutation that was reported to improve thermostability in violet-light excitable uvGFP[21] yielded hyperfolder mutants with considerable resistance to 1 M sodium hydroxide (NaOH) solutions of pH ≥ 13 (we were able to determine hyperfolder FP extinction coefficients by collecting alkaline denaturation time-course data) (Extended Data Fig. 5 and Supplementary Methods). hfYFP-S147P/V206K/L195M behaved as a stronger monomer in the OSER assay than did hfYFP (Extended Data Fig. 3d,e); the L195M mutation originated de novo from a PCR error. The V206K mutation (on β-strand 10) largely preserved the GdnHCl stability of multiple mutants (Extended Data Fig. 4d). We named the hfYFP-S147P/V206K/L195M variant 'mhYFP'. mhYFP further demonstrates that diverse stability-enhancing characteristics can be structurally engineered into FPs with minimal perturbation to spectral properties (Extended Data Table 1 and Extended Data Fig. 2).

### Compatibility with fixatives and ExM

Paraformaldehyde (PFA) and glutaraldehyde (Glut) are the most common histological preservatives in biology[22], and an FP that can retain the greatest fluorescence in aldehyde fixatives would have considerable advantages in any downstream optical application. HEK293T cells expressing eGFP and hfYFP lost approximately 20% of their fluorescence after fixation with 4% PFA in phosphate buffered saline (PBS),

pH 7.4. Interestingly, consistent with the thermal and chemical stability results, mNG and eYFP fared the worst against PFA, keeping only 42% and 60% of their original fluorescence, respectively (Fig. 3a). hfYFP retained 75% of its fluorescence after fixation with a 4% PFA + 5% Glut solution compared with 65% for mGL and eGFP, and 29% for mNG (Fig. 3b,c). mhYFP is compatible with proExM[3] (Fig. 3d) and retains 16% more fluorescence than eGFP at the end of the process (Fig. 3e). The data highlight the diversity of responses to aldehyde fixatives between FPs and demonstrate the degree to which experiments are impacted by fluorescence quenching at the earliest stage of sample processing.

### CLEM

For use in conditions compatible with electron microscopy, FPs must not only survive primary aldehyde fixation, but they must also thrive in the presence of secondary fixatives and electron-dense reagents. Primary among such reagents is OsO$_4$, which is electron-dense and cross-links both proteins and unsaturated lipids. hfYFP and mhYFP tolerated doses of OsO$_4$ matched only by mEos4b, a green-to-red photoswitchable FP specifically engineered for OsO$_4$ resistance for CLEM[4]. We tested the same FPs that we compared in the other stability assays and found that none could tolerate OsO$_4$ concentrations used for CLEM except mEos4b, hfYFP and mhYFP (Fig. 4a). In a time-course experiment with FPs incubated for 1 h in 1% OsO$_4$ (the concentration most often used for sample preservation), hfYFP retained greater fluorescence than mEos4b and mhYFP, while all other FPs—including sfGFP—were almost totally quenched before the first 10-min time point (Fig. 4b).

We subjected eGFP, mGL and mhYFP to a modified electron microscopy fixation protocol (4% PFA and 0.2% Glut, followed by

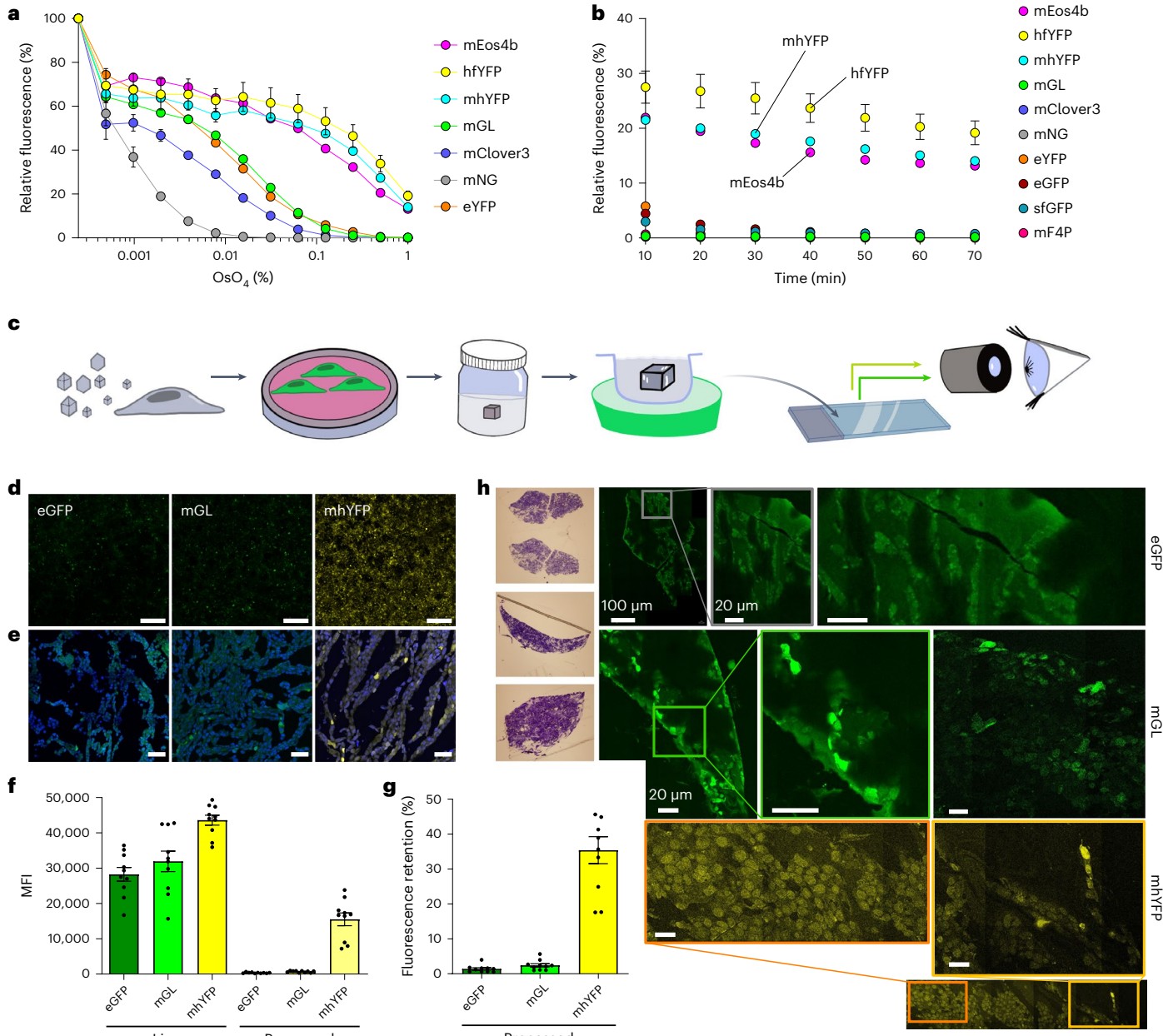

**Fig. 4 | Resilience of hfYFP during electron microscopy preservation.**
**a**, OsO4 dose–response curve using purified FPs after 1 h of incubation at RT. Mean ± s.e.m., $n = 3$ experiments. **b**, Fluorescence of purified FPs in 1% OsO4, recorded at 10-min intervals. Mean ± s.e.m., $n = 3$ experiments. **c**, Workflow for evaluating the performance of FPs in electron microscopy (EM). FPs were expressed in the cytoplasm using AAV transduction and imaged using confocal microscopy. Cultures were then postfixed with EM fixative (4% PFA and 0.2% Glut), collected in bovine serum albumin and embedded in agarose. The agarose-embedded cells were incubated in 1% OsO4 solution for 1 h, after which the osmicated tissue in agarose was embedded in OCT compound on dry ice for cryosectioning. Images of mounted cryosections were collected using the same settings as for live imaging, to evaluate fluorescence retention. **d**, HEK293T cells imaged using confocal microscopy and 488 nm laser excitation after AAV transduction of cytosolic eGFP, mGL or mhYFP. Magnification, ×10. Scale bars, 0.5 mm. Imaging parameters were identical between FPs. **e**, Representative images of cultures viewed at ×63 magnification with DAPI staining. Imaging parameters were identical between FPs and different from those used in **e**. Scale bars, 20 μm. **f**, Background-subtracted MFI units of live and osmicated-and-OCT-embedded cultures. Mean ± s.e.m., $n = 10$ ROIs per FP per condition. These raw intensity values should not be used for brightness comparison because different settings were used for each FP. **g**, Cellular fluorescence retention after OsO4 incubation and OCT embedding, expressed as a percentage relative to the same live cells. Mean ± s.e.m., $n = 10$ ROIs per FP per condition. **h**, Left, toluidine blue staining was used to verify the presence of cells before preparation. Cells were fixed in EM aldehyde fixative, followed by HPF and freeze substitution, 1% OsO4 incubation, dehydration with 100% acetone, HM20 resin infiltration and UV polymerization. Right, laser-scanning confocal images show 100-nm-thick sections of fluorescent HEK293 cells expressing eGFP (top), mGL (middle) and mhYFP (bottom). Scale bars, 20 μm, unless otherwise indicated. Images in **d**, **e** and **h** are representative of two replicate experiments.

1% OsO4 and embedding in agarose and optimal cutting temperature (OCT) compound—that is, polyvinyl alcohol and polyethylene glycol) and quantified the retained fluorescence (Fig. 4c). HEK293 cells were

transduced with adeno-associated virus (AAV) particles and live fluorescent cells were imaged by confocal microscopy at ×10 (Fig. 4d) and ×63 magnification (Fig. 4e) before and after processing, using the

same settings. The processed mhYFP cells retained ~35% of the initial live-cell fluorescence (Fig. 4f), a 14- and 25-fold fluorescence retention improvement compared with mGL and eGFP, respectively (Fig. 4g). mhYFP also retained fluorescence after acrylate-based resin embedding in a high-pressure freezing (HPF)/freeze substitution protocol, with fluorescence levels well above background (Fig. 4h).

## Structure-guided engineering of LSS FPs

hfYFP (PDB: 7UGR), mhYFP (PDB: 7UGS) and FOLD6 (PDB: 7UGT) crystallized with the symmetry of space groups C222₁, C222₁ and P6₄, diffracting to 1.7 Å, 1.6 Å and 1.2 Å, respectively. Each FP crystallized as a monomer with no asymmetric unit. Structures were solved by molecular replacement using homology models generated from Clover (PDB: 5WJ2)[23]. Having examined structure–activity relationships in our hfYFP, mhYFP and FOLD6 crystal structures (Supplementary Information), we sought to generate companion hyperfolder FPs for spectral multiplexing that could be excited exclusively by 405-nm illumination. We destabilized the deprotonated chromophore ('B-state')[24] (Supplementary Fig. 3a,b) by introducing the T65S/Y203I mutations found in mT-Sapphire that are primarily responsible for its 405-nm excitation band[25], essentially producing an hfYFP/mT-Sapphire chimera. Our hfYFP-G65S/Y203I/V206K mutant was 405-nm-excitable, indicating a protonated chromophore population, but it still retained an excitation band at 460 nm (Extended Data Fig. 6a).

We designed a structurally targeted library to eliminate the B-band by mutating positions 203, 204, 206, 221, 222 and 223. These residues on β-strands 10 and 11, especially those at positions 203 and 222, are crucial for the orientation of chromophore-associated side chains such as S205 that participate in excited-state proton transfer and directly influence the FP's spectral properties (Fig. 5a,b)[26]. We mutated the six residues simultaneously using a Gibson Assembly overlap approach with oligonucleotides carrying a targeted set of degenerate codons to minimize library size to 384 combinations (Extended Data Fig. 6b and Supplementary Methods).

After screening E. coli colonies by eye for high green emission after 405-nm and low green emission after 470-nm excitation, mutants were identified by spectroscopic analysis with B-band excitation greatly reduced compared with hfYFP-G65S/Y203I (Extended Data Fig. 6c). We extracted soluble protein from a subset of mutants and denatured the clarified lysate using the same GdnHCl kinetic unfolding strategy we employed to identify hfYFP. After a single round of screening, a mutant with an optimal combination of high GdnHCl stability (Extended Data Fig. 6d) and no B-band excitation was sequenced, yielding hfYFP-G65S/Y203I/Q204E/E222D/R223F, which we named LSSA12. LSSA12 was more stable in GdnHCl than mT-Sapphire, mAmetrine and eGFP (Extended Data Fig. 6e). Site-directed mutagenesis confirmed the functional importance of the E222D and G65S mutations in LSSA12 (Extended Data Fig. 6f).

Since hfYFP can tolerate avFP knockout substitutions (Extended Data Fig. 2), we produced a diversified LSSA12 library by purifying the 12 best plasmids from the LSSA12 screen, diluting the DNA with original template to back-cross the library and recombining the genes in a staggered extension process[27] reaction spiked with MnCl₂ to increase the mutational frequency (Supplementary Methods). The most stable clones with optimal excitation spectra and high GdnHCl stability were sequenced, yielding LSSmGFP, which is hfYFP-T43S/G65S/L68Q/H77N/K140N/Y203I/V206K. Similar to mAmetrine[28], both LSSA12 and LSSmGFP display a single excitation peak, but mT-Sapphire has two peaks (Extended Data Fig. 7a). To confirm the function of Gln68, we mutated mT-Sapphire to generate mT-Sapphire-V68Q (mT-Sapphire2, which we did not characterize further) and confirmed that V68Q alone was sufficient to decrease mT-Sapphire's residual 488-nm excitation by 2.8-fold, almost to zero (Extended Data Fig. 7a). Therefore, the V68Q substitution is a seemingly new and apparently generalizable solution for enhancing spectral tuning in 405-nm-excitable avFPs.

LSSmGFP persisted longer in GdnHCl than LSSA12, mT-Sapphire and mAmetrine (Extended Data Fig. 7b). LSSmGFP and LSSA12 are more acid-tolerant than mAmetrine ($pK_a = 6.3$) and mT-Sapphire ($pK_a = 5.1$), having $pK_a$ values of 4.6–4.7 (Table 1). They are also more resistant to H₂O₂ denaturation (Extended Data Fig. 7c). Their melting temperatures ($T_m$) are high, at 84.8 °C and 93.9 °C, respectively, comparable to sfGFP's and hfYFP's (Table 1 and Extended Data Fig. 7d). Cells transfected with LSSmGFP retained 72% of their live-cell fluorescence after fixation with 4% PFA, compared with 75% retained fluorescence for LSSmGFP, 62% for mT-Sapphire and only 41% for mAmetrine (Extended Data Fig. 7e). Similar to hfYFP, LSSmGFP and LSSA12 surpassed the stability of other FPs in Glut (Extended Data Fig. 7f). Under laser-scanning confocal illumination, LSSmGFP's photostability is comparable to mT-Sapphire's and twice as long as mAmetrine's (Extended Data Fig. 7g), while its molecular brightness is roughly equal (Table 1).

LSSA12 and LSSmGFP, as well as mT-Sapphire and mAmetrine, behaved as monomers in cultured cells (Extended Data Fig. 7h–i). LSSmGFP localized properly to common intracellular targets (LSSA12 was not tested) (Extended Data Fig. 7j–n). LSSmGFP and LSSA12 enjoy similar advantages as hfYFP, including the absence of cysteine residues, low $pK_a$, tolerance of fixatives, high chemical and thermal stability, and a single excitation band.

Based on the clean excitation spectra of LSSmGFP and LSSA12, we hypothesized that these FPs would perform well in cells when coexpressed with mGL (Fig. 5c), and that mT-Sapphire and eGFP would show bleed-through between the 405-nm and 470-nm channels (Fig. 5d). We transfected eGFP + mT-Sapphire, or mGL + LSSmGFP, into HeLa cells as actin or H2B fusions, respectively. As expected, 470-nm widefield excitation inappropriately coexcited mT-Sapphire + eGFP, whereas LSSmGFP + mGL were properly separated into their respective channels with no bleed-through (Fig. 5e). Along with greater thermostability and chemical stability, LSSmGFP and LSSA12 offer improved excitation profiles relative to mT-Sapphire, without cross-excitation artifacts.

## Fluorescence-assisted protein purification

Most human proteins are insoluble when expressed in E. coli at 37 °C and are sequestered into inclusion bodies, which must be extracted and purified under chaotropic conditions typically involving buffers of 6 M GdnHCl[29]. We aimed to exploit the high GdnHCl stability of hfYFP (Fig. 2a,c) and LSSmGFP by using them as fusion tags to visualize all stages of protein purification under denaturing conditions during immobilized metal affinity chromatography (IMAC).

We generated constructs of hexahistidine (His₆)-tagged eGFP, hfYFP or LSSmGFP, with the FP separated from mScarlet-I[30], Bacillus circulans xylanase (Bcx) or streptavidin (SAV)[31] (collectively, Proteins of Interest, or POIs) by a linker containing a TEV protease cleavage site (Fig. 5g). We developed a purification protocol using an hfYFP–mScarlet fusion construct (Extended Data Fig. 8) and then purified SAV fusions of eGFP, hfYFP and LSSmGFP under denaturing conditions using Ni-NTA chromatography with all solutions containing 6 M GdnHCl. With hfYFP and LSSmGFP, we successfully visualized every stage of bacterial lysis, inclusion body isolation, solubilization, purification, refolding during dialysis, proteolytic cleavage and isolation of the fusion protein, using inexpensive 470-nm LED strips (or 405-nm for LSSmGFP) and orange filter glasses. Elution was monitored by eye using LED illumination at the benchtop (Supplementary Videos 1 and 2). The purified SAV was functional after dialysis and refolding in Tris buffer, exhibiting 33% binding activity relative to purified SAV obtained from a commercial source (Fig. 5h). As expected from the results of the unfolding experiments, eGFP immediately went dark (denatured) during inclusion body solubilization (Fig. 5f, arrow), while hfYFP and LSSmGFP remained brightly fluorescent throughout all purification steps. We tested the durability of the fluorescence in 6 M GdnHCl and found that LSSmGFP (and mGL) lost substantial fluorescence after overnight refrigeration in denaturing solution, whereas hfYFP's fluorescence remained stable (Extended Data Fig. 9). Therefore, hfYFP is a better choice for lengthy purification experiments.

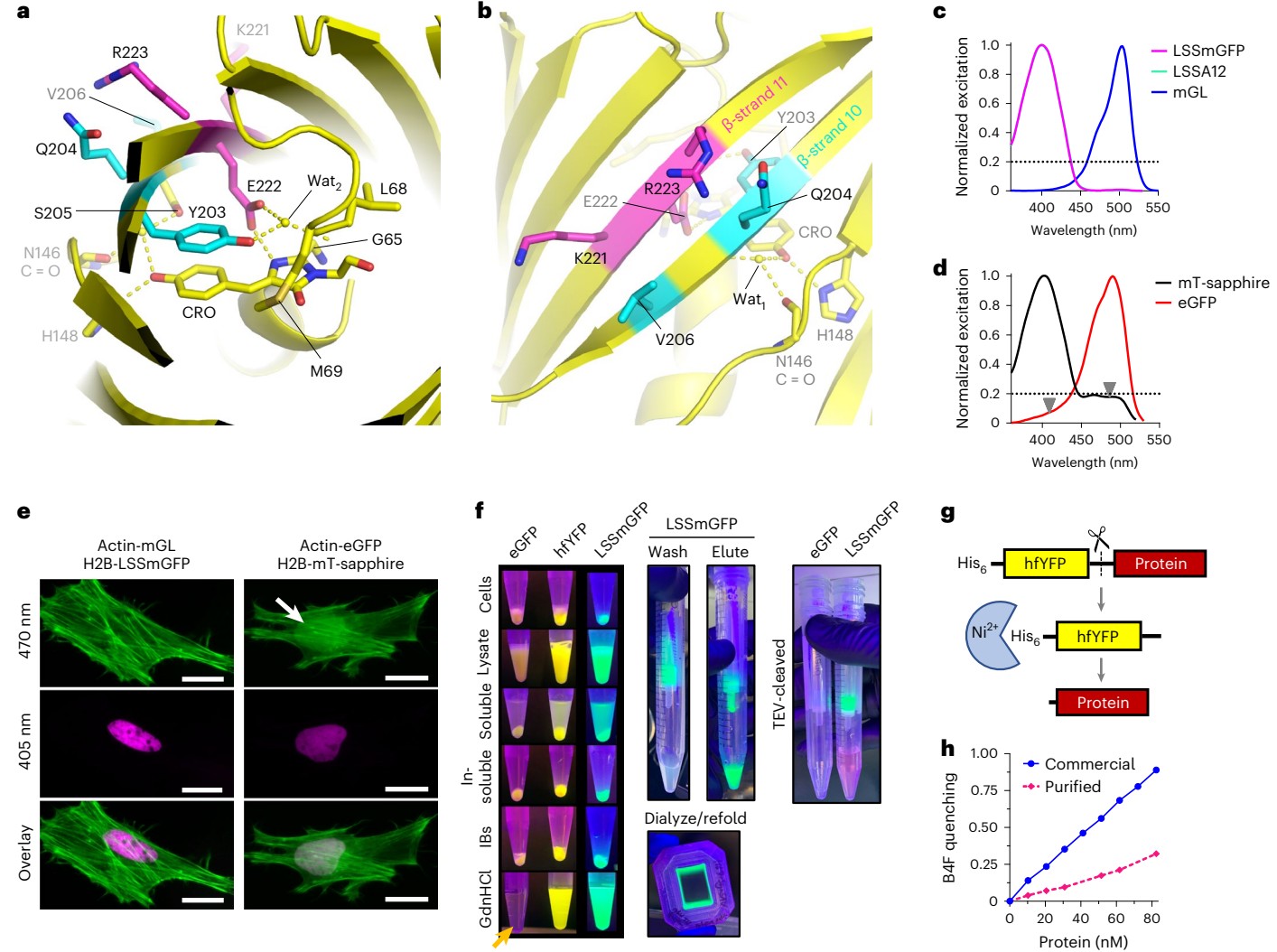

**Fig. 5 | Structure-guided engineering of LSS GFPs and their use in fluorescence-assisted protein purification. a**, Chromophore environment in the hfYFP crystal structure (PDB: 7UGR). Residues on β-strands 10 (cyan) and 11 (magenta), which were targeted to generate the LSS FP libraries, are indicated. Yellow dashes, hydrogen bonds. **b**, Surface of the hfYFP structure, facing β-strands 10 and 11. **c**, Excitation spectra of mGL, LSSmGFP and LSSA12. The latter two FP spectra overlap. **d**, Excitation spectra of mT-Sapphire and eGFP. Arrows indicate wavelength ranges where cross-excitation would be expected from typical 405-nm or 470–491-nm excitation sources. **e**, Live HeLa cells transfected with LifeAct-eGFP and H2B-mT-Sapphire or LifeAct-mGL and H2B-LSSmGFP. Excitation at 470 nm excites both mT-Sapphire and eGFP (white arrow), whereas LSSmGFP is not excited by 470 nm. Scale bars, 25 μm. Images are representative of at least two replicate experiments. **f**, Fluorescence-assisted purification of SAV from *E. coli* inclusion bodies (IBs) using eGFP, hfYFP or LSSmGFP fusions as depicted in **g**. eGFP is immediately denatured during IB solubilization with 6 M GdnHCl (orange arrow) and never regains fluorescence. hfYFP and LSSmGFP remain fluorescent throughout all stages of denaturing Ni-NTA purification (6 M GdnHCl present in all solutions), dialysis-based refolding of SAV in native buffer, enzymatic cleavage with His6-TEV protease and re-isolation of cleaved SAV in native buffer by Ni-NTA chromatography (see Extended Data Fig. 8 for workflow). hfYFP is illuminated using a 470-nm LED at the benchtop and photographed through an orange long-pass filter; LSSmGFP: 405-nm LED excitation without emission filter. **g**, The fusion construct used for purification contains an N-terminal His6-tagged hfYFP (or eGFP or LSSmGFP for the example in **f**) and C-terminal POI separated by a flexible linker containing a TEV cleavage site. After cleavage, His6-TEV and His6-hfYFP are adsorbed to Ni-NTA resin and the flow-through is collected to obtain the POI. **h**, Fluorescence of biotin-4-fluorescein (B4F) is quenched upon SAV binding[38]. The refolded SAV protein isolated after cleavage was active.

Interestingly, hfYFP and LSSmGFP might function as solubility tags that enhance expression of the C-terminal fusion protein. When fusions were expressed in *E. coli* at 37 °C under strong induction conditions to maximize inclusion body formation, we obtained 30% and 60% less mScarlet-I in the insoluble fraction (Extended Data Fig. 1c,d) and 290% and 220% more mScarlet-I in the soluble fraction from the hfYFP– and LSSmGFP–mScarlet fusions, respectively, than from the eGFP–mScarlet construct. Compared with the eGFP–Bcx fusion, 820% and 800% more fusion protein was obtained from the soluble fraction of hfYFP and LSSmGFP fusions, while a marginal 10% and 20% more SAV was collected. In the clarified lysogeny broth (LB) medium (without cells), 930% and 470% more mScarlet-I fusion protein was detected for the hfYFP and LSSmGFP constructs, respectively, compared with eGFP. Almost no eGFP–Bcx fusion construct was detected in the media, while high amounts were visible for the hfYFP and LSSmGFP fusions (Extended Data Fig. 1c,d).

Altogether, fusing an insoluble protein (or a soluble one, in the case of mScarlet-I) to hfYFP or LSSmGFP generated much more soluble and less insoluble fusion protein than the equivalent eGFP fusions for three diverse proteins that have no appreciable similarity in sequence or structure. None of the C-terminal fusion proteins contain cysteines, so the improvement cannot be attributed to eliminating disulfide-mediated aggregation in the POIs. hfYFP and LSSmGFP have potential to serve as

solubility tags or at least as markers for fluorescence-assisted protein purification in addition to the numerous applications that we have described for proExM, CLEM and protein engineering.

## Discussion

In this study, we have described the engineering, thorough characterization and application of unusually durable FPs in stability assays, expansion microscopy (proExM), aldehyde fixation, sample preparation steps for CLEM and fluorescence-assisted protein purification. Through structure-guided engineering and screening based on GdnHCl unfolding kinetics and melting temperature ($T_m$) alongside rigorous spectroscopic characterization, we produced a series of avFP mutants demonstrating superior stability over sfGFP in diverse biochemical assays.

In addition to the CLEM application, we achieved our sub-aims of advancing and better understanding the stability of mGL (Fig. 2): we eliminated cysteines without disrupting fluorescence (Supplementary Fig. 5a–d); ensured no 405-nm excitability (Fig. 1b); ensured that the brightness of human cells expressing the FPs would be no less than those expressing eGFP or eYFP (Fig. 1e and Supplementary Fig. 5g); described structure–function correlates of the thermodynamic stability (Supplementary Information); and demonstrated that the 'hyperfolder' FPs can tolerate unusual substitutions, even to the highly conserved C48, C70 and W57 residues. The W57F mutant of hfYFP, with an 18-amino-acid genetic code entirely lacking the stabilizing Trp and Cys residues that are conserved across all avFPs, was brighter and more stable than even wild-type sfGFP (Extended Data Fig. 2). Benefiting from the hfYFP crystal structure, we eliminated hfYFP's 514-nm excitability and produced two exclusively 405-nm excitable GFPs with an LSS, LSSA12 and LSSmGFP. These LSS FPs overcome the cross-excitation problem of mT-Sapphire (Fig. 5c–e) without sacrificing molecular brightness (Table 1). LSSmGFP has high chemical and thermodynamic stability and the same molecular brightness as mAmetrine, while lasting twice as long under laser-scanning confocal illumination before photobleaching (Extended Data Fig. 7g).

Our data suggest that misfolding and lower soluble protein yield (Extended Data Fig. 1) are principally responsible for the diminished brightness of cysteine- and tryptophan-substituted avFPs, and that the brightness and stability deficits incurred by such radical structural perturbations can largely be corrected without modifying the spectral properties (Extended Data Fig. 2h). Using structure-guided engineering, we conferred the NaOH resistance that we first identified in FOLD6 into hfYFP (Extended Data Fig. 5) and enhanced hfYFP's monomericity in one step (Supplementary Fig. 6), producing mhYFP whose spectral properties are identical to hfYFP's (Extended Data Table 1). Of all the mutants and the handful of bright avFPs that we compared, hfYFP showed the greatest structural plasticity, implying that it will tolerate random mutagenesis and circular permutation at least as well as sfGFP. Indeed, hfYFP proved to be an excellent template for engineering LSSA12 and LSSmGFP. hfYFP performs well in fusions, and mhYFP offers a more monomeric option at a negligible cost to stability.

hfYFP's stability is uncommon for any class of protein, and perhaps most notable is its peculiar tolerance of GdnHCl solutions of 7 M concentration (Fig. 2c), practically indefinitely (Fig. 2a). hfYFP's resilience was not an idiosyncratic response to guanidinium: apart from the GdnHCl and GdnSCN kinetic- and equilibrium unfolding experiments, hfYFP retained more fluorescence than eGFP, sfGFP, mClover3, mNG, eYFP and even mGL, at higher temperatures and for greater lengths of time (Supplementary Fig. 2), in the presence of $H_2O_2$ (Fig. 2h), and after exposure to PFA (Fig. 3d), PFA/Glut (Fig. 3e) and even $OsO_4$ (Fig. 4a,b). The relatively high acid stability ($pK_a = 5.6$) and chloride resistance mean that hfYFP can be deployed in more organelles and other cellular environments without fear of artifacts. We found mNG to be by far the least stable FP we tested against temperature, guanidinium, $H_2O_2$, PFA, Glut and $OsO_4$—this FP should be used in challenging applications with great caution.

Even small gains in fluorescence retained after chemical fixation or other quenching processes can amplify cellular brightness differences, and vice versa. mGL, with its 6.0-fold greater fluorescence than eGFP in mammalian cells[6] compared with 2.4-fold for hfYFP (Fig. 3a), may be better suited for proExM, while hfYFP should be more advantageous for CLEM due to its tolerance of osmication matching that of mEos4b (Fig. 4a,b). Of course, there is no single FP that can suit every possible application, but hfYFP has proven itself to be a versatile tool for routine imaging and shows promise in the tested super-resolution modalities.

The high-resolution crystal structures of hfYFP, mhYFP and FOLD6 that we solved to 1.7-Å, 1.6-Å and 1.2-Å resolution, respectively, offer excellent templates for biosensor engineering, particularly when combined with our characterization data (Extended Data Table 1), library development approach for the GFPs/YFPs (Supplementary Note), LSS FPs and structural interpretations (Supplementary Information). We speculate that the dense hydrophobic packing of the hfYFP chromophore environment, in addition to surface mutations and interactions that stabilize the barrel structure, may help the protein resist solvation effects that have been proposed as part of a two-stage mechanism for GdnHCl-induced protein denaturation[32], thereby preserving fluorescence perhaps even while the FP is in a partially unfolded or molten globule or state. Deeper insight into hfYFP's stability should be revealed by experiments that pair the unfolding data and crystal structures with biophysical structure–activity correlates, such as free energy calculations, regional and whole-protein solvent-accessible surface area, protein volume and electrostatic interactions, with comparisons with eYFP, Citrine and Venus. Molecular dynamics simulations would clarify the nature of the GdnHCl and NaOH resistance and possibly reveal mechanisms that could in theory be transferred to other FPs beyond hfYFP—potentially even sequence-diverse ones. Additional value remains to be unlocked from the hfYFP, mhYFP and FOLD6 crystal structures.

hfYFP and FOLD6 have potential to serve similar functions as sfGFP, such as multiple epitope tag insertion[33], extremophile research[34], bimolecular fluorescence complementation[35], sensor stabilization[36], circular permutation and random mutagenesis[11]. Meanwhile, the stability improvements of the hyperfolder FPs open doors to new applications that have not yet been realized for expression-enhanced biosensors, perhaps along with concomitant decreases in cytotoxicity due to the improvements in protein folding and solubility. On the other hand, we do not yet know whether the high stability of hyperfolder FPs impacts their metabolic fate in cells. Studies have indicated that some FPs, such as mCherry, can accumulate in lysosomes and resist proteasomal degradation[13,37]. Therefore, it is worth verifying that fusions with hyperfolder proteins exhibit their normal stability and half-life, particularly when planning sensitive assays such as those involving protein turnover.

hfYFP is a versatile protein that may find use in expansion microscopy (proExM), CLEM and tissue clearing. Besides those applications, hfYFP and LSSmGFP enable fluorescence-assisted protein purification and might even act as solubility tags (Fig. 5f,g). Likewise, LSSmGFP performed well in assays in which hfYFP excelled, including protein purification, and they may find similar uses. The hyperfolder FP crystal structures can serve as launch points for engineering new biosensors, and we have already made progress in developing hfYFP-based biosensors and further stabilized variants with unique properties. Biotechnological applications that were previously complicated or irresolvable due to sfGFP's limitations, such as the presence of cysteines and the susceptibility to chemical denaturation, may now be in reach.

## Online content

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

## Methods

### Fluorescence-assisted protein purification

For a visual aid, see the flowchart in Extended Data Fig. 8. Fusions of hfYFP and mScarlet-I, Bcx or SAV were transformed into *E. coli* TOP10 cells and expressed overnight in 50 ml of LB medium supplemented with 100 μg μl$^{-1}$ ampicillin and 0.2% arabinose. All fusions were grown at 37 °C for 18 h and centrifuged in 50-ml conical tubes, and the pellets were frozen at −80 °C.

Strips of UV (405 nm) and blue (470 nm) LEDs with a wide viewing angle were affixed to a shelf approximately 0.6 m above the work surface and wired to toggle switches. All steps of the following purification process could be visualized under 405-nm or 470-nm LED illumination as desired for LSSmGFP or hfYFP, using yellow (Arrowhead Forensics) or orange (Invitrogen) filter goggles, respectively.

The frozen pellets from 50 ml of culture were thawed at RT and lysed in B-PER II reagent (Thermo Scientific) supplemented with DNAse I. After 5–10 min of incubation at RT, lysate was sonicated on ice until homogeneous, transferred into several 1.5-ml Eppendorf tubes and centrifuged at 20,000g for 10 min at 4 °C in a benchtop microcentrifuge. The soluble fraction was collected. The insoluble pellet was washed with PBS plus 1% Triton X-100, briefly sonicated and centrifuged, and the supernatant containing residual soluble protein was discarded. This inclusion body pellet was washed twice more as described using PBS without the Triton X-100. Inclusion bodies were resuspended one sample at a time by dispensing Denaturing Buffer (20 mM phosphate, 300 mM NaCl, 6 M GdnHCl, pH 7.4) containing 10 mM imidazole into the tube and immediately sonicating the sample on ice until fully solubilized (-10 s), yielding a brightly fluorescent and homogeneous solution.

The fusion proteins were purified from solubilized inclusion body solutions using HisPur Ni-NTA resin (Thermo Scientific, no. 88223). All purification steps were performed using the mentioned denaturing purification buffer containing imidazole at a final concentration appropriate for equilibration, washing or elution steps (10, 25 or 250 mM imidazole, respectively). Protein elution was monitored under LED illumination, and the fluorescent eluate was collected (Supplementary Videos 1 and 2). The eluate was loaded into 10K MWCO dialysis cassettes (Thermo Scientific) and dialyzed in 50 mM Tris-HCl, pH 8.0, with gentle magnetic stirring at 4 °C overnight.

The concentration of the dialyzed sample was measured using its computed molecular extinction coefficient and absorbance at 280 nm. Next, His$_6$-tagged AcTEV protease (Invitrogen, no. 12575023) was added for cleavage according to manufacturer instructions. IMAC-incompatible chemicals such as dithiothreitol and EDTA were omitted. Cleavage proceeded at 4 °C for at least 24 h.

The cleaved POI was then isolated by Ni-NTA chromatography under native conditions in the mentioned Purification Buffer without GdnHCl. His$_6$-TEV protease and His$_6$-hfYFP or His$_6$-LSSmGFP are adsorbed onto the Ni-NTA resin while the POI remains in solution. After binding, the column is unplugged and the flow-through containing the purified POI is isolated. See Extended Data Fig. 8 for a flowchart. Enzymatic activity of SAV was quantified as described using fluorescence quenching of biotin-4-fluorescein (CAS no. 1032732-74-3)[38].

Note: mScarlet-I and SAV refolded during dialysis under the described conditions as part of the hfYFP fusion construct, but Bcx did not. Refolding of denatured proteins is a complex topic that is abundantly addressed elsewhere. Conditions to maximize refolding efficiency and catalytic activity of each POI must be optimized. It is necessary to consider whether the refolding buffer is compatible with the specific protease that is intended to cleave the construct (in this case, TEV protease), and also whether the refolding buffer is compatible with IMAC. Otherwise, dialysis steps should be included.

### Thermal denaturation and isothermal melting

FP was adjusted to 1 μM final concentration in tris-NaCl-glycerol (TNG) buffer and 50 μl was dispensed into replicate wells of a clear 96-well quantitative PCR plate. The plate was sealed with optical adhesive and heated in a Bio-Rad C1000 Touch thermal cycler equipped with a CFX96 Real-Time System and fluorescein amidite filter. Temperature was brought to 25 °C and measured three times in 30-s intervals to obtain the baseline fluorescence value before heating at a rate of 0.3 °C per min to a final temperature of 100 °C. Datasets containing the negative first derivative of the change in fluorescence (−d(RFU)/dT) were exported from the Bio-Rad CFX96 software, background subtracted and normalized to the average of the baseline fluorescence value at 25 °C. The melting temperature ($T_m$) is the x value when the normalized intensity y value = 1. Melting temperatures for the LSS FPs were determined using the thermofluor assay as described[39], using SYPRO Orange dye (Thermo Fisher) and the carboxyrhodamine filter set with the same equipment.

For the isothermal melting experiment, samples were prepared as described and the thermal cycler was programmed using the gradient setting to heat individual plate rows to 66.6 °C, 69.8 °C, 74.4 °C, 80.1 °C, 84.9 °C and 87.7 °C, using the maximum ramp rate. The temperatures were maintained for 5 h while fluorescence measurements were acquired every 1 min using the fluorescein amidite filter set. Data were background subtracted and plotted relative to the first data point of the series.

### Kinetic unfolding

Purified FP stocks were diluted tenfold into TNG buffer, pH 7.5, with and without 7 M GdnHCl (Fisher Scientific, no. BP178–500), for final concentrations of 0.1 μM FP and 6.3 M GdnHCl (or 0 M GdnHCl for the native control samples). The plate was immediately sealed with optical adhesive (Bio-Rad), and the first fluorescent measurement was recorded within -15 s of dilution ($\lambda_{ex}/\lambda_{em}$ = 495/525 nm, or 405/525 nm for the LSS FPs). Short-term and long-term unfolding curves were generated using 10-s or 1-min sampling interval for total experiment durations of 1 h or 12 h, respectively. The fluorescence ratio of the unfolding protein to the same FP's native control wells was plotted for each FP ('fraction folded'). Data points were fit to single- or double-exponential equations where indicated. Preliminary screening of mutant libraries for generating the LSS constructs was performed using clarified lysate and then confirmed using purified protein for final datasets.

### Equilibrium unfolding

Equilibrium unfolding experiments were conducted based on a reported method[40]. FPs were diluted to 0.1 μM final concentration in solutions of GdnHCl or GdnSCN in TNG buffer, pH 7.5, except that dithiothreitol was omitted in this experiment. Plates were sealed with optical adhesive and stored in the dark at RT for 24 h before collecting measurements using endpoint scans of $\lambda_{ex}/\lambda_{em}$ = 495/525 nm. Data were plotted relative to the fluorescence intensity value of the individual FP at 0 M Gdn (TNG buffer only).

### H$_2$O$_2$ and chloride sensitivity assays

Stocks of 1 μM FP were diluted tenfold into solutions of H$_2$O$_2$ (Fisher Scientific, no. H325-100) in PBS, pH 7.4, for final concentrations of 0.1 μM protein and 0–27% v/v H$_2$O$_2$ (16 solutions). Fluorescence was measured after exactly 15 min of incubation at RT using endpoint scans of $\lambda_{ex}/\lambda_{em}$ = 495/525 nm, or 405/525 nm for the LSS FPs. Timing is critical for achieving reproducible results in this experiment since H$_2$O$_2$ is used in great excess, even at the lowest concentrations, resulting in very rapid denaturation of all FPs within 15–60 min.

To measure chloride sensitivity, the same dilution procedure was used. Samples were then incubated in solutions of HEPES-NaOH, pH 7.5, with chloride concentrations ranging from 0 to 500 mM (16 solutions) for 24 h in the dark at RT before measurement using endpoint scans of $\lambda_{ex}/\lambda_{em}$ = 495/525 nm.

## Fluorescence retention, fixatives

A fresh ampule of methanol-free 32% (w/v) PFA (Electron Microscopy Sciences) was diluted to 4% (v/v) in PBS, pH 7.4 (Gibco). The remaining 32% (w/v) PFA liquid was then used to prepare a solution of 4% (v/v) PFA and 5% (v/v) Glut in PBS, pH 7.4, from a 25% (w/v) stock bottle that was stored sealed under refrigeration. To improve cell adherence, PBS++ was prepared by supplementing a 10× PBS, pH 7.4 (Gibco), stock bottle with $MgCl_2$ and $CaCl_2$ solutions before dilution to 1× working concentration and 0.1 mM final concentration of each salt.

HEK293T cells were cultured in Matrigel-coated black 96-well tissue plates and transfected with cytosolic expression plasmids using 0.1 µg of DNA per well and Turbofect transfection reagent. Medium was changed the following morning. After 48-h total expression time, cells were washed twice using phenol red-free medium and imaged at ×10 magnification on the mentioned widefield Keyence BZ-X700 plate reader microscope using a PlanFluor DL ×10 0.30/15.20-mm Ph1 air objective, GFP filter cube for GFPs and YFPs, or the mentioned 405/525-nm filter cube for LSS FPs. Manually focused images of 960 × 720 pixels (2 × 2 binning) resolution were acquired in 8-bit TIFF format. Imaging settings were specific to each FP to provide optimal exposure without oversaturation. Acquisition settings were recorded, and the same settings were used for prefixation and postfixation imaging per FP.

After imaging the live cells (prefixation image), samples were incubated with RT fixatives for 15 min at RT. Afterward, the cells were gently washed three times for 5 min each using PBS++. The washed cells were imaged using the settings loaded from the appropriate live-cell fluorescence image for each well. Pre- and postfixation images were registered using ImageJ. The live-cell image was thresholded, cell regions of interest (ROIs) were applied to the postfixation image by creating a mask and the mean fluorescence intensity (MFI) of each cell before and after fixation was collected. Data were expressed as a ratio of the postfixation MFI relative to the live-cell MFI (percentage 'fluorescence retention') and analyzed statistically by one-way analysis of variance (ANOVA) with multiple comparisons.

## Cellular brightness and protein solubility

The brightness of human cells expressing transfected FPs ('cellular brightness') was obtained using the P2A plasmid coexpression strategy[41] and methods as described[6]. Bacterial brightness and protein solubility measurements were performed as described[6]. Briefly, bacteria were grown overnight in LB broth supplemented with 100 µg ml$^{-1}$ ampicillin ('media'). The following day, cultures were adjusted to identical optical density ($OD_{600}$ = 0.6) and used to inoculate at a 1:100 v/v ratio fresh medium supplemented with 0.2% arabinose to induce expression. After overnight growth at 37 °C, 100 µl of culture from each replicate was dispensed into a 96-well optical plate (Corning), and fluorescence measurements were acquired using a plate reader with $\lambda_{ex}/\lambda_{em}$ = 495/525 nm. Data were background subtracted and normalized to one of the eGFP wells. The remaining cultures were pelleted and frozen at −80 °C for the protein solubility experiment.

To obtain soluble protein, pellets were resuspended in PBS and lysed by freeze–thaw cycling between 37 °C water and liquid nitrogen, followed by sonication on ice and high-speed centrifugation at 4 °C to obtain the soluble fraction. The insoluble pellet was washed with ice-cold PBS, and this new supernatant containing residual soluble protein was discarded. The washed insoluble pellet was resuspended in ice-cold PBS and homogenized by brief sonication to yield the insoluble fraction.

Total protein concentration of the soluble and insoluble fractions was determined using the BCA Protein Assay Kit (Pierce) with a BSA standard curve. Samples were adjusted to matching protein concentration and boiled for 10 min at 100 °C in the presence of SDS-containing loading dye supplemented with 0.2 M dithiothreitol. Samples were analyzed by SDS–PAGE on 4–15% Tris-Glycine gels (Bio-Rad), and

the 27-kDa band representing the FP monomer was quantified using densitometry tools in ImageJ and normalized to the eGFP value. For the protein fusions shown in Fig. 5g and Extended Data Fig. 1c,d, the 57-kDa, 51-kDa and 44-kDa bands were quantified for fusions of FPs with mScarlet-I, Bcx and SAV, respectively.

## Cell-free OsO$_4$ tolerance assay

OsO$_4$ dose–response curves were produced using an established protocol[4]. Briefly, purified FPs were dialyzed into 50 mM sodium phosphate buffer, pH 7.4, and normalized by concentration at $A_{280}$. A solution of 4% (w/v) OsO$_4$ (Electron Microscopy Sciences) was diluted to a final concentration of 1% and serially diluted in 96-well black, clear-bottom optical plates (Greiner). Pure protein was added to the wells to a final concentration of 1 µM, and the plate was immediately sealed with optical adhesive (Bio-Rad). After 10 min of incubation at RT, kinetic scans of FPs in OsO$_4$ solution were performed over a 10-min period at 1-min intervals using both the green (ex/em, 480/515 nm; bandwidth, 5 nm/5 nm) and yellow (ex/em, 493/517 nm; bandwidth, 5 nm/5 nm) channels. All measurements were performed with identical fluorometer settings.

## Fixation and embedding for the cellular OsO$_4$ experiment

The protocol for fixation and embedding from Paez-Segala and colleagues[4] was followed, with some modifications. HEK293 cells (ATCC) were maintained in Eagle's Minimum Essential Medium (EMEM), containing 1 mM sodium pyruvate, 4 mM L-glutamine, 4.5 g ml$^{-1}$ glucose and 1.5 g ml$^{-1}$ sodium bicarbonate. Complete growth medium was prepared by addition of fetal bovine serum to 10% (w/v) final concentration.

To induce eGFP, mGL and mhYFP cytosolic expression, cells were cotransduced with AAV particles with FP expression under control of a Cre-dependent *CAG* promoter (Addgene) along with AAV-CAG-Cre virus to induce expression. Approximately $2 × 10^6$ cells were seeded on 100-mm plates for collection and fixation. For live fluorescence measurements, ~$8 × 10^5$ live HEK293 cells were seeded on 35-mm glass-bottom dishes (MatTek).

Brightness quantitation of the live cells before and after aldehyde/OsO$_4$ processing was performed using a laser-scanning confocal microscope (Zeiss LSM 880, with ZEN Black software) with the following settings: 488-nm laser (0.5% power); ×10 objective; emission range 495–560 nm; GaAsP detector gain of 715 V. Imaging settings for mhYFP used the 488-nm laser at 0.45% power; ×10 objective; emission range 400–700-nm detection; and 600-V detector voltage. Images were quantified using Fiji (ImageJ). For background correction, the MFIs of 10 fields of view containing nonfluorescent cells were subtracted from each of 10 ROIs containing the live fluorescent cells to be quantified.

After imaging the live cells, cultures were fixed with CLEM fixative (4% (w/v) PFA + 0.2% (w/v) Glut) (Electron Microscopy Sciences) in 100 mM phosphate buffer and then collected in 10% BSA using a rubber spatula. Cells were then mixed with a 2% (w/v) agarose solution and pelleted. After the agarose solidified, it was cut using a scalpel into squares and submerged in a 1% (w/v) solution of OsO$_4$ for 1 h at RT.

Osmicated pellets in agarose were then embedded in OCT frozen tissue specimen medium (Fisher Scientific, no. 23–730–571). Next, 8–10-mm sections of the osmicated pellets were cut on a cryotome (Leica), placed on a glass slide and mounted with antifade mounting medium (Invitrogen). Images of the mounted cells were then captured using the same imaging settings as described for the live cells.

## HPF and freeze substitution

Live fluorescent cells seeded in 35-mm tissue culture dishes were fixed in CLEM fixative, collected with a 20% (w/v) BSA solution and assembled into the HPF specimen carrier (Type A, 0.1/0.2 mm; Type B, flat; Techno Trade). The carrier assembly was then introduced into the sample holder and frozen in the HPF machine per the manufacturer's instructions (Wohlwend HPF Compact 01 High-Pressure Freezer; Techno

Trade). Cell tissue was then freeze-substituted, osmicated with a 1% $OsO_4$ (w/v) solution, dehydrated in 100% (w/v) acetone and embedded in Lowicryl HM20 resin (Electron Microscopy Sciences). After the resin was polymerized using UV lamp exposure, plastic blocks were cut into 100-μm sections for both toluidine blue staining and fluorescence visualization. Cells were mounted on glass slides using antifade mounting medium (Invitrogen).

## Reporting summary

Further information on research design is available in the Nature Research Reporting Summary linked to this article.

## Data availability

Crystal structures have been uploaded to the PDB under accession codes 7UGR (hfYFP), 7UGS (mhYFP) and 7UGT (FOLD6). Plasmids have been deposited at Addgene. DNA sequences of the fluorescent proteins have been uploaded to GenBank: OP373686 (hfYFP), OP373687 (mhYFP), OP373688 (LSSA12), OP373689 (LSSmGFP). Additional data that support the findings of this study are available from the corresponding author upon reasonable request.

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

## Acknowledgements

We thank O. Boudker and A. Menon (Weill Cornell) for generous use of their fluorimeter. At Janelia, we thank M. Rodriguez (Molecular Biology) for help with the CAG-AAV plasmid design; A. Gutu (Tool Translation Team) for help with virus transductions; N. Iyer (EM Shared Resource) for help with the high-pressure freezing/freeze substitution EM method; S. Michael (Histology) for help with cryosectioning and mounting; P. Tillberg, M. Copeland, D. Alcor and B. Blanco for help with ExM experiments; and the entire Imaging Shared Resource. We thank J. Naffin-Olivos for assessing the diffraction quality of an early hfYFP crystal. This research used resources of the Advanced Photon Source, a US Department of Energy (DOE) Office of Science User Facility operated for the DOE Office of Science by Argonne National Laboratory. This study was supported by startup funding from the Weill Cornell Brain & Mind Research Institute and Neuroscience Department.

## Author contributions

B.C.C. conceived the project, performed experiments and analyzed data. M.G.P.-S. performed and analyzed CLEM experiments. C.F.L. solved and refined the crystal structures. B.C.C. wrote the manuscript. G.A.P. and L.L.L. supervised the project. All authors edited the manuscript.

## Competing interests

B.C.C. and G.A.P. have filed a patent application (U.S. Provisional Application No.: 63/388,051) describing the hyperfolder proteins and fluorescence-assisted protein purification. M.G.P.-S., L.L.L. and C.F.L. declare no competing interests.

## Additional information

**Extended data** is available for this paper at https://doi.org/10.1038/s41592-022-01660-7.

**Correspondence and requests for materials** should be addressed to Benjamin C. Campbell.

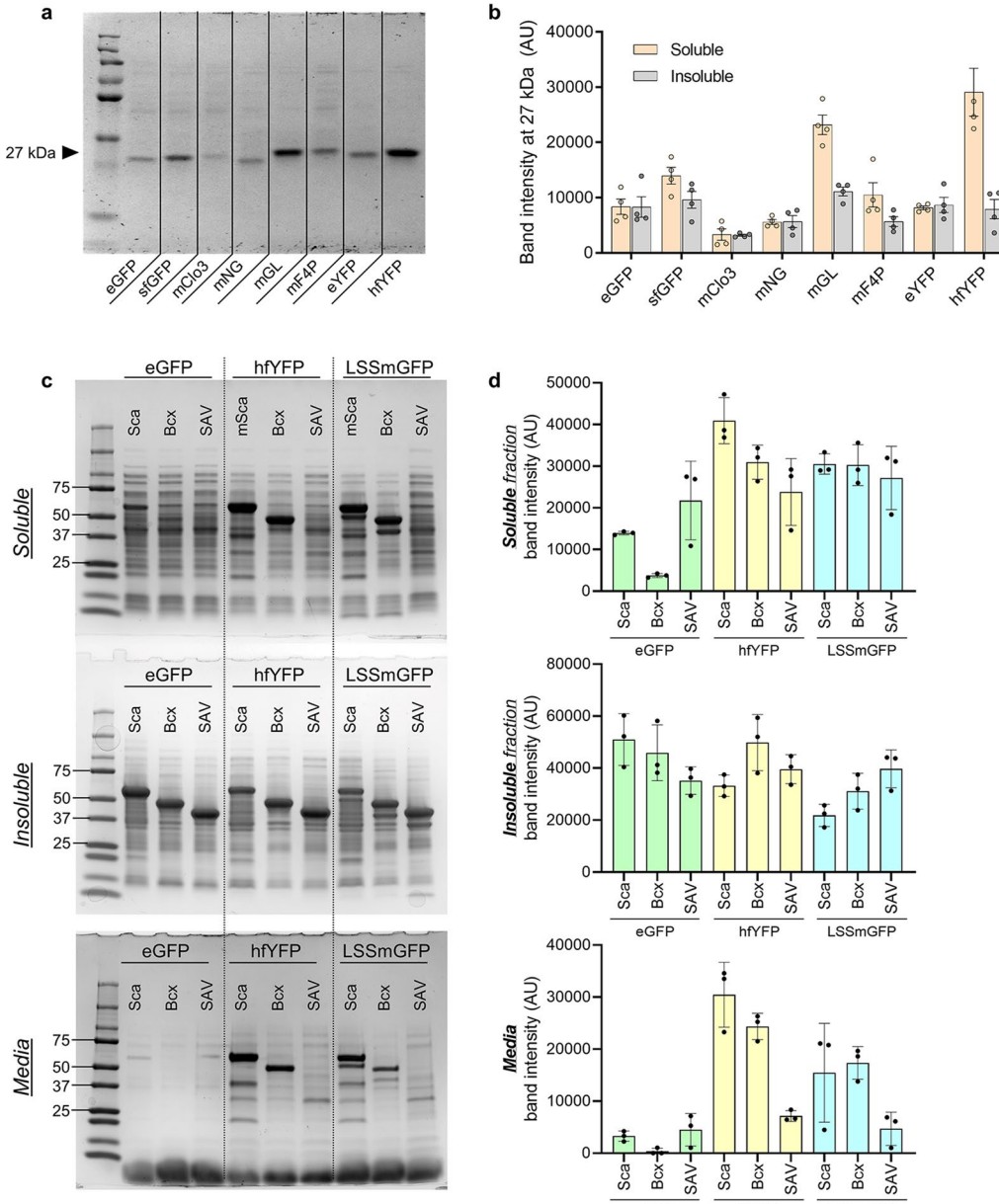

**Extended Data Fig. 1 | Protein yields obtained from *E. coli* lysate fractions.**
**a**, Coomassie gel of soluble protein extracted from *E. coli* lysate after overnight expression. Black arrowhead indicates the fluorescent protein band. The same quantity of protein was run in each lane, so the band intensity at ~27 kDa indicates soluble FP yield. The pictured gel is representative of 4 replicate experiments. **b**, Quantification of the ~27 kDa band using ImageJ's densitometry tools. The total protein extracted from each culture is the sum of the soluble and insoluble fractions. Mean ± s.d., n = 4 experiments. **c**, Coomassie gels of soluble protein (top) insoluble protein (middle) and protein from the media of the same

cultures without cells (bottom). Equal quantity of protein was run in each lane as determined by BCA assay, except for the media condition where equal volume was used without adjustment. The molecular weight (MW) predicted by ExPASy for the FP fusions to mScarlet-I (Sca), *Bacillus circulans* xylanase (Bcx), and streptavidin (SAV) are approximately 57 kDa, 51 kDa, and 44 kDa, respectively. The MW of an avFP is ~27 kDa. Gels pictured are representative of 3 replicate experiments. **d**, Quantification of the protein fusion bands from gels in **a** using standard ImageJ densitometry tools. Mean ± s.d., n = 3 experiments.

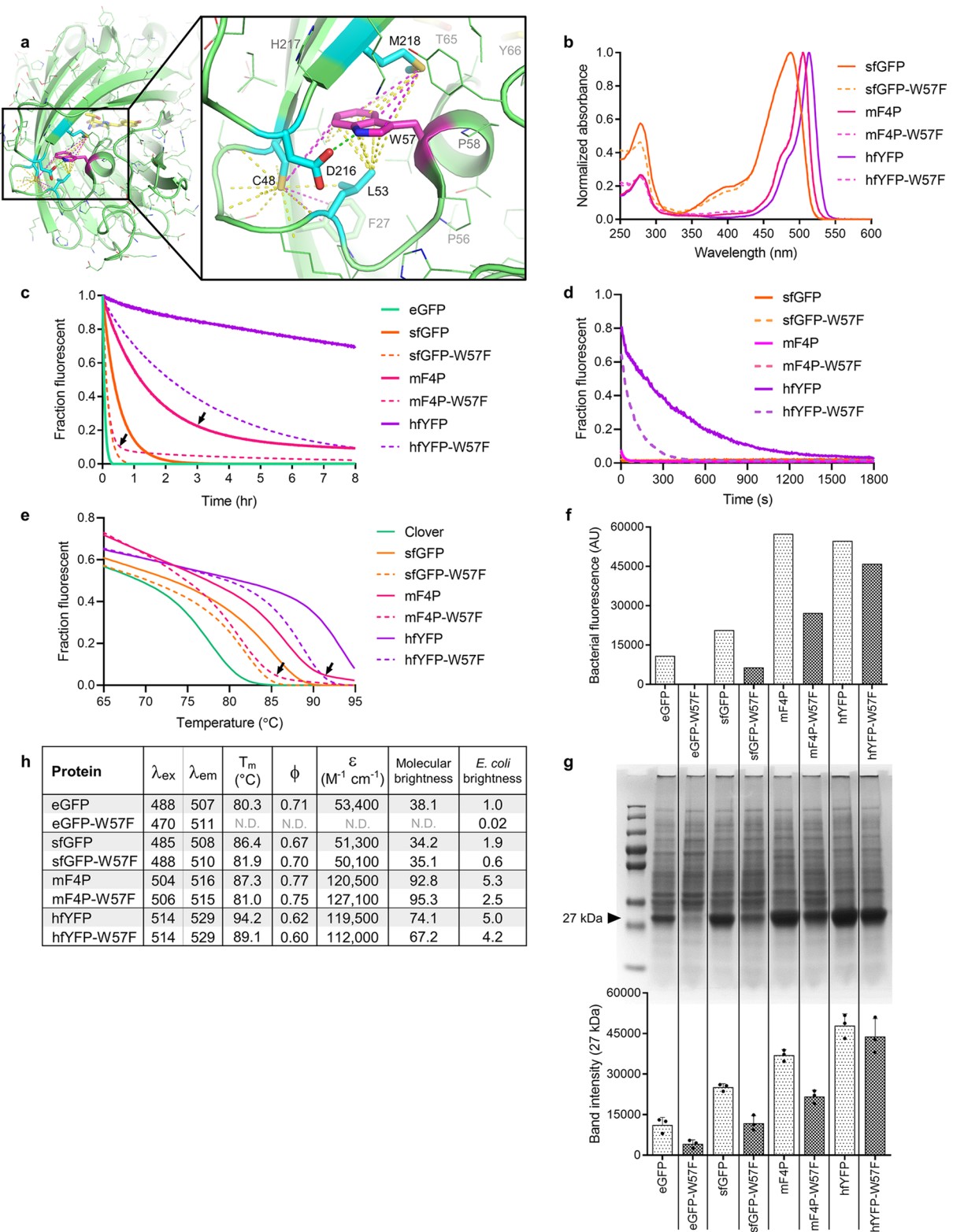

**Extended Data Fig. 2 | Rational engineering and characterization of fluorescent proteins using only 18 of the 20 naturally occurring amino acids. a**, Location of Trp57 in a high-resolution eGFP crystal structure (PDB: 4EUL). Interactions of particular interest are indicated by dashed lines. Yellow: vdW interactions, including the weakly polar interactions from C48. Magenta: sulfur-aromatic interactions of approximately 5.5 Å distance to the ring centroid. Green: H-bonds. **b**, Absorbance spectra of W57F mutants and parental templates. **c**, Isothermal melting of W57F mutants at 80 °C. mF4P and mF4P-W57F denature in a double-exponential manner featuring a prolonged slow phase after ~80–90% of the initial fluorescence is quenched. **d**, Kinetic unfolding in 3.6 M GdnSCN,

pH 7.5, with $\lambda_{ex}/\lambda_{em}$ = 495/525 nm. **e**, Melting curves of W57F mutants compared to the original proteins. **f**, Whole-cell fluorescence intensity of *E. coli* cultures grown overnight before extraction of soluble protein from the same cultures. One experiment is shown. **g**, Quantification of soluble protein yield by measuring intensity of the ~27 kDa band representing the FP monomer (Supplementary Methods). Mean ± s.d., n = 3 experiments. **h**, Spectroscopic characterization table of W57F mutants. mF4P-W57F and hfYFP-W57F entirely lack cysteine and tryptophan residues. Molecular brightness = ($\phi \times \epsilon$)/10³. *E. coli* brightness: fluorescence data from **f** with experimental samples normalized to eGFP.

| Protein | $\lambda_{ex}$ | $\lambda_{em}$ | $T_m$ (°C) | $\phi$ | $\epsilon$ (M⁻¹ cm⁻¹) | Molecular brightness | *E. coli* brightness |
|---|---|---|---|---|---|---|---|
| eGFP | 488 | 507 | 80.3 | 0.71 | 53,400 | 38.1 | 1.0 |
| eGFP-W57F | 470 | 511 | N.D. | N.D. | N.D. | N.D. | 0.02 |
| sfGFP | 485 | 508 | 86.4 | 0.67 | 51,300 | 34.2 | 1.9 |
| sfGFP-W57F | 488 | 510 | 81.9 | 0.70 | 50,100 | 35.1 | 0.6 |
| mF4P | 504 | 516 | 87.3 | 0.77 | 120,500 | 92.8 | 5.3 |
| mF4P-W57F | 506 | 515 | 81.0 | 0.75 | 127,100 | 95.3 | 2.5 |
| hfYFP | 514 | 529 | 94.2 | 0.62 | 119,500 | 74.1 | 5.0 |
| hfYFP-W57F | 514 | 529 | 89.1 | 0.60 | 112,000 | 67.2 | 4.2 |

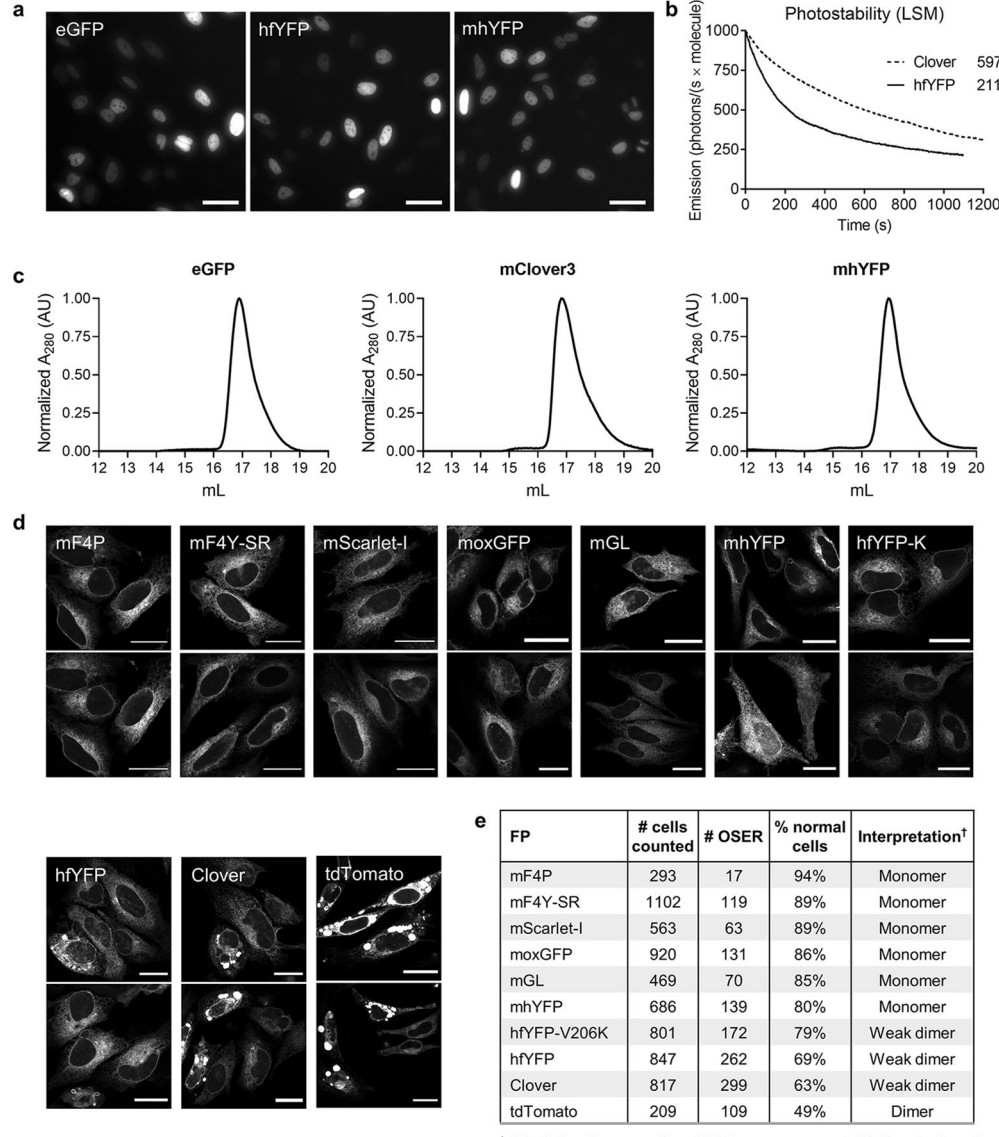

| FP | # cells counted | # OSER | % normal cells | Interpretation[†] |
|---|---|---|---|---|
| mF4P | 293 | 17 | 94% | Monomer |
| mF4Y-SR | 1102 | 119 | 89% | Monomer |
| mScarlet-I | 563 | 63 | 89% | Monomer |
| moxGFP | 920 | 131 | 86% | Monomer |
| mGL | 469 | 70 | 85% | Monomer |
| mhYFP | 686 | 139 | 80% | Monomer |
| hfYFP-V206K | 801 | 172 | 79% | Weak dimer |
| hfYFP | 847 | 262 | 69% | Weak dimer |
| Clover | 817 | 299 | 63% | Weak dimer |
| tdTomato | 209 | 109 | 49% | Dimer |

[†] We define "monomer" as OSER assay score >80%; "weak dimer," 50 to 79%; "dimer," <50%.

**Extended Data Fig. 3 | Additional characterization of hyperfolder proteins.**
**a**, HeLa cells transfected with plasmids encoding H2B-eGFP, H2B-hfYFP, or H2B-mhYFP show healthy nuclear morphology. Several anaphase cells are visible in the mhYFP image. Scale bars, 50 μm. **b**, Time to photobleach H2B-FP transfected HeLa cells from 1000 to 500 photons $s^{-1}$ molecule$^{-1}$ under laser-scanning confocal illumination (Supplementary Methods). Photostability $t_{1/2}$ values in seconds are displayed next to each FP label in the figure legend. **c**, Gel filtration chromatography using 10 μM purified fluorescent protein in a Superdex S200 Increase 10/300 GL sizing column with elution monitored at 280 nm. The monomer fraction elutes at ~17.0 mL and the dimer ~15.5 mL. Data are normalized to the maximum $A_{280}$ value for each FP. **d**, Representative HeLa cells imaged 16 hr

after chemical transfection with CytERM-FP plasmids. The organized smooth endoplasmic reticulum assay (OSER) is a cellular FP aggregation protocol. Cells with large, bright aggregates, like those in tdTomato, are scored as 'OSER.' Normal cells have healthy reticular ER structure, nuclei, and no OSER structures (see Supplementary Methods; refer to definitions by Costantini et al., *Traffic*, 2012). Scale bars, 25 μm. **e**, OSER assay results analyzed using scoring criteria described in **d**. Images in **a** and **d** are representative of 2 and at least 3 replicate experiments, respectively. Abbreviations: mGL, mGreenLantern; mNG, mNeonGreen; hfYFP, hyperfolder YFP; mhYFP, monomeric hyperfolder YFP; sfGFP, superfolder GFP; hfYFP-K, hfYFP-V206K.

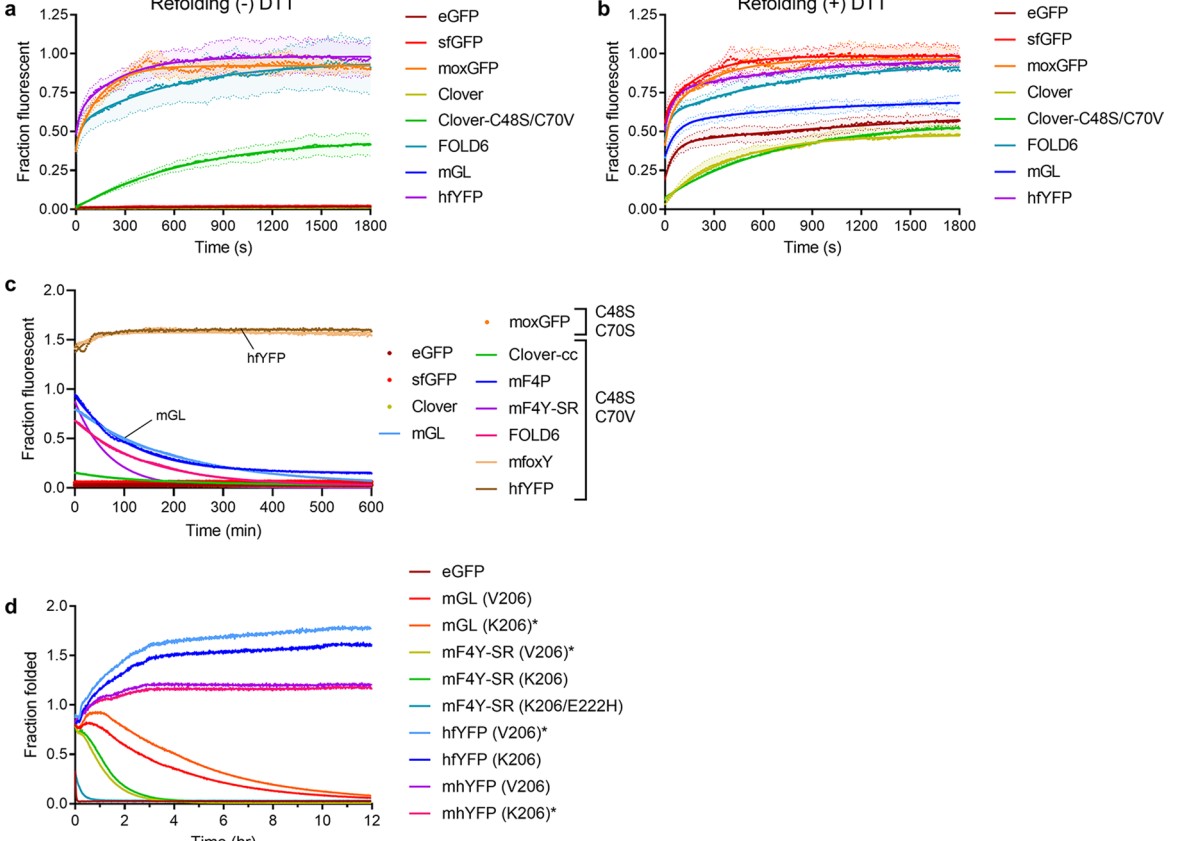

**Extended Data Fig. 4 | Screening FPs for unfolding and refolding rates.**
**a**, Refolding of FPs after denaturation, relative to untreated native samples. Purified FPs (1 μM concentration) were melted at 95 °C for 10 min in the presence of 6.3 M GdnHCl, pH 7.5, and were allowed to cool to room temperature (RT). Refolding was initiated by 10-fold dilution into fresh buffer lacking GdnHCl for 0.1 μM final protein concentration. Mean ± s.d., n = 3 experiments. **b**, Refolding performed as described, with addition of reducing agent dithiothreitol (DTT) in the GdnHCl and refolding buffers. Final concentrations: [GdnHCl] = 0.63 M, [DTT] = 1 mM. All data points are shown with double exponential curve fits superimposed. Mean ± s.d., n = 3 experiments. **c**, Kinetic unfolding of several purified FPs in 6.3 M buffered GdnHCl, pH 7.4. Each data point is normalized

to that of the same FP in buffer without Gdn, which was run in parallel. FPs containing cysteine substitutions are indicated. One-phase nonlinear regression is plotted for all FPs except for those with slope = 0 (that is, denatured instantly). FPs that denatured instantly ($y = 0$ for $x \geq 0$) upon exposure to GdnHCl are indicated by a colored dot instead of a line. Mean ± s.d., n = 3 experiments. 'Clover-cc' signifies the C48S/C70V mutations to distinguish it from unmodified Clover here. **d**, Kinetic unfolding of V206 and K206 hyperfolder mutants in GdnHCl. FPs were unfolded in 6.3 M GdnHCl, pH 7.5, as described (Methods). Asterisks indicate the final mutants, that is, mGreenLantern, mF4Y-SR, hfYFP, and mhYFP, whose properties are reported in Extended Data Table 1 and elsewhere.

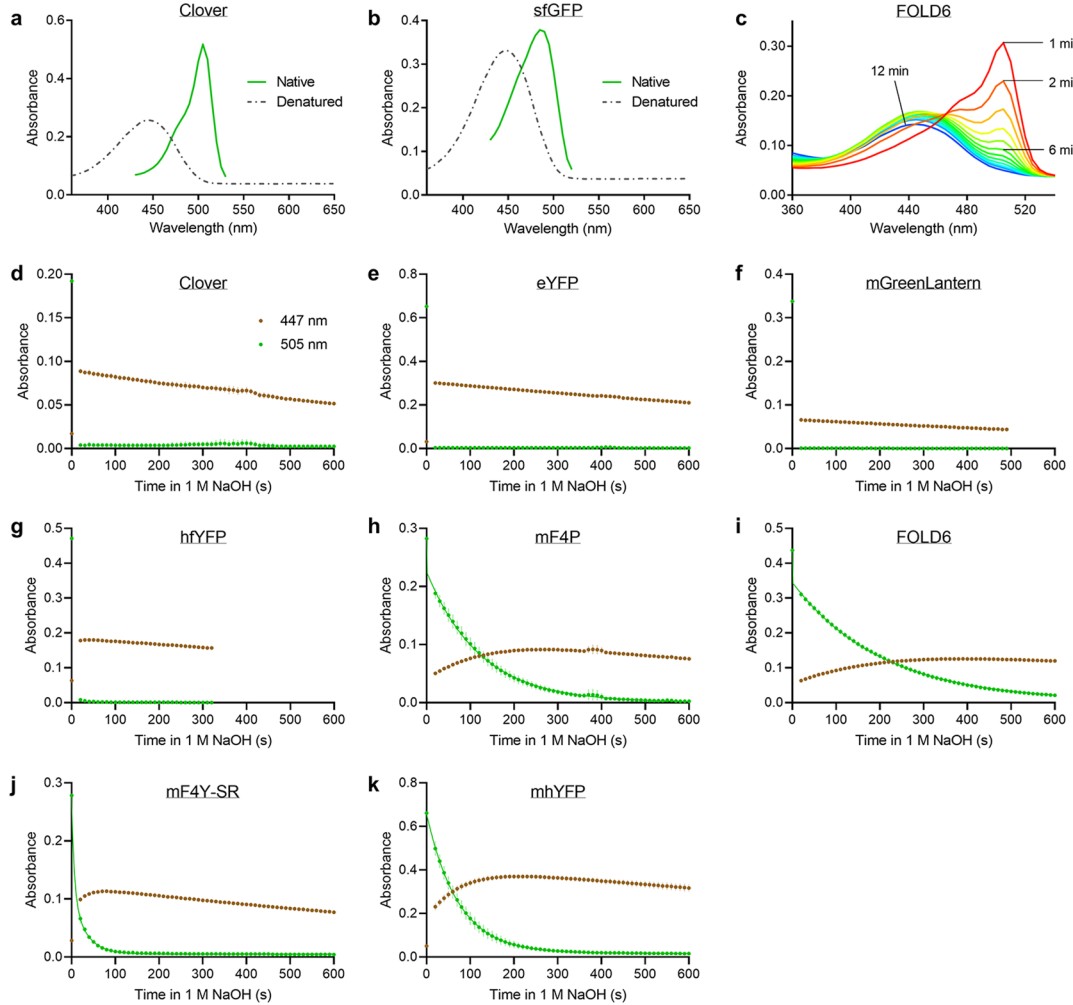

**Extended Data Fig. 5 | Behavior of FPs in sodium hydroxide solution.**
**a**, Clover and **b**, sfGFP absorbance scans of the native and alkali-denatured protein. **c**, Kinetic absorbance scans of the FOLD6 protein during alkali denaturation. Absorbance was collected every 1 min for 12 min. Mean ± s.d., n = 2 experiments. **d-k**, Time-dependent FP denaturation in 1 M NaOH (Supplementary Methods). Absorbance at 447 nm (brown) and 505 nm (green) was measured every 10 s, beginning promptly after mixing 2 M NaOH into a cuvette containing the same volume of FP solution ($A_{505} \approx 0.3{-}0.9$) in phosphate buffered saline (PBS) for a final concentration of 1 M NaOH. The initial value at

t = 0 s is the absorbance of the same native protein stock solution when diluted separately 1:1 into PBS without NaOH. Curve fits for 505 nm absorbance data: mono-exponential, mhYFP; double-exponential, mF4Y-SR, mF4P, FOLD6. Absorbance of the alkali-denatured avGFP-type chromophore peaks at ~447 nm ($\varepsilon_{447} = 44{,}000\ M^{-1}\ cm^{-1}$). The native absorbance maxima for all FPs in this figure are between 488–514 nm. Background subtracted absorbance values are shown on the $y$-axis (not normalized). Mean ± s.d., n = 3 experiments; n = 2 experiments for mGreenLantern and hfYFP.

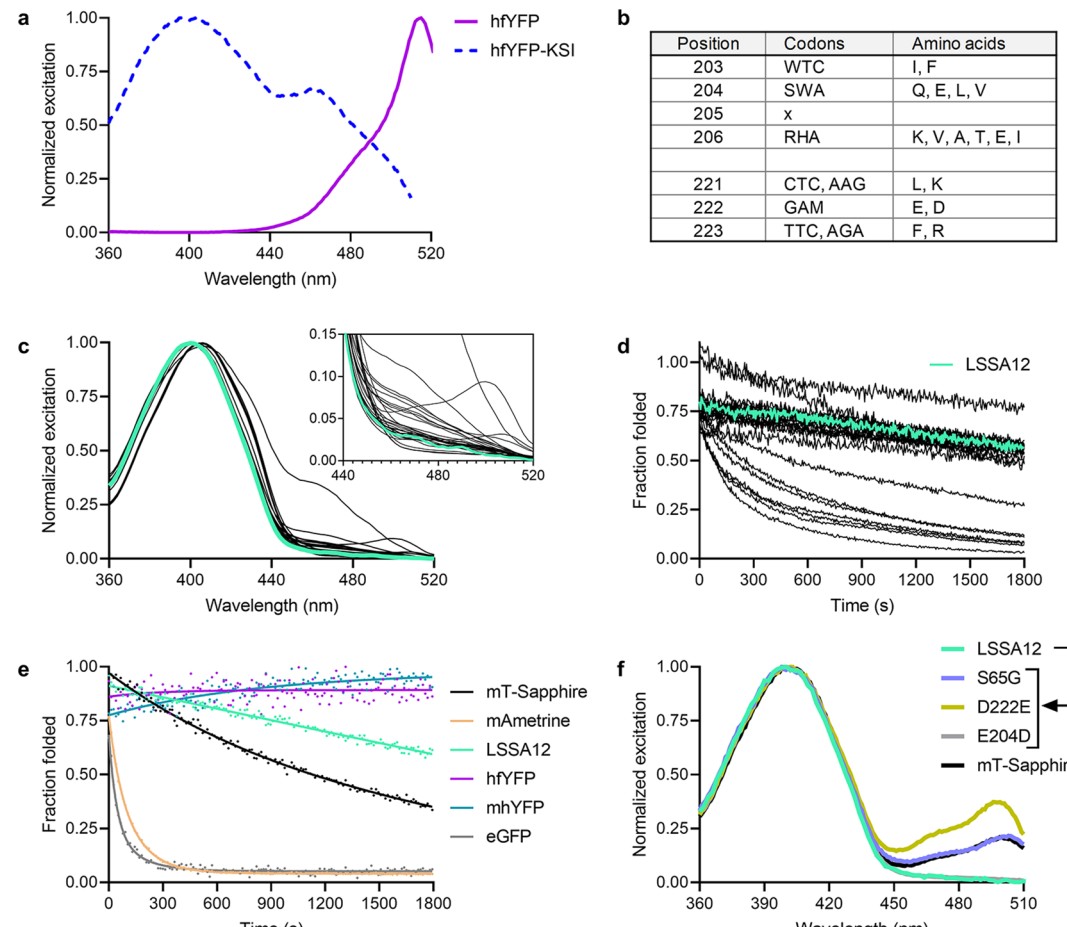

**Extended Data Fig. 6 | Library generation and screening of GFPs with a large Stokes shift. a**, Excitation spectrum of hfYFP and hfYFP-KSI (hfYFP-V206K/G65S/Y203I). The G65S/Y203I mutations gave rise to a 405 nm excitable band and an ~50 nm hypsochromic shift of the B-band. The V206K mutation does not alter the excitation spectrum. **b**, Degenerate codon sets chosen to mutate hfYFP-KSI at residues shown in Fig. 5a, b to produce an LSS FP without B-band excitation. **c**, Excitation spectra of 33 mutants from the library in **b** that were selected from LB-agar plates based on high 410-nm and low 470-nm excitation when observed by eye under LED illumination. Light green trace: LSSA12. Inset: zoom of the B-band excitation range. **d**, Kinetic unfolding of clarified lysate from 23 mutants from library **c** in 6.3 M GdnHCl, pH 7.5. **e**, Kinetic unfolding of purified FPs. All data points are shown with overlayed trend line (one-phase nonlinear regression). Mean values for n = 2 experiments. **f**, Excitation spectra of LSSA12 point mutants. The spectra reinforce the importance of residues S65 and D222 for eliminating the B-band excitation of LSSA12. That the excitation spectra of mT-Sapphire and LSSA12-S65G overlap, as do LSSA12 and LSSA12-E204D.

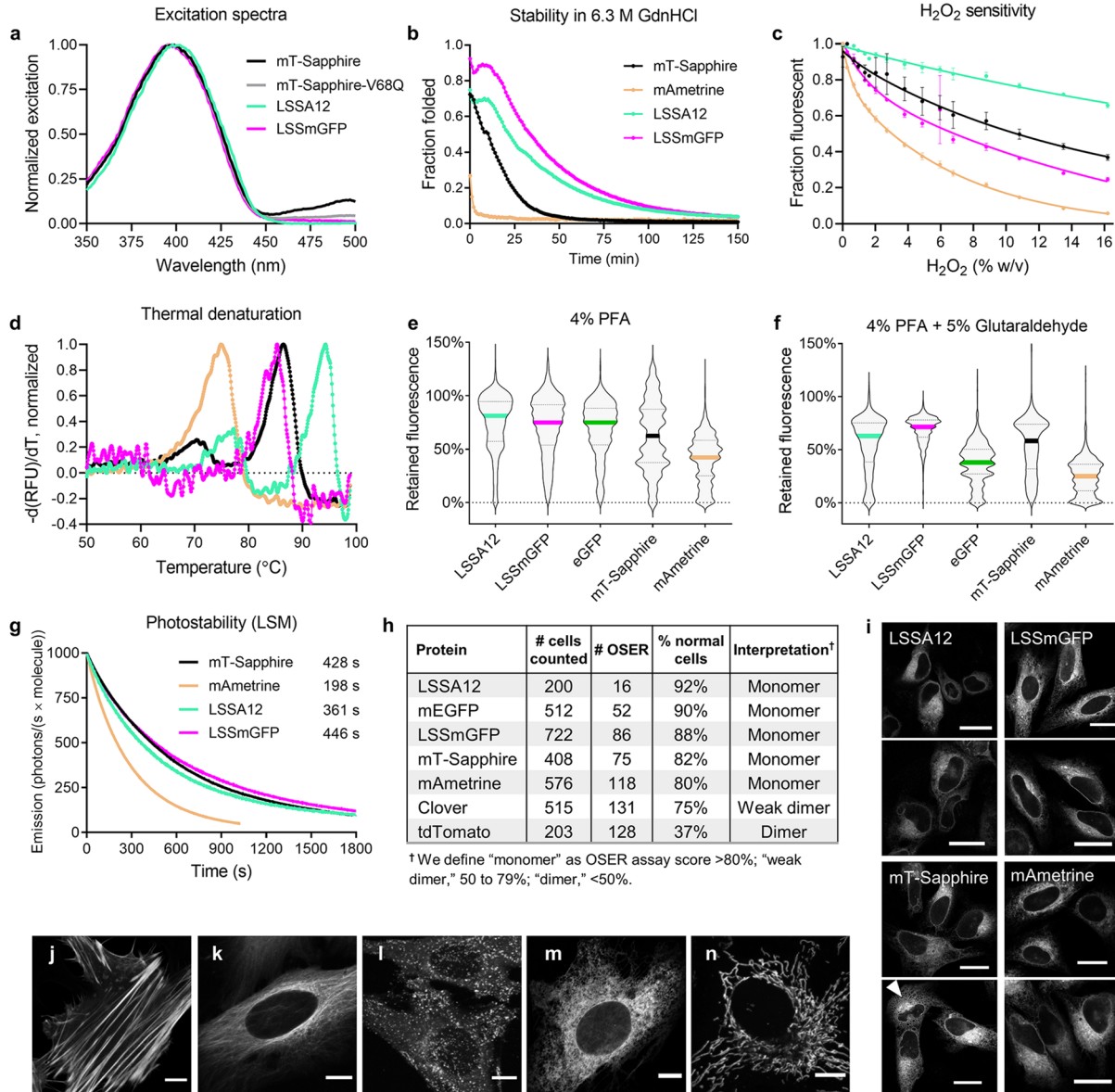

**Extended Data Fig. 7 | Additional characterization of LSSmGFP. a**, Excitation spectra of LSS FPs. Introducing the V68Q mutation into mT-Sapphire greatly diminished the 488 nm excitation band, suggesting generalizability of this LSSmGFP-derived mutation for spectral tuning of LSS FPs. **b**, Kinetic unfolding of purified FPs at 1 µM concentration in buffered GdnHCl 6.3 M solution, pH 7.4. Mean of n = 2 experiments. **c**, Purified FP at 0.1 µM concentration incubated at RT in 16 different concentrations of $H_2O_2$ in Tris buffer, pH 7.4, for exactly 15 min. All data points are shown and fit to a mono-exponential decay equation. Mean ± s.d., n = 2 experiments. **d**, Melting curve derivative plots generated from the thermofluor assay. Here, SYPRO Orange is used instead of endogenous fluorescence (Methods). $T_m$ values are reported in Table 1. **e**, Fluorescence retained by HEK293T cells after fixation using room temperature 4% PFA in PBS, pH 7.4, or **f**, 4% PFA + 5% Glut in PBS, pH 7.4. Cytosolic expression, n > 1100 cells for each condition per FP. Center line indicates the median. Statistics for **e** and **f** can

be found in Supplementary Fig. 8. **g**, Time to photobleach H2B-FP transfected HeLa cells from 1000 to 500 photons s⁻¹ molecule⁻¹ under confocal illumination (147 µW). Interpolated $t_{1/2}$ values in seconds are displayed next to each FP label in the figure legend. Mean values from n ≥ 30 cells per FP are plotted. **h**, OSER assay results analyzed using scoring criteria described in Extended Data Fig. 3d. **i**, Images of LSS FPs in the OSER assay when expressed from pCytERM-FP fusions. White wedge in the second mT-Sapphire image indicates a highly stressed cell with sheeted ER that was rarely seen in other FPs but was commonly encountered in mT-Sapphire. These cells were excluded from quantitation but might represent an important effect. Scale bars, 25 µm. **j-n**, Live HeLa cells imaged after overnight transfection using plasmids encoding: **j**, LifeAct-7aa-LSSmGFP; actin. **k**, LSSmGFP-6aa-tubulin; tubulin. **l**, LSSmGFP-15aa-clathrin; clathrin. **m**, pCytERM-LSSmGFP; endoplasmic reticulum. **n**, COX8A[×4]-4aa-LSSmGFP; mitochondria. Scale bars: 10 µm. Images from **i-n** are representative of at least 3 experiments.

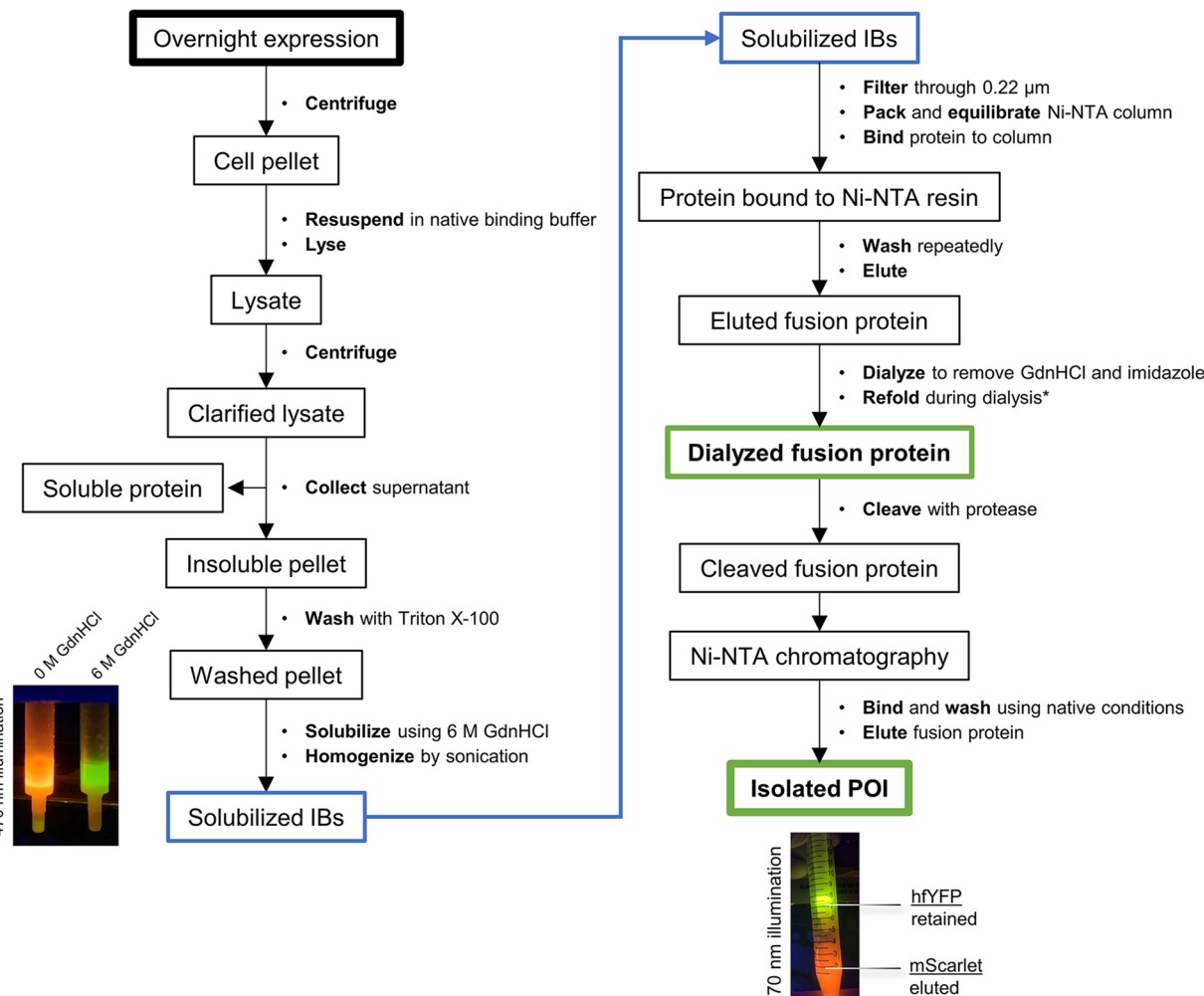

**Extended Data Fig. 8 | Benchtop fluorescence-assisted protein purification workflow.** As described in Methods, proteins were purified by Ni-NTA chromatography (immobilized metal affinity chromatography, or IMAC) under fully denaturing conditions with all buffers containing 6 M GdnHCl. If cleavage of the fusion protein is unimportant for downstream assays, the experiment is completed once the fusion protein is dialyzed out of denaturing purification buffer. Otherwise, the fusion construct illustrated in Fig. 5g is cleaved using TEV protease in a dialysis cassette (or after dialysis), and the unbound, cleaved protein of interest (POI) is isolated using IMAC—His$_6$-TEV and His$_6$-hfYFP are adsorbed onto the resin while the untagged and cleaved POI elutes in native buffer of choice for collection. All steps of purification can be visualized using 470 nm illumination for the hfYFP fusion (405 nm for the LSSmGFP fusion). The final image ('Isolated POI') depicts successful isolation of mScarlet from the hfYFP-mScarlet fusion in PBS after cleavage. Note (*): if the refolding buffer inhibits TEV protease activity, a separate dialysis step is recommended to improve activity. Similarly, if the TEV protease buffer selected is incompatible with IMAC (for example, if nontrivial amounts of DTT or EDTA are present), then a dialysis step should be performed after cleavage, before IMAC.

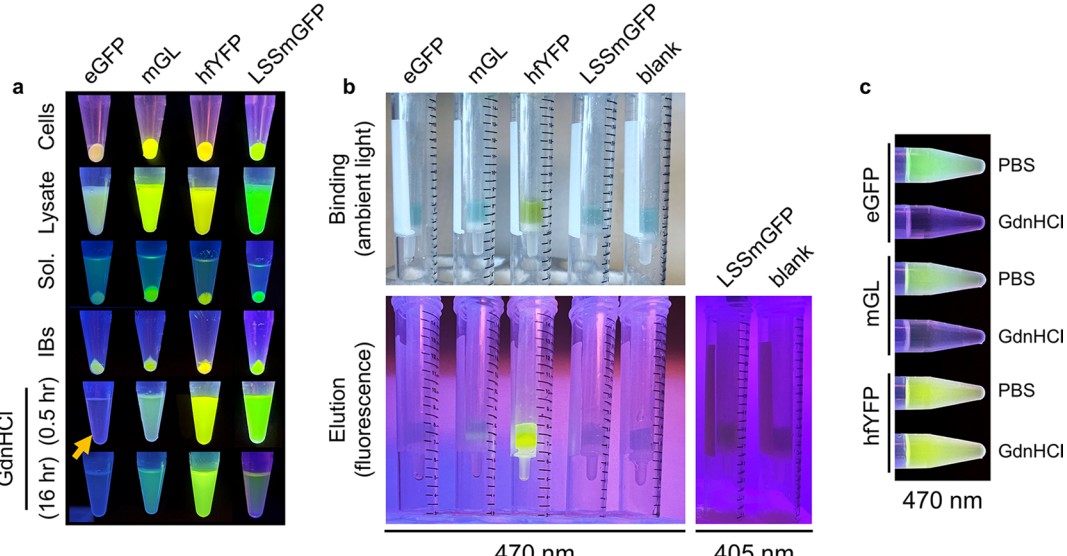

**Extended Data Fig. 9 | Brightness of FPs during fluorescence-assisted protein purification after overnight storage in 6 M GdnHCl at 4 °C. a**, Fluorescence-assisted purification of streptavidin (SAV) from *E.coli* inclusion bodies (IBs) using eGFP, mGL, hfYFP, or LSSmGFP fusions as depicted in Fig. 5g. The same protocol as in Fig. 5f was used, except here the FPs were refrigerated overnight at 4 °C after GdnHCl solubilization. eGFP is immediately denatured during IB solubilization using 6 M GdnHCl (orange arrow). hfYFP remains fluorescent throughout all stages of denaturing Ni-NTA purification (6 M GdnHCl present in all solutions) (see Extended Data Fig. 8 for workflow). FPs were photographed through an orange long-pass filter under overhead 470 nm LED illumination (405 nm for LSSmGFP) at the benchtop. **b**, Samples from **a** in denaturing solution under ambient light after binding to Ni-NTA resin. Fluorescence images taken after multiple wash steps show that mGL and LSSmGFP are largely quenched after overnight incubation in 6 M GdnHCl at 4 °C, whereas hfYFP fluorescence remains stable. **c**, Soluble fractions isolated from FPs that were expressed from the pBAD vector (no fusion). Samples were diluted in TNG buffer with or without GdnHCl (to 6 M final concentration) to the same final absorbance and photographed after overnight refrigeration. eGFP and mGL were mostly nonfluorescent after 16 hr at 4 °C, whereas hfYFP remained stable, indicating that it is the preferred FP for multi-day purification experiments. TNG buffer: 50 mM Tris-HCl, 150 mM NaCl, 10% glycerol, pH 7.4.

**Extended Data Table 1 | Biochemical characterization of selected FPs and their brightness in human cells.** Biochemical characterization of selected FPs and their brightness in human cells. Note: except for moxGFP, eGFP-C48S/C70V, and Clover-C48S/C70V, data above the 'FOLD4' row are cited from Campbell et al., *PNAS*, 2020, for the purpose of structure-activity comparison. All data in the table were generated in our lab in this study unless otherwise stated. $\phi$ = quantum yield; $\varepsilon$ = extinction coefficient at peak absorbance. N.D.: not determined. [a]Molecular brightness = $(\phi \times \varepsilon)/10^3$. [b]moxGFP is human codon-optimized sfGFP-C48S/C70V; we have re-characterized it for this study. [c]In BE(2)-M17 cells. See Supplementary Fig. 5g for results from additional human cell lines. [d]The quantum yield of moxGFP is cited from Costantini et al., *Nat. Comm.*, 2015. All other data in the table were generated in our lab. [e]Molecular brightness of moxGFP was determined using our EC value and the cited QY value. Mean±: s.d., n≥3 experiments; n=2 experiments for $T_m$ and $\lambda_{ex}/\lambda_{em}$ values

| Protein | $\lambda_{ex}$ | $\lambda_{em}$ | $\phi$ | ± | $\varepsilon$ (M⁻¹ cm⁻¹) | ± | Molecular brightness[a] | Cellular brightness[c] rel. eGFP | $pK_a$ | ± | $T_m$ (°C) | Maturation (min) |
|---|---|---|---|---|---|---|---|---|---|---|---|---|
| eGFP | 488 | 507 | 0.71 | 0.010 | 53,400 | 700 | 38 | 1.1 | 6.0 | 0.19 | 80.3 | 28 |
| eGFP-C48S/C70V | 488 | 510 | 0.68 | 0.041 | 48,200 | 1,500 | 32 | 0.0 | 6.0 | 0.17 | N.D. | N.D. |
| Superfolder GFP | 485 | 508 | 0.67 | 0.023 | 51,300 | 1,400 | 34 | 1.7 | 6.2 | 0.12 | 86.4 | 37 |
| moxGFP[b] | 487 | 509 | 0.58[d] | N/A | 52,500 | 3,200 | 30[e] | 0.9 | 6.1 | 0.12 | 79.5 | 50 |
| Clover | 505 | 515 | 0.75 | 0.048 | 96,600 | 1,400 | 72 | 1.8 | 6.5 | 0.21 | 79.5 | 15 |
| Clover-C48S/C70V | 506 | 516 | 0.75 | 0.029 | 99,700 | 6,100 | 75 | 2.4 | 6.5 | 0.17 | 76.6 | 31 |
| mClover3 | 506 | 516 | 0.80 | 0.015 | 100,000 | 2,800 | 80 | 1.6 | 6.1 | 0.02 | 80.1 | 37 |
| mNeonGreen | 506 | 516 | 0.78 | 0.037 | 115,400 | 2,200 | 90 | 4.1 | 5.6 | 0.25 | 68.0 | 18 |
| eYFP | 513 | 527 | 0.60 | 0.055 | 94,100 | 2,800 | 58 | 1.9 | 6.8 | 0.29 | 72.9 | >37 |
| FOLD1 | 503 | 514 | 0.70 | 0.016 | 99,700 | 2,000 | 70 | 3.4 | 5.8 | 0.14 | 87.6 | N.D. |
| mF1Y | 503 | 517 | N.D. | N.D. | 104,600 | 570 | 77 | 3.9 | N.D. | N.D. | 84.9 | 7 |
| mF1VPK | 503 | 513 | 0.75 | 0.013 | 107,700 | 3,200 | 81 | 3.2 | N.D. | N.D. | N.D. | N.D. |
| mF2BK (DMD) | 503 | 514 | 0.71 | 0.029 | 101,300 | 2,700 | 72 | 3.1 | 5.6 | 0.16 | 87.0 | 9 |
| **mGreenLantern** | 503 | 514 | 0.72 | 0.017 | 101,800 | 630 | 74 | 7.4 | 5.6 | 0.01 | 87.2 | 14 |
| mF3CPK | 503 | 513 | 0.72 | 0.014 | 100,700 | 4,500 | 73 | 2.4 | 5.9 | 0.27 | 86.5 | N.D. |
| FOLD4 | 504 | 513 | 0.69 | 0.015 | 108,200 | 3,300 | 75 | 3.5 | 6.0 | 0.09 | 86.2 | 16 |
| F4P | 503 | 513 | 0.76 | 0.017 | N.D. | N.D. | N.D. | 2.9 | N.D. | N.D. | 87.4 | N.D. |
| mF4P | 504 | 516 | 0.75 | 0.032 | 120,500 | 3,600 | 93 | 3.0 | 5.9 | 0.20 | 87.3 | 12 |
| mF4Pti | 506 | 516 | 0.77 | 0.041 | 118,200 | 5,300 | 91 | 1.7 | 6.3 | 0.08 | 85.2 | 25 |
| mF4Y | 504 | 516 | 0.80 | 0.038 | 100,000 | 2,000 | 83 | 3.1 | N.D. | N.D. | 85.2 | 16 |
| mF4Y-SR | 505 | 516 | 0.73 | 0.024 | 107,200 | 3,500 | 81 | 3.5 | 5.6 | 0.02 | 84.3 | 6 |
| mF4Y-RKH | 497 | 513 | 0.56 | 0.022 | N.D. | N.D. | N.D. | N.D. | 6.4 | 0.20 | 85.6 | N.D. |
| FOLD6 | 505 | 515 | 0.73 | 0.032 | 118,200 | 1,100 | 86 | 2.3 | 5.9 | 0.21 | 90.0 | 25 |
| mF6-H169L | 505 | 515 | 0.75 | 0.046 | 108,900 | 5,300 | 84 | 1.1 | 6.7 | 0.23 | 89.8 | N.D. |
| mfoxY | 512 | 529 | 0.51 | 0.006 | 110,000 | 3,200 | 56 | 2.7 | 5.5 | 0.04 | 92.8 | 25 |
| **hfYFP** | 514 | 529 | 0.60 | 0.020 | 119,500 | 1,400 | 74 | 2.6 | 5.6 | 0.23 | 94.2 | 21 |
| **mhYFP** | 515 | 529 | 0.62 | 0.019 | 124,000 | 3,200 | 77 | 2.5 | 5.7 | 0.02 | 92.8 | 27 |
| mfoxYtiPMKH | 509 | 514 | N.D. | N.D. | N.D. | N.D. | N.D. | N.D. | 8.5 | 0.02 | 88.9 | N.D. |
| mfoxY-Y203F | 508 | 520 | 0.72 | 0.035 | 120,000 | 3,400 | 89 | 1.1 | 5.6 | 0.03 | 94.5 | N.D. |

# nature research

# Reporting Summary

Nature Research wishes to improve the reproducibility of the work that we publish. This form provides structure for consistency and transparency in reporting. For further information on Nature Research policies, see our Editorial Policies and the Editorial Policy Checklist.

## Statistics

For all statistical analyses, confirm that the following items are present in the figure legend, table legend, main text, or Methods section.

| n/a | Confirmed | |
|---|---|---|
| ☐ | ☒ | The exact sample size (*n*) for each experimental group/condition, given as a discrete number and unit of measurement |
| ☐ | ☒ | A statement on whether measurements were taken from distinct samples or whether the same sample was measured repeatedly |
| ☐ | ☒ | The statistical test(s) used AND whether they are one- or two-sided *Only common tests should be described solely by name; describe more complex techniques in the Methods section.* |
| ☒ | ☐ | A description of all covariates tested |
| ☒ | ☐ | A description of any assumptions or corrections, such as tests of normality and adjustment for multiple comparisons |
| ☐ | ☒ | A full description of the statistical parameters including central tendency (e.g. means) or other basic estimates (e.g. regression coefficient) AND variation (e.g. standard deviation) or associated estimates of uncertainty (e.g. confidence intervals) |
| ☐ | ☒ | For null hypothesis testing, the test statistic (e.g. *F*, *t*, *r*) with confidence intervals, effect sizes, degrees of freedom and *P* value noted *Give P values as exact values whenever suitable.* |
| ☒ | ☐ | For Bayesian analysis, information on the choice of priors and Markov chain Monte Carlo settings |
| ☒ | ☐ | For hierarchical and complex designs, identification of the appropriate level for tests and full reporting of outcomes |
| ☒ | ☐ | Estimates of effect sizes (e.g. Cohen's *d*, Pearson's *r*), indicating how they were calculated |

*Our web collection on statistics for biologists contains articles on many of the points above.*

## Software and code

Policy information about availability of computer code

| Data collection | Biotek Gen5 software (v2.09.2), BZ-X Analyzer (v1.4.0.1), ZEN microsocopy software (latest version available at the time; late 2019). |
|---|---|
| Data analysis | GraphPad Prism 8, Microsoft Excel, XDS, Phaser, PHENIX, Coot |

For manuscripts utilizing custom algorithms or software that are central to the research but not yet described in published literature, software must be made available to editors and reviewers. We strongly encourage code deposition in a community repository (e.g. GitHub). See the Nature Research guidelines for submitting code & software for further information.

## Data

Policy information about availability of data

All manuscripts must include a data availability statement. This statement should provide the following information, where applicable:

- Accession codes, unique identifiers, or web links for publicly available datasets
- A list of figures that have associated raw data
- A description of any restrictions on data availability

Crystal structures have been uploaded to the PDB under accession codes 7UGR (hfYFP), 7UGS (mhYFP), and 7UGT (FOLD6). Plasmids have been deposited at Addgene. DNA sequences of the fluorescent proteins have been uploaded to GenBank: OP373686 (hfYFP), OP373687 (mhYFP), OP373688 (LSSA12), OP373689 (LSSmGFP). Additional data that support the findings of this study are available from the corresponding author upon reasonable request.

# Field-specific reporting

Please select the one below that is the best fit for your research. If you are not sure, read the appropriate sections before making your selection.

☒ Life sciences ☐ Behavioural & social sciences ☐ Ecological, evolutionary & environmental sciences

For a reference copy of the document with all sections, see nature.com/documents/nr-reporting-summary-flat.pdf

# Life sciences study design

All studies must disclose on these points even when the disclosure is negative.

| | |
|---|---|
| Sample size | Biochemical and spectroscopic assays involving purified protein were highly reproducible, repeated >=2-3 times. Technical replicates were used within replicate experiments, and sample sizes and methodology are indicated. For the experiment involving fluorescence retention after fixation, whole tile scans of chamber slides were used for quantification. For the CLEM experiment, multiple random FOVs were used for scoring. Cellular assays were repeated on different days and the data were averaged as indicated. In some figures, error bars are omitted for clarity, such as when thousands of data points are present from kinetic measurements. Number of experimental replicates are listed in the figure legends. Sample sizes are comparable to field standard assays from the most recent several years for cell-based characterization experiments. We concluded that one experiment (glyoxal fixation) showed over-expression artifacts, and a decision was made during revision to remove the experiment, it was not repeated. |
| Data exclusions | Data were not excluded. |
| Replication | We have found the experiments to be reproducible. Experiments were typcially repeated several times with consistent results. |
| Randomization | No experimental groups, so no need for randomization. |
| Blinding | No experimental groups, so no need for blinding. |

# Reporting for specific materials, systems and methods

We require information from authors about some types of materials, experimental systems and methods used in many studies. Here, indicate whether each material, system or method listed is relevant to your study. If you are not sure if a list item applies to your research, read the appropriate section before selecting a response.

### Materials & experimental systems

| n/a | Involved in the study |
|---|---|
| ☐ | ☒ Antibodies |
| ☐ | ☒ Eukaryotic cell lines |
| ☒ | ☐ Palaeontology and archaeology |
| ☒ | ☐ Animals and other organisms |
| ☒ | ☐ Human research participants |
| ☒ | ☐ Clinical data |
| ☒ | ☐ Dual use research of concern |

### Methods

| n/a | Involved in the study |
|---|---|
| ☒ | ☐ ChIP-seq |
| ☒ | ☐ Flow cytometry |
| ☒ | ☐ MRI-based neuroimaging |

## Antibodies

| | |
|---|---|
| Antibodies used | Gt a GFP (Abcam, #ab6673); Gt a GFP (Novus, #NB1001770); Ms a GFP (Invitrogen, #A-11120) |
| Validation | These are commonly used, highly validated commercial antibodies. |

## Eukaryotic cell lines

Policy information about cell lines

| | |
|---|---|
| Cell line source(s) | HEK293T (ATCC, CRL-3216), HeLa (ATCC, CCL-2), and BE(2)-M17 cells (ATCC, CRL-2267) |
| Authentication | Provided by ATCC and authenticated by morphology comparison to ATCC sample images. |
| Mycoplasma contamination | Cells were determined mycoplasma negative using a commercial fluorescence kit. |
| Commonly misidentified lines (See ICLAC register) | To our knowledge and from ICLAC, these cell lines are not commonly misidentified. |

