## [Peer Review File · Nature Methods]

Peer Review Information

Manuscript Title: Ultrastable fluorescent proteins for advanced microscopy

Corresponding author name(s): Benjamin C. Campbell

Editorial Notes:

Reviewer Comments & Decisions:

Decision Letter, initial version:

Dear Ben,

Thank you for your letter detailing how you would respond to some of the reviewer concerns regarding your Article, "Ultrastable fluorescent proteins for advanced microscopy". We have decided to invite you to revise your manuscript as you have outlined, before we reach a final decision on publication. With regards to the 2-color CLEM experiment, we think this would be a nice addition, but only if it can be achieved in a reasonable time (say 3 months or less).

- * include a point-by-point response to the reviewers and to any editorial suggestions
- * please underline/highlight any additions to the text or areas with other significant changes to facilitate review of the revised manuscript
- * address the points listed described below to conform to our open science requirements

* ensure it complies with our general format requirements as set out in our guide to authors at www.nature.com/naturemethods

* resubmit all the necessary files electronically by using the link below to access your home page

[Redacted] This URL links to your confidential home page and associated information about manuscripts you may have submitted, or that you are reviewing for us. If you wish to forward this email to co-authors, please delete the link to your homepage.

We hope to receive your revised paper within 3 months. If you cannot send it within this time, please let us know. In this event, we will still be happy to reconsider your paper at a later date so long as nothing similar has been accepted for publication at Nature Methods or published elsewhere.

OPEN SCIENCE REQUIREMENTS

REPORTING SUMMARY AND EDITORIAL POLICY CHECKLISTS

IMAGE INTEGRITY

DATA AVAILABILITY

Please include a “Data availability” subsection in the Online Methods. This section should inform readers about the availability of the data used to support the conclusions of your study, including accession codes to public repositories, references to source data that may be published alongside the paper, unique identifiers such as URLs to data repository entries, or data set DOIs, and any other statement about data availability. At a minimum, you should include the following statement: “The data that support the findings of this study are available from the corresponding author upon request”, describing which data is available upon request and mentioning any restrictions on availability. If DOIs are provided, please include these in the Reference list (authors, title, publisher (repository name), identifier, year). For more guidance on how to write this section please see:

<http://www.nature.com/authors/policies/data/data-availability-statements-data-citations.pdf>

MATERIALS AVAILABILITY

ORCID

Sincerely,
Rita

Rita Strack, Ph.D.
Senior Editor
Nature Methods

Reviewers' Comments:

Reviewer #1:

In this manuscript Benjamin C. Campbell et al. developed extremely chemically- and thermodynamically-stable yellow hfYFP and green LSSmGFP fluorescent proteins, solved their crystal structures and successfully applied them for the proExM and CLEM microscopy techniques. First, authors developed chemically-stable yellow hfYFP protein using rational mutagenesis followed by screening for high GdnHCl stability and thermostability. Benjamin C. Campbell et al. further characterized spectral properties of the hfYFP protein in vitro and its brightness and behavior in fusions when expressed in mammalian cells. Authors next demonstrated superior chemical-stability of the hfYFP and its monomeric variant mhYFP in various denaturants (such as GdnHCl, GdnSCN, temperature, and H₂O₂) as compared to other FPs. Benjamin C. Campbell et al. next showed that hfYFP can successfully withstand conditions used in expansion microscopy (proExM), histology, and electron microscopy techniques. Authors also solved crystal structures of hfYFP, mhYFP and FOLD6 proteins and suggested structural basis of their

high stability. Using these structures they developed green fluorescent proteins with a large Stokes shift, LSSmGFP and LSSA12, which have superior thermal and chemical stability and ensure cross-bleed-free two-color live cells imaging as compared to available mT-Sapphire and EGFP proteins. Finally, Benjamin C. Campbell et al. applied the hfYFP and LSSmGFP proteins as fusion tags to enable fluorescence-assisted protein purification under denaturing conditions. Overall, the way of the protein engineering and a set of the stable green and yellow fluorescent proteins developed in this manuscript are interesting for broad readers from different fields. Besides expansion and electron microscopy, hfYFP can be potentially applicable to bimolecular fluorescence complementation, super-resolution imaging techniques (STED nanoscopy) and other advanced microscopy methods and biological applications demanding fluorescent proteins that can preserve its fluorescence under rigid sample preparation conditions, such as tissue clearing. hfYFP and LSSmEGFP proteins can also be used for the development of the stable and expression-enhanced indicators.

Minor points:

1. Main text, line 28. Misprint "...in advances microscopy methods..." should be replaced with "...in advanced microscopy methods..."
2. Main text, lines 133 and 389. In the main text and Table S3 authors compared stabilities of sfGFP and hfYFP in 7M GdnHCl but in Fig2a legend the same comparison is shown for 6.3 M GdnHCl. Please, correct "7M" for "6.3 M".
The same wrong 7M concentration is mentioned below in line 389.
3. Main text, line 182. mhYFP is mentioned in line 182 but on Figure 3a hfYFP is shown. Please, correct "mhYFP" for "hfYFP".
4. Main text, lines 232-233 and Figure 4f and 4g. hfYFP fluorescence is discussed in the text but Fig.4f and 4g are related with mhYFP. Please, correct.
5. Main text, line 261. Please, correct misprint "... the possibility surface..." to "...the possibility of surface..."
6. Main text, line 267 and Figure S11a. For better understanding of impact of T65S mutation, T65 residue might be included on Figure S11a.
7. Figure S11b and S11c. Please, define crosses on the Fig S11b,c.
8. Main text, line 306 and Figure S15. I couldn't find Fig. S15l panel. Please, correct.
Also, I suggest that Fig S15k panel corresponds to mitochondria localization and Fig S15j – to endoplasmic reticulum. Please, correct by swapping panel j and k on Figure S15.
9. Main text, line 322. Authors cite Fig. S15b to refer to the high HdnHCl stability on the LSSmGFP protein. However, Fig. 15b describes thermal stability of the LSSmGFP protein. Reference to Fig.S15c would be appropriate in case authors correct X-axis label on Fig. S15c from "H2O2 (%/w/v)" to "Time".
10. Error-prone libraries, line 923. Could authors explain the meaning of the following phrase: "the reaction was PCR purified"?
11. Line 1558, Figure S3d. Green line corresponding to "Clover-cc" is not seen in the Figure S3d.

Also, on the Figure S3d I could not ascribe grey line (3rd from the bottom) to any of FPs shown on Figure S3d.

12. For convenience of the readers, I would recommend to provide supplementary Figure containing alignment of the amino acids sequences for the major developed proteins, i.e. at least for hfYFP, mhYFP, LSSmGFP and LSSA12.

Overall, the publication in Nature Methods is appropriate and highly recommended.

Reviewer #2:

Remarks to the Author:

Based on the previously developed mGreenLantern (mGL), the authors developed the well-folded cysteine-free FP hyperfolder YFP (hfYFP) and monomeric mhYFP, and investigated the applications and performances of these FPs in advanced microscopy methods such as proExM and CLEM. Next, they solved the crystal structures of hfYFP, mhYFP and a cysteine-free mGL variant called FOLD6, and engineered two FPs with large Stokes shift (LSSA12 and LSSmGFP) based on these structures. Finally, the authors successfully visualized the different steps of protein purification using the chemical stability of hfYFP and LSSmGFP.

Compared with the existing common FPs (eGFP, eYFP, mClover3, and mNeonGreen, etc.), the chemical stabilities of hfYFP and mhYFP have been improved greatly, but these properties have no significant advantages compared with those of mGL, and some properties are even worse. For example, compared with mGL (Table S2), the molecular brightnesses of hfYFP and mhYFP are about the same as that of mGL, but the cellular brightness is only 1/3 of mGL, and T_m is only slightly increased (92.8 vs 87.2).

In specific advanced imaging applications, mhYFP does not possess any single property or performance that cannot be surpassed by the best FPs reported. For example, in the application of proExM, hfYFP/mhYFP has no greater advantage than mGL. Conversely, as the authors described, "mGreenLantern, with its 6.0-fold greater fluorescence than eGFP in mammalian cells compared to 2.4-fold for hfYFP, may be best suited for proExM." Nor does mhYFP outperform mEos4b in CLEM applications, although it survives CLEM sample preparation but its fluorescence retention only matches mEos4b's. hfYFP performs better, but it is not a monomer. Since samples imaged with one method (eg, ExM) are difficult to image with another method (eg, CLEM) due to the different handling of samples in different applications or imaging techniques, it is obvious that, to obtain the best imaging results, users will choose mGL for ExM imaging, mEosEM/mEos4b for CLEM imaging, and not mhYFP. Most of the time, what we need is a single event champion, not an all-around champion.

Last, in biological applications, the authors did not provide biological new insights or discoveries using the developed FPs.

Main concern :

1. The manuscript mainly focused on hfYFP. However, because hfYFP has weak dimerization property, it is not optimal as a fusion tag to label target proteins, especially when the target protein is a membrane protein, has weak dimerization property, or has a high expression level, labeling with hfYFP will affect the localization of the target protein. This is even more important for imaging techniques such as ExM and CLEM that require more precise and accurate localization of target protein. Obviously, the author also realized this problem and developed the monomeric mhYFP. The photochemical properties of mhYFP are the same as those of hfYFP, but mhYFP is a real monomer. Therefore, this manuscript should use mhYFP, rather than hfYFP, as the final tagging FP and compare the properties of mhYFP with other FPs. In addition, oligomeric properties tend to increase the extension coefficient and Mol. bright of FPs. The comparison of these properties in the text table should include mhYFP. In addition, mhYFP mutants are derived from mGL, which must be added as a control when comparing the properties and performances of different FPs.

2. In order to highlight certain chemical properties or application advantages of the developed FPs, the authors biased selection of some FPs for comparison, but did not select the best FPs with specific properties that have been reported. For example, mGL is the previously developed FP that survived GdnHCl treatment and proExM (PNAS, 2020, Benjamin C. Campbell, et al), in which mGL has already been shown to have obvious advantages over eGFP and is currently the best FP for proExM. hfYFP and mhYFP are from mGL, however when the authors compared certain biochemical properties of different FPs, some FPs were selected for comparison with a biased selectivity. In detail, in Figure 1b-1g the FPs selected for comparison were different, and mGL was not selected for comparison. mGL is only compared in Table S2 for biochemical properties and brightness. The mF4Y-SR mutant suddenly appeared only in Fig.3e and not in other Fig.3 figures. In Fig. 4d-g, mEos4b was not selected for comparison as a control to demonstrate the CLEM imaging performance of mhYFP in cells. For another example, based on the anti-GdnHCl properties of hfYFP and LSSmGFP, the authors propose that they can be used to achieve fluorescence-assisted benchtop purification of inclusion body proteins under fully denaturing conditions. As a template, mGL has anti-GdnHCl properties. When visualizing the different steps of protein purification using hfYFP and LSSmGFP (Fig. 5), it is necessary to compare with mGL and prove that the performances are better than mGL. Also, mAmetrine should be selected as a control in Fig. S15a. etc.

Additionally, the authors' claim in the Discussion that "we produced a series of avFP mutants demonstrating superior stability over sfGFP in every assay that we performed." is incorrect. Many figures in the manuscript used eGFP instead of sfGFP as a control (Fig. 1c, Fig. 3a-f, Fig. 4d-g, Fig. 5f). It is more reasonable to use sfGFP as a control rather than eGFP when discussing the stability of FPs and their applications. Taking Fig.5f and Fig.S17 as examples, when used as solubility tags, the aggregation properties of FPs themselves will affect the solubility properties of POIs. eGFP is a weak dimer. mGL and mEGFP are monomers. It is not reasonable to compare hfYFP and LSSmGFP with eGFP rather than mEGFP/sfGFP.

Minor concern

1. When I first saw "ultrastable" in the title I thought "photostable". I recommend using "chemical stable" instead of "ultrastable".
2. There is no representative cell image by proExM using different FPs in Fig.3.
3. mEos4b is not the best OsO₄-tolerant fluorescent protein. In Fig.4 the authors need to compare mhYFP with mEosEM (Nature Methods, 2020). The performances of these FPs in GMA and Epon-embedded samples also need to be compared.
4. How do LSSmGFP and LSSA12 perform in ExM applications compared to mhYFP?
5. The title of the manuscript is "Ultrastable FPs for advanced microscopy". Because super-resolution (SR) imaging is an important component of advanced microscopy methods, as the authors described in their introduction, the authors need to discuss the application of the developed FPs in SR and compare their advantages and disadvantages with existing proteins.
6. The quality of images by advanced microscopies depends on the retained fluorescence of FP. Since different FPs have different molecular and cellular brightness, the retained fluorescence relative to eGFP should be compared in Fig.3 and Fig.4 as in Fig.1e.
7. The photostability of mGL is not good. How about hfYFP?
8. Fig.4: 1) Fig.4d and 4e: Inconsistent with the description of Line 231: which one is before? Which one is after? 2) hfYFP is used in d and e, but mhYFP is used in statistical graphs f and g; mhYFP is marked in h, but hfYFP is described in the text (Line 234-235). 3) The fluorescence signals of eGFP and mGL in Fig. 4h are very strong, and the significant advantage of mhYF over eGFP and mGL cannot be seen from the figure.
9. Sometimes eGFP is used in the manuscript, and sometimes EGFP is used, which needs to be unified.
10. mAmetrine should be compared in Fig.5 and Fig.S15. mAmetrine and mGL are a good pair, which shows no bleed-through between the 405 nm and 470 nm channels. And there are many FPs reported for dual-color imaging. The advantages of the mGL/LSSmGFP pair is not significant when it is only used for the general dual-color imaging techniques. The authors should demonstrate the advantages of the mhYFP/LSSmGFP pair in applications based on their chemical stability, for example in ExM imaging.
11. There are two bs in Table S3.
12. It appears that the localization of mhYFP in Fig. 3f is different from those of the other proteins. Does mhYFP labeling affect H2B localization? More localization images of mhYFP need to be shown as in Fig.1h-1m.
13. The proportion of normal cells for Clover measured in Fig. S6 is 63%, while that in Fig. S15f is 75%. How to explain such a big difference? The proportion of normal cells for Clover is 75% (Fig. S15f), which is defined as weak dimer. However, in Fig. S6 hfYFP-V206K (79%) and mhYFP (80%) were defined as monomers. What is the basis for defining the aggregation state of a FP as weak dimer or monomer?
14. In Fig.S2, there is a typo that S147P should be S147R.

Reviewer #3:

Remarks to the Author:

This manuscript describes the engineering of robustly folding and super stable fluorescent proteins that can withstand harsh chemical conditions and high temperatures. Such proteins would have broad applications in expansion microscopy, correlated light and electron microscopy, and as visualization tags during protein purification. The manuscript is a tour-de-force and presents a comprehensive analysis of structural features that contribute to rapid folding and high stability. The supplementary information is a treasure trove of valuable insights that would be extremely useful for the broader protein engineering community. The tools themselves (hfYFP, mhYFP, LSSmGFP, LSSA12) appear to present notable strengths over currently available FPs for advanced applications. The experiments are well described. The methods and analysis are thorough and rigorous. The conclusions are well-supported by the data presented. The referencing is appropriate. The manuscript is clear and easy to follow. I think this is a strong contribution and I support publication. There are a few minor issues (noted below) that would improve clarity.

- 1) I don't understand why there is an increase in fluorescence for mGreenLantern and Hyperfolder YFP (Fig 2a and 2c). Can the authors speculate on/explain the origin of this increase?
- 2) Figure 3: (c) and (f) would benefit from inclusion of a scale bar. The quality of the images is poor in my pdf (even when zoomed in on a computer screen) – but the morphology of mhYFP in (f) appears drastically different (it looks punctate?) compared to the others. Is there a localization/morphology difference? This seems important to evaluate for histology purposes. Lastly, given the variability in signal in (a), (d), and (e), these would benefit from statistical analysis.
- 3) Figure 4: I think scale bars are present in (d) and (e) but they are so tiny I can't read them. Please add visible, readable scale bars.
- 4) The manuscript would benefit from some sort of qualified statement about the potential risks of super-stable proteins. There is a lot of accumulated knowledge in the FP community about what FP to use and what FP to avoid for a particular application. These nuances are not always passed on to the broader biological community. For example, the community generally accepts that over long periods of time mCherry can be toxic and can accumulate in lysosomes. The protein appears not to be efficiently broken down in the lysosome and because of mCherry's low pKa, it retains fluorescence. I have heard discussions at meetings and workshops that perhaps mCherry isn't efficiently degraded because it is too stable. I imagine the same problem could occur for fusions to super stable proteins. This may not be an issue if users are fusing hfYFP to a long lived structural protein to be visualized in CLEM or proExM, but it could be a significant issue on a short-lived signaling protein. It seems wise to include a qualified statement that users should check whether fusion alters the protein stability/half-life.

Author Rebuttal to Initial comments

Responses are in blue

Reviewers' Comments:

Reviewer #1:

In this manuscript Benjamin C. Campbell et al. developed extremely chemically- and thermodynamically-stable yellow hfYFP and green LSSmGFP fluorescent proteins, solved their crystal structures and successfully applied them for the proExM and CLEM microscopy techniques. First, authors developed chemically-stable yellow hfYFP protein using rational mutagenesis followed by screening for high GdnHCl stability and thermostability. Benjamin C. Campbell et al. further characterized spectral properties of the hfYFP protein in vitro and its brightness and behavior in fusions when expressed in mammalian cells. Authors next demonstrated superior chemical-stability of the hfYFP and its monomeric variant mhYFP in various denaturants (such as GdnHCl, GdnSCN, temperature, and H₂O₂) as compared to other FPs.

Benjamin C. Campbell et al. next showed that hfYFP can successfully withstand conditions used in expansion microscopy (proExM), histology, and electron microscopy techniques. Authors also solved crystal structures of hfYFP, mhYFP and FOLD6 proteins and suggested structural basis of their high stability. Using these structures they developed green fluorescent proteins with a large Stokes shift, LSSmGFP and LSSA12, which have superior thermal and chemical stability and ensure cross-bleed-free two-color live cells imaging as compared to available mT-Sapphire and EGFP proteins.

Finally, Benjamin C. Campbell et al. applied the hfYFP and LSSmGFP proteins as fusion tags to enable fluorescence-assisted protein purification under denaturing conditions. Overall, the way of the protein engineering and a set of the stable green and yellow fluorescent proteins developed in this manuscript are interesting for broad readers from different fields. Besides expansion and electron microscopy, hfYFP can be potentially applicable to bimolecular fluorescence complementation, super-resolution imaging techniques (STED nanoscopy) and other advanced microscopy methods and biological applications demanding fluorescent proteins that can preserve its fluorescence under rigid sample preparation conditions, such as tissue clearing. hfYFP and LSSmEGFP proteins can also be used for the development of the stable and expression-enhanced indicators.

Minor points:

1. Main text, line 28. Misprint “...in advances microscopy methods...” should be replaced with “...in advanced microscopy methods...”
2. Main text, lines 133 and 389. In the main text and Table S3 authors compared stabilities of sfGFP and hfYFP in 7M GdnHCl but in Fig2a legend the same comparison is shown for 6.3 M GdnHCl. Please, correct “7M” for “6.3 M”.
The same wrong 7M concentration is mentioned below in line 389.
3. Main text, line 182. mhYFP is mentioned in line 182 but on Figure 3a hfYFP is shown. Please, correct “mhYFP” for “hfYFP”.
4. Main text, lines 232-233 and Figure 4f and 4g. hfYFP fluorescence is discussed in the text but Fig.4f and 4g are related with mhYFP. Please, correct.
5. Main text, line 261. Please, correct misprint “... the possibility surface...” to “...the possibility of surface...”
6. Main text, line 267 and Figure S11a. For better understanding of impact of T65S mutation, T65 residue might be included on Figure S11a.
7. Figure S11b and S11c. Please, define crosses on the Fig S11b,c.
8. Main text, line 306 and Figure S15. I couldn't find Fig. S15l panel. Please, correct.
Also, I suggest that Fig S15k panel corresponds to mitochondria localization and Fig S15j – to endoplasmic reticulum. Please, correct by swapping panel j and k on Figure S15.
9. Main text, line 322. Authors cite Fig. S15b to refer to the high HdnHCl stability on the LSSmGFP protein. However, Fig. 15b describes thermal stability of the LSSmGFP protein. Reference to Fig.S15c would be appropriate in case authors correct X-axis label on Fig. S15c from “H2O2 (%/w/v)” to “Time”.
10. Error-prone libraries, line 923. Could authors explain the meaning of the following phrase: “the reaction was PCR purified”?
11. Line 1558, Figure S3d. Green line corresponding to “Clover-cc” is not seen in the Figure S3d.
Also, on the Figure S3d I could not ascribe grey line (3rd from the bottom) to any of FPs shown on Figure S3d.
12. For convenience of the readers, I would recommend to provide supplementary Figure containing alignment of the amino acids sequences for the major developed proteins, i.e. at least for hfYFP, mhYFP, LSSmGFP and LSSA12.

Overall, the publication in Nature Methods is appropriate and highly recommended.

Reviewer #2:

Remarks to the Author:

Based on the previously developed mGreenLantern (mGL), the authors developed the well-folded cysteine-free FP hyperfolder YFP (hfYFP) and monomeric mhYFP, and investigated the applications and performances of these FPs in advanced microscopy methods such as proExM and CLEM. Next, they solved the crystal structures of hfYFP, mhYFP and a cysteine-free mGL variant called FOLD6, and

engineered two FPs with large Stokes shift (LSSA12 and LSSmGFP) based on these structures. Finally, the authors successfully visualized the different steps of protein purification using the chemical stability of hfYFP and LSSmGFP.

Compared with the existing common FPs (eGFP, eYFP, mClover3, and mNeonGreen, etc.), the chemical stabilities of hfYFP and mhYFP have been improved greatly, but these properties have no significant advantages compared with those of mGL, and some properties are even worse.

It is very rare that any one reagent is the best-in-class along all dimensions. Rather, reagents will be best along one or several dimensions, with the specific experiments guiding reagent choice. This is indeed the case here as well. We do not believe, and did not mean to suggest, that hfYFP and mhYFP are superior to all other FPs along all axes. Rather, hfYFP and mhYFP are best-in-class for some specific applications, which enable new types of experiments. If the reviewer has in mind any specific language from the manuscript that seems overstated, please refer us directly to that and we will amend.

Specifically, we believe that (m)hfYFP:

- Has incredible thermodynamic stability, significantly higher than mGL.
- Performs dramatically better than mGL in EM (we do not claim superiority to mEos4, mEosEM, etc.). mGL was effectively non-functional in our EM conditions.
- Allows simple and robust protein purification. We did not specifically test mGL in this assay, but its cysteines (of which (m)hfYFP has none) will likely complicate the folding and trafficking of fusions.
- Allows robust refolding, which again mGL may struggle with due to cysteine content.
- Has substantially lower pKa (~5.6) than most commonly used FPs (~6-6.5), providing more acid-insensitivity in uses around physiological pH.
- Will serve as templates for biosensor engineering, given the crystal structures and structure-activity relationships.

We also note that we have compared hfYFP to another monomeric cysteine-less avFP, moxGFP, which has been widely advertised for its performance in the secretory pathway. As summarized in manuscript lines 112-116, hfYFP has twice the molecular brightness, three times the brightness in cells, lower pKa, improved thermodynamic stability, twice the speed of chromophore maturation, slower unfolding, and better refolding, and it also contains no cysteines. Therefore, we expect hfYFP to outperform the most popular cysteine-less “mox” FP in similar use cases while offering many additional advantages.

For example, compared with mGL (Table S2), the molecular brightnesses of hfYFP and mhYFP are about the same as that of mGL, but the cellular brightness is only 1/3 of mGL, and T_m is only slightly increased (92.8 vs 87.2).

That is only one of the many measures of thermodynamic stability that we measured. In many assays, the thermodynamic stability of (m)hfYFP was markedly higher than mGL:

- In Fig. 2a, (m)hfYFP never unfolded in 6.3 M GdnHCl (up to 3 months), whereas mGL unfolded fairly quickly (although more slowly than EGFP and mF4Y). See also Fig. S9.
- In Fig. 2b, (m)hfYFP unfolded much more slowly than mGL in 3.6 M GdnSCN.
- In Fig. 2e, isothermal unfolding showed that mGL unfolds much more quickly than (m)hfYFP at 87C. See also Fig. S7; holds for $T \geq 70C$.
- In Fig. 2h, (m)hfYFP was much more stable than mGL in 1-20% H₂O₂.
- In Fig. S5, (m)hfYFP provided a higher fraction of soluble protein than mGL.
- In Fig. S8, (m)hfYFP was more stable than mGL in 2 M NaOH.
- See also Table S3 for quantification of previous measurements.

In specific advanced imaging applications, mhYFP does not possess any single property or performance that cannot be surpassed by the best FPs reported.

We respectfully disagree with the reviewer. The thermodynamic stability of (m)hfYFP is far and away the largest of any FP in the literature. This allowed the targeted removal of cysteine residues, which has led to dramatic destabilization of previous GFP-based fluorophores including sfGFP, as we show. That in turn makes the protein purification pipeline work robustly - we do not think that this would be possible with any other FP.

We have compared eGFP, sfGFP, and moxGFP (cysteineless sfGFP) to hfYFP in a number of experiments and show that hfYFP can tolerate detrimental mutations that render even sfGFP almost entirely nonfluorescent. Notably, the destabilized hyperfolder mutant hfYFP-W57F, even with an 18 amino acid genetic code, still outperforms wild-type sfGFP in a variety of stability assays (Fig. S10). hfYFP-W57F even retains $T_m = 89.1^\circ C$, greater than wild-type mGreenLantern's or sfGFP's ($T_m = 87.2$ and $86.4^\circ C$, respectively), tolerating mutations that are detrimental and usually prohibitive to avFP fluorescence.

For these reasons, among others, hfYFP can potentially serve as a superior engineering template to sfGFP, or at least it can offer a very appealing option for biosensor construction that was unavailable previously. In this manuscript, we engineered LSSmGFP and LSSA12 from hfYFP in a structure-guided manner, having extensively detailed the mutations important to hfYFP stability—and these LSS-FPs are generally more stable than mT-Sapphire and mAmetrine, retaining many of the hfYFP advantages.

Lastly, we'd like to disclose that we have already produced a variety of biosensors from hfYFP, including redox-sensitive fluorescent proteins that outperform all others in their class in our extensive head-to-head comparisons. Applying the same core redox-sensing mutations into the eGFP or sfGFP backgrounds

produced nonfluorescent, misfolded product. It appears likely that hfYFP does indeed offer advantages in biosensor design.

For example, in the application of proExM, hfYFP/mhYFP has no greater advantage than mGL. Conversely, as the authors described, “mGreenLantern, with its 6.0-fold greater fluorescence than eGFP in mammalian cells compared to 2.4-fold for hfYFP, may be best suited for proExM.”

We do not claim that (m)hfYFP is better than mGL at proExM.

Nor does mhYFP outperform mEos4b in CLEM applications, although it survives CLEM sample preparation but its fluorescence retention only matches mEos4b's.

We also do not claim that its CLEM performance is better than mEos4b/mEosEM. The mere existence of a 2nd protein that can tolerate those fixation conditions is noteworthy, however. It is likely that (m)hfYFP will enable 2-color CLEM applications - but that is beyond the scope of this paper.

hfYFP performs better, but it is not a monomer.

hfYFP is not a “pure” monomer (69% normal), no, but it is much more monomeric than many common proteins treated essentially universally as monomers, including Emerald (59%), superfolder GFP (62%), Venus/Citrine (36%), mRuby2/3 (14%), mTagBFP2 (49%), mKate2 (59%), TagRFP (30%), mCardinal (41%), and YPet (62%). mhYFP (79%) is additionally more monomeric than the classic proteins EGFP (76%) and Clover (73%) (Cranfill et al., Nat. Met., 2016).

Since samples imaged with one method (eg, ExM) are difficult to image with another method (eg, CLEM) due to the different handling of samples in different applications or imaging techniques, it is obvious that, to obtain the best imaging results, users will choose mGL for ExM imaging, mEosEM/mEos4b for CLEM imaging, and not mhYFP. Most of the time, what we need is a single event champion, not an all-around champion.

We wholeheartedly agree. We did not intend to imply that (m)hfYFP is an all-around champion - please point us to specific language that the reviewer feels is misleading. Rather, we present (m)hfYFP as a remarkably stable FP - almost certainly the most stable FP ever made - and shows several best-in-class applications (e.g., protein purification) and several other quite-good-but-not-quite-best-in-class applications. We can foresee many situations in which users will opt for a protein with excellent thermostability (e.g. biosensor engineering, grafting of antibody CDRs into the loops to create “fluorobodies”, hyperthermophile research, fusion for crystallization, etc.). The EM fixation resistance, coupled with the lower pKa, means that (m)hfYFP could work in CLEM in acidic compartments better than mEos4/mEosEM. It could also enable 2-color CLEM alongside one of those proteins. Although we

didn't characterize it here, LSSmGFP might also be CLEM-compatible - as it is almost as stable as hfYFP in our hands - which would dramatically improve CLEM options.

We also believe that, in addition to the utility of (m)hfYFP as tools in their own right, that the engineering process, with much structure determination and site-directed mutagenesis, offers a strong roadmap for researchers wishing to engineer greater thermodynamic stability into proteins. We have tried to make the manuscript as useful as possible along this axis, as well.

Last, in biological applications, the authors did not provide biological new insights or discoveries using the developed FPs.

We would first point out that very few original publications of tools reveal profound new biological insights. That is simply not the purpose of such papers, and indeed would serve to obscure the engineering process. That said, we believe that the new reagents pave the way for such discoveries, by us and others, such as:

1. Superfolder GFP has been extensively studied for its stability and folding properties as well as its use as a platform for biosensor development. We offer three crystal structures that can serve these functions just as sfGFP has, and we demonstrate that hfYFP surpasses sfGFP in stability assays that have been positively correlated with biosensor performance. hfYFP goes well beyond sfGFP, tolerating mutations that render sfGFP nonfluorescent, and we extensively characterized a large library of mutants that will be valuable to developers. Besides, the study offers multiple fully characterized FPs that can function as stand-alone probes. It is reasonable to assume that hfYFP will fill many of the same roles as sfGFP, and probably more due to its unique properties that offer new opportunities—including its ability to withstand 7 M GdnHCl at room temperature for many weeks or months, something that sfGFP could tolerate for only a minute. In preliminary work external to this manuscript, we have shown that only using the (m)hfYFP scaffold were we able to make best-in-class redox biosensors for cell use - none of the other scaffolds were up to the task. We believe that this may extend to other sensor classes.
2. The potential of hfYFP to serve as a solubility tag is very intriguing, and insights like this provide avenues for further research to expand on these findings. We expect that our high-resolution crystal structures will lead to biophysical insights about the nature of chemical resistance of FPs—and perhaps proteins in general—when structural and computational biologists perform molecular dynamics (and other) simulations and other probes on them.
3. Using those structures, we engineered a highly stable LSS-GFP and modified its 488 nm absorbance using structure-guided mutagenesis. We found that just one novel mutation that we identified was sufficient to correct the 488 nm absorbance problem of mT-Sapphire, and this provides structure-

function insight that may apply to FPs of other lineages. Thus, the engineering pipeline and specific mutations identified here may help to improve even distantly related FPs and sensors.

Main concern :

1. The manuscript mainly focused on hfYFP. However, because hfYFP has weak dimerization property, it is not optimal as a fusion tag to label target proteins, especially when the target protein is a membrane protein, has weak dimerization property, or has a high expression level, labeling with hfYFP will affect the localization of the target protein. This is even more important for imaging techniques such as ExM and CLEM that require more precise and accurate localization of target protein. Obviously, the author also realized this problem and developed the monomeric mhYFP. The photochemical properties of mhYFP are the same as those of hfYFP, but mhYFP is a real monomer. Therefore, this manuscript should use mhYFP, rather than hfYFP, as the final tagging FP and compare the properties of mhYFP with other FPs.

We appreciate these points about the value of monomericity, and that is indeed why we produced mhYFP. Although our study dedicates significant time and attention to mhYFP, characterizing it alongside hfYFP in most experiments, we could further emphasize the importance of preferring mhYFP for proExM and CLEM in those sections. We would also point out that monomeric fluorescent proteins are not always preferred. For filling of cells or organelles, less monomeric proteins are often preferred - we would use hfYFP in these applications.

Furthermore, we do in fact compare mhYFP in almost all applications. Specifically:

- Fig. 1 compares cellular brightness.
- Fig. 2 focuses on hfYFP because mhYFP was developed much later. That said, we did measure thermodynamic stability of mhYFP as T_m , which was almost identical to hfYFP.
- Fig. 3 compares hfYFP to mhYFP in every panel.
- Fig. 4 does in fact use mhYFP and not hfYFP for CLEM, so its performance is demonstrated relative to eGFP and mGL there. We can make this clearer in the text.
- Fig. 5 uses hfYFP, but mhYFP should allow such engineering as well. We can make this point explicit in the text.

In addition, oligomeric properties tend to increase the extension coefficient and Mol. bright of FPs. The comparison of these properties in the text table should include mhYFP.

We thank the reviewer for the excellent suggestion. We will include mhYFP in Table 1.

In addition, mhYFP mutants are derived from mGL, which must be added as a control when comparing the properties and performances of different FPs.

mGL is used as a control in many experiments here, and we have characterized its biochemical properties in great detail in the mGL study alongside the same controls including eGFP, eYFP, mNeonGreen, mClover3, etc., using the same or similar techniques and even reagents. We can make this point clearer in the text.

2. In order to highlight certain chemical properties or application advantages of the developed FPs, the authors biased selection of some FPs for comparison, but did not select the best FPs with specific properties that have been reported. For example, mGL is the previously developed FP that survived GdnHCl treatment and proExM (PNAS, 2020, Benjamin C. Campbell, et al), in which mGL has already been shown to have obvious advantages over eGFP and is currently the best FP for proExM. hfYFP and mhYFP are from mGL, however when the authors compared certain biochemical properties of different FPs, some FPs were selected for comparison with a biased selectivity. In detail, in Figure 1b-1g the FPs selected for comparison were different, and mGL was not selected for comparison.

We did not purposely select FPs for comparison with “bias.” Rather, we tried to select a range of commonly used FPs throughout. Obviously, we could not compare to all FPs for the laborious EM, proExM, and protein-purification experiments - we tried to pick the most reasonable comparisons in those cases. eYFP, eGFP, and superfolder GFP are among the most commonly used FPs in biology - we felt that comparisons to these were perhaps the most useful that we could make. Although we appreciate that mGL is a useful benchmark, it is still a new protein, and biologists are much more familiar with eGFP and eYFP. We use the Fig. 1 panels to direct attention to very specific and clear points about hfYFP, not to compare it against numerous controls - which we have already measured in the mGL study, often in the same assays using the same methods. Nonetheless, this study greatly expands our knowledge about mGL through numerous experiments in which it is used as a standard. We will make more clear in text the numerous comparisons that have already been done in the mGL paper.

Key points of the mentioned panels:

1b) eYFP absorbs 405 nm light, whereas hfYFP and mhYFP do not. We have already shown the spectrum of mGL in the mGL paper and we wished to compare Y203-type avFPs here.

1c) hfYFP matures faster than eYFP. mGL's maturation is measured in the mGL paper using the same methods. We wanted to show that the widely used eYFP has a maturation problem that hfYFP does not.

1d) mGL is included here.

1e) mGL cellular brightness is shown in the mGL paper against the same controls using the same method.

1f-g) hfYFP has a lower pKa and is less chloride sensitive than eYFP. We know that mGL has a very low pKa, and chloride sensitivity is not a concern in avGFPs or suspected for mGL.

mGL is only compared in Table S2 for biochemical properties and brightness.

mGL was fully characterized in the mGL paper. In the hfYFP paper, we interrogate its chemical resistance properties in as much detail as hfYFP's. We will do a better job of referencing data published in the mGL paper.

The mF4Y-SR mutant suddenly appeared only in Fig.3e and not in other Fig. 3 figures.

We were curious to see how mF4Y-SR would perform, and since we had the data, decided to include it. If it's distracting, though, mF4Y-SR could be removed from the figure.

In Fig. 4d-g, mEos4b was not selected for comparison as a control to demonstrate the CLEM imaging performance of mhYFP in cells.

The performance of mEos4b is known and has been reported. We aimed to compare constitutively fluorescent FPs here - that is, an in-class comparison. We also showed that (m)hfYFP resist OsO₄, the primary quenching reagent, at least as well as mEos4b. We do not mean for (m)hfYFP to supplant mEos4b - indeed, it is possible that they would complement it in 2-color applications.

For another example, based on the anti-GdnHCl properties of hfYFP and LSSmGFP, the authors propose that they can be used to achieve fluorescence-assisted benchtop purification of inclusion body proteins under fully denaturing conditions. As a template, mGL has anti-GdnHCl properties. When visualizing the different steps of protein purification using hfYFP and LSSmGFP (Fig. 5), it is necessary to compare with mGL and prove that the performances are better than mGL.

Although mGL exhibits strong resistance to GdnHCl, hfYFP simply did not denature under GdnHCl. That gave us the idea for the purification system.

Also, mAmetrine should be selected as a control in Fig. S15a. etc.

mAmetrine is shown in every panel of Fig. S15 except for S15a. The excitation spectrum of mAmetrine has been reported elsewhere, but we have the data and can include the trace in S15a if that would be helpful to readers.

Additionally, the authors' claim in the Discussion that "we produced a series of avFP mutants demonstrating superior stability over sfGFP in every assay that we performed." is incorrect. Many figures in the manuscript used eGFP instead of sfGFP as a control (Fig. 1c, Fig. 3a-f, Fig. 4d-g, Fig. 5f). It is more reasonable to use sfGFP as a control rather than eGFP when discussing the stability of FPs and their applications. Taking Fig.5f and Fig.S17 as examples, when used as solubility tags, the aggregation properties of FPs themselves will affect the solubility properties of POIs. eGFP is a weak dimer. mGL and mEGFP are monomers. It is not reasonable to compare hfYFP and LSSmGFP with eGFP rather than mEGFP/sfGFP.

This is a reasonable criticism of this statement. We will tone it down in the text. Thank you for this recommendation.

Minor concern

1. When I first saw "ultrastable" in the title I thought "photostable". I recommend using "chemical stable" instead of "ultrastable".

This word is in somewhat common usage in the scientific literature: a Google Scholar search for "ultrastable" yields ~40,000 hits. "Ultrastable" with "protein" returns ~8000 hits, all the way back to the 80s and 90s, and with the same meaning as we have: "extremely thermodynamically stable and resistant to denaturation." Thus, we do not think that the word is problematic in this context. But if the editor agrees that this usage is confusing, we would be happy to change it.

2. There is no representative cell image by proExM using different FPs in Fig.3.

The images that we collected for the proExM quantitation are enormous tile scans of whole culture wells, so in our opinion there is little to gain from those images. We could perform an expansion experiment and image an organelle illuminated by mhYFP, though.

3. mEos4b is not the best OsO₄-tolerant fluorescent protein. In Fig.4 the authors need to compare mhYFP with mEosEM (Nature Methods, 2020). The performances of these FPs in GMA and Epon-embedded samples also need to be compared.

mEosEM was published in 2020, after we performed the CLEM experiments in late 2018 (there were enormous COVID-induced delays in this project). At the time, mEos4b was the state of the art. Although the performance of our FPs in GMA and Epon would be very interesting to know, we haven't tested those resins and we make no claims about mGL or hfYFP performance in them. And again, we are not

claiming that (m)hfYFP is superior to any Eos protein for CLEM - they are different types of FP - one constitutive, one photoconvertible. They are not interchangeable - rather, they might be used alongside.

4. How do LSSmGFP and LSSA12 perform in ExM applications compared to mhYFP?

They are compatible with proExM, but we haven't quantified their retained fluorescence in the way that we did with the other FPs, since we engineered these quite a while after performing the quantitative ExM experiments.

5. The title of the manuscript is "Ultrastable FPs for advanced microscopy". Because super-resolution (SR) imaging is an important component of advanced microscopy methods, as the authors described in their introduction, the authors need to discuss the application of the developed FPs in SR and compare their advantages and disadvantages with existing proteins.

We have demonstrated that hfYFP survives OsO₄ and a standard CLEM preparation protocol. One application is that hfYFP can survive the process at all—very few FPs can. The FPs tolerated proExM with fluorescence retention values that we display, and we use eGFP, mClover3, mNG, eYFP, and mGL as controls (Fig. 3a). We can discuss these issues more in the text.

6. The quality of images by advanced microscopies depends on the retained fluorescence of FP. Since different FPs have different molecular and cellular brightness, the retained fluorescence relative to eGFP should be compared in Fig.3 and Fig.4 as in Fig.1e.

We show the molecular and cellular brightness values of each FP in different figures, but we can put together a table that unifies these values and try to come up with an appropriate mathematical formula to bring that all together. It would further emphasize the advantages of mGL and hfYFP in these experiments compared to eGFP, sfGFP, and eYFP, for example.

7. The photostability of mGL is not good. How about hfYFP?

We recently performed laser-scanning confocal photobleaching experiments, and our preliminary data suggest that hfYFP is less photostable than eYFP, similar to Venus and Citrine. We do not know of another avYFP that comes close to eYFP's photostability. We will put the photostability data in the revised manuscript - thanks for this reminder.

8. Fig.4: 1) Fig.4d and 4e: Inconsistent with the description of Line 231: which one is before? Which one is after? 2) hfYFP is used in d and e, but mhYFP is used in statistical graphs f and g; mhYFP is marked in h, but hfYFP is described in the text (Line 234-235). 3) The fluorescence signals of eGFP and

mGL in Fig. 4h are very strong, and the significant advantage of mhYF over eGFP and mGL cannot be seen from the figure.

Thank you for catching this, and we apologize for the confusion. These are typos and we will correct them. It was mhYFP that was used throughout Fig. 4d-h, not hfYFP.

9. Sometimes eGFP is used in the manuscript, and sometimes EGFP is used, which needs to be unified.

Thank you; we will do this.

10. mAmetrine should be compared in Fig.5 and Fig.S15. mAmetrine and mGL are a good pair, which shows no bleed-through between the 405 nm and 470 nm channels. And there are many FPs reported for dual-color imaging. The advantages of the mGL/LSSmGFP pair is not significant when it is only used for the general dual-color imaging techniques. The authors should demonstrate the advantages of the mhYFP/LSSmGFP pair in applications based on their chemical stability, for example in ExM imaging.

mAmetrine has a clean excitation spectrum with no peak in the 488 nm window, so we fully expect that it would pair well with mGL. Something we aimed to demonstrate through LSSmGFP is the value of our crystal structures in guiding the engineering of new FPs. This led to LSSmGFP, with good chemical stability and a clean excitation spectrum due to a novel mutation uncovered in our screening (which was also effective at improving mT-Sapphire). It's true that there are various FPs for dual-color imaging approaches, most notably mT-Sapphire and mAmetrine for the green-yellow range. We simply wished to demonstrate the practical consequences of these excitation traces in cells, since the impact on imaging may not be obvious to a broad audience when interpreting only the excitation scans. Fig. 5e shows using photographs that cross-excitation from the eGFP/mT-Sapphire pair occurs under routine imaging conditions, consistent with the spectral scans.

11. There are two bs in Table S3.

Thank you!

12. It appears that the localization of mhYFP in Fig. 3f is different from those of the other proteins. Does mhYFP labeling affect H2B localization? More localization images of mhYFP need to be shown as in Fig.1h-1m.

We routinely use hfYFP for nuclear targeting with H2B and have not found it to affect H2B localization. The photographed cells are probably dividing, or we are seeing an overexpression artifact in these cells. We would be happy to select more representative images of H2B localization and include them.

13. The proportion of normal cells for Clover measured in Fig. S6 is 63%, while that in Fig. S15f is 75%. How to explain such a big difference? The proportion of normal cells for Clover is 75% (Fig. S15f), which is defined as weak dimer. However, in Fig. S6 hfYFP-V206K (79%) and mhYFP (80%) were defined as monomers. What is the basis for defining the aggregation state of a FP as weak dimer or monomer?

The OSER assay is highly—some would say notoriously—variable. The scores depend greatly on transfection conditions, experimental timing, focusing and brightness of cells, scoring criteria, and the passage number of cells. We make our absolute best effort to reproduce the protocol described in our methods section and in Costantini et al., *Traffic*, 2012, and Cranfill et al., *Nat. Met.*, 2016, but there is an expected degree of variability inherent in this morphological assay, which is scored by eye.

Source	mEGFP	mGL	moxGFP	Clover	mTSapphire	tdTomato
Cranfill et al., Nat. Met. , 2016	98.1%	N.D.	N.D.	72.9%	95.5%	57.6%
Campbell et al., PNAS , 2020	86%	86%	N.D.	71%	N.D.	29%
Fig. S6c	N.D.	85%	86%	63%	N.D.	49%
Fig. S15f	90%	N.D.	N.D.	75%	82%	37%

As shown above, the values we obtained in our study for the monomeric proteins were all at least 80%. Consequently, we continue to use the >80% healthy morphology definition to define monomericity as we used it in the mGL paper, based on the performance of the controls in our hands. We consider our experimental data and interpretations to be in alignment with literature reports and within a very reasonable error range.

We noticed that we have interpreted hfYFP-V206K (79%) as monomeric. If we are to apply a strict 80% cutoff for monomericity, then we should instead refer to hfYFP-V206K as a “weak dimer” and will make that correction.

Our scoring criteria do not appear to be mentioned in the manuscript, so we will be sure to include them.

14. In Fig.S2, there is a typo that S147P should be S147R.

Thank you!

Reviewer #3:

Remarks to the Author:

This manuscript describes the engineering of robustly folding and super stable fluorescent proteins that can withstand harsh chemical conditions and high temperatures. Such proteins would have broad

applications in expansion microscopy, correlated light and electron microscopy, and as visualization tags during protein purification. The manuscript is a tour-de-force and presents a comprehensive analysis of structural features that contribute to rapid folding and high stability. The supplementary information is a treasure trove of valuable insights that would be extremely useful for the broader protein engineering community. The tools themselves (hfYFP, mhYFP, LSSmGFP, LSSA12) appear to present notable strengths over currently available FPs for advanced applications. The experiments are well described. The methods and analysis are thorough and rigorous. The conclusions are well-supported by the data presented. The referencing is appropriate. The manuscript is clear and easy to follow. I think this is a strong contribution and I support publication. There are a few minor issues (noted below) that would improve clarity.

- 1) I don't understand why there is an increase in fluorescence for mGreenLantern and Hyperfolder YFP (Fig 2a and 2c). Can the authors speculate on/explain the origin of this increase?
- 2) Figure 3: (c) and (f) would benefit from inclusion of a scale bar. The quality of the images is poor in my pdf (even when zoomed in on a computer screen) – but the morphology of mhYFP in (f) appears drastically different (it looks punctate?) compared to the others. Is there a localization/morphology difference? This seems important to evaluate for histology purposes. Lastly, given the variability in signal in (a), (d), and (e), these would benefit from statistical analysis.
- 3) Figure 4: I think scale bars are present in (d) and (e) but they are so tiny I can't read them. Please add visible, readable scale bars.
- 4) The manuscript would benefit from some sort of qualified statement about the potential risks of super-stable proteins. There is a lot of accumulated knowledge in the FP community about what FP to use and what FP to avoid for a particular application. These nuances are not always passed on to the broader biological community. For example, the community generally accepts that over long periods of time mCherry can be toxic and can accumulate in lysosomes. The protein appears not to be efficiently broken down in the lysosome and because of mCherry's low pKa, it retains fluorescence. I have heard discussions at meetings and workshops that perhaps mCherry isn't efficiently degraded because it is too stable. I imagine the same problem could occur for fusions to super stable proteins. This may not be an issue if users are fusing hfYFP to a long lived structural protein to be visualized in CLEM or proExM, but it could be a significant issue on a short-lived signaling protein. It seems wise to include a qualified statement that users should check whether fusion alters the protein stability/half-life.
 r long periods of time mCherry can be toxic and can accumulate in lysosomes. The protein appears not to be efficiently broken down in the lysosome and because of mCherry's low pKa, it retains fluorescence. I have heard discussions at meetings and workshops that perhaps mCherry isn't efficiently degraded because it is too stable. I imagine the same problem could occur for fusions to super stable proteins. This may not be an issue if users are fusing hfYFP to a long lived structural protein to be visualized in CLEM or proExM, but it could be a significant issue on a short-lived signaling

protein. It seems wise to include a qualified statement that users should check whether fusion alters the protein stability/half-life.

Decision Letter, first revision:

Dear Ben,

Your Article, "Chemically stable fluorescent proteins for advanced microscopy", has now been seen again by three reviewers. As you will see from their comments below, two reviewers now sign off on your manuscript, while one still has concerns.

With regards to the remaining concerns of referee 2, we share only one, which is that we would like to see whether mGL could be used as-is in the protein purification application. You may already have these data (we hope you do). We do not think the outcome of this experiment will influence our decision on your paper, but we do think it should be very clear to readers what practical advantages the hyperfolder variants have over their precursor in this case.

We therefore invite you to revise your manuscript to address this final concern.

- * in this case it is not necessary to provide a point-by-point rebuttal. You can simply summarize the updates in your cover letter.
- * please underline/highlight any additions to the text or areas with other significant changes to facilitate review of the revised manuscript
- * address the points listed described below to conform to our open science requirements
- * ensure it complies with our general format requirements as set out in our guide to authors at www.nature.com/naturemethods
- * resubmit all the necessary files electronically by using the link below to access your home page

[Redacted] URL links to your confidential home page and associated information about manuscripts you may have submitted, or that you are reviewing for us. If you wish to forward this email to co-authors, please delete the link to your homepage.

We hope to receive your revised paper within four weeks. If you cannot send it within this time, please let us know. In this event, we will still be happy to reconsider your paper at a later date so long as nothing similar has been accepted for publication at Nature Methods or published elsewhere.

OPEN SCIENCE REQUIREMENTS

REPORTING SUMMARY AND EDITORIAL POLICY CHECKLISTS

IMAGE INTEGRITY

DATA AVAILABILITY

We strongly encourage you to deposit all new data associated with the paper in a persistent repository where they can be freely and enduringly accessed. We recommend submitting the data to discipline-specific and community-recognized repositories; a list of repositories is provided here:

<http://www.nature.com/sdata/policies/repositories>

All novel DNA and RNA sequencing data, protein sequences, genetic polymorphisms, linked genotype and phenotype data, gene expression data, macromolecular structures, and proteomics data must be deposited in a publicly accessible database, and accession codes and associated hyperlinks must be provided in the “Data Availability” section.

Please include a “Data availability” subsection in the Online Methods. This section should inform readers about the availability of the data used to support the conclusions of your study, including accession codes to public repositories, references to source data that may be published alongside the paper, unique identifiers such as URLs to data repository entries, or data set DOIs, and any other statement about data availability. At a minimum, you should include the following statement: “The data that support the findings of this study are available from the corresponding author upon request”, describing which data is available upon request and mentioning any restrictions on availability. If DOIs are provided, please include these in the Reference list (authors, title, publisher (repository name), identifier, year). For more guidance on how to write this section please see: <http://www.nature.com/authors/policies/data/data-availability-statements-data-citations.pdf>

MATERIALS AVAILABILITY

ORCID

Nature Methods is committed to improving transparency in authorship. As part of our efforts in this direction, we are now requesting that all authors identified as ‘corresponding author’ on published papers create and link their Open Researcher and Contributor Identifier (ORCID) with their account on the Manuscript Tracking System (MTS), prior to acceptance. This applies to primary research papers only. ORCID helps the scientific community achieve unambiguous attribution of all scholarly contributions. You can create and link your ORCID from the home page of the MTS by clicking on ‘Modify my Springer Nature account’. For more information please visit www.springernature.com/orcid.

Sincerely,
Rita

Rita Strack, Ph.D.
Senior Editor
Nature Methods

Reviewers' Comments:

Reviewer #1:

Remarks to the Author:

In the revised manuscript Benjamin C. Campbell et al. appropriately and satisfactorily addressed all of the numerous points raised by reviewers. The manuscript was substantially improved by the additional data concerning FPs brightness comparison after expansion microscopy and fixation of the cells and punctate morphology studies of the cells expressing H2B-FPs fusions.

Overall, I feel that the paper is now acceptable for publication in Nature Methods.

Reviewer #2:

Remarks to the Author:

The author answered most of the points I raised, but some important issues related to comparisons were not addressed.

Probes are developed to meet specific biological applications. In addition to the properties of probes, biologists pay more attention to the performance of probes in specific applications. I completely agree with the authors that (m)hfYFP has higher thermodynamic stability than mGL, and will serve as templates for biosensor engineering, given the crystal structures and structure-activity relationships. However, the quality of certain properties of a probe may do not determine how well it will perform when imaged in the final biological sample (point 1 below). Therefore, the authors must demonstrate the significant advances over mGL and existing probes in specific advanced microscopy applications.

The manuscript describes the thermal and chemical stability of (m)hfYFP, and three specific applications in advanced microscopies: proExM, CLEM, and fluorescence-assisted protein purification. I totally agree

with the in-class comparison for specific applications. However, I don't think choosing a range of commonly used FPs to compare with (m)hfYFP is a good in-class comparison for advanced microscopy, nor is it "the most reasonable comparisons" as the author responded. It is because commonly used FPs cannot be used for these advanced microscopies that researchers have developed advanced FPs for them. The title and focus of the manuscript are Chemically stable fluorescent proteins for advanced microscopy, thus, I think the comparison of (m)hfYFP with the probes used for advanced microscopies, but not the common FPs, is the real in-class comparison. The authors must highlight the significant advancements of (m)hfYFP in advanced microscopies by comparing (m)hfYFP with the best in-class probes that have been reported for the specific applications. Alternative probes/FPs for advanced microscopy may be quite-good, but probably not good enough to be published in the high-impact journal Nature Methods.

- 1) proExM: hfYFP has better thermal and chemical stability than mGL, but it performs worse than mGL when used in proExM. The authors need to discuss the properties of the probes that are required for use in proExM imaging, which is important for the development of proExM probes. Also, it would be interesting to know if mEos4b can be used in proExM.
- 2) CLEM: I agree that (m)hfYFP performs dramatically better than mGL in EM. But mEos4, mEosEM, CoGFPV0 (Sci Rep 10, 21871, 2020, <https://doi.org/10.1038/s41598-020-78879-x>), not mGL, are reported probes suitable for CLEM imaging. I'm very much in favor of the in-class comparison. The authors compared (m)hfYFP with mEos4b (Fig. 4b), indicating that they believe that (m)hfYFP and mEos4b are the same type of probes for CLEM imaging, but this contradicts the authors' response of "(m)hfYFP and mEos4b are different types of FP - one constitutive, one photoconvertible" (Minor concern, point 3). Both mEos4b and mEosEM are photoconvertible FPs. If the authors believe that mEos4, mEosEM, and CoGFPV0 are the same type of FP used for CLEM imaging, they cannot just cite mEos4b without citing mEosEM and CoGFPV0, nor can they exclude mEosEM and CoGFPV0 from the comparison (Fig. 4b). If they consider (m)hfYFP to be a constitutive FP for CLEM, they must compare the performance of (m)hfYFP with CoGFPV0, a constitutively fluorescent FP, in the final EM samples.
- 3) fluorescence-assisted protein purification: The use of chemically stable FP for fluorescence-assisted protein purification is a good innovation and application, but the comparison of hfYFP with mGL is also important because mGL performance directly affects the advancement of the manuscript. Finding that mGL can be used for fluorescence-assisted protein purification is not the same thing as engineering a FP based on mGL that can be better used for fluorescence-assisted protein purification. I cannot accept that (m)hfYFP was not compared to mGL in the specific advanced microscopies just because mGL (published in 2020, not new actually) "is still a new protein, and biologists are much more familiar with eGFP and eYFP". In addition, from the performance of hfYFP and mGL in proExM imaging, the final imaging quality will be affected by multiple sample preparation steps, which may be inconsistent with or not only determined by chemical stability. The possibility that mGL performs better than (m)hfYFP for

fluorescence-assisted protein purification cannot be ruled out. Why develop (m)hfYFP based on mGL for visualization of protein purification steps if mGL were already fine?

Reviewer #3:

Remarks to the Author:

The authors have addressed all my questions/concerns (which were minor in the first place). I am strongly supportive of publication. I think this paper has significant value on the "tool front" and the "engineering front".

Please see my prior comments regarding conclusions, referencing, etc (all of which were fine).

I thank the authors for their comprehensive answer to my question regarding the increase in fluorescence in Figure 2a and 2c. I have new appreciation for the complexity of these kinetic unfolding traces. The issue was a minor one as it doesn't change conclusions in the manuscript; it was mostly my curiosity. I don't recommend changing the presentation of the data in Figure 2a and 2c and it doesn't seem necessary to explain in further detail.

Author Rebuttal, first revision:

Reviewer #1

In this manuscript Benjamin C. Campbell et al. developed extremely chemically- and thermodynamically-stable yellow hfYFP and green LSSmGFP fluorescent proteins, solved their crystal structures and successfully applied them for the proExM and CLEM microscopy techniques. First, authors developed chemically-stable yellow hfYFP protein using rational mutagenesis followed by screening for high GdnHCl stability and thermostability. Benjamin C. Campbell et al. further characterized spectral properties of the hfYFP protein in vitro and its brightness and behavior in fusions when expressed in mammalian cells. Authors next demonstrated superior chemical-stability of the hfYFP and its monomeric variant mhYFP in various denaturants (such as GdnHCl, GdnSCN, temperature, and H₂O₂) as compared to other FPs.

Benjamin C. Campbell et al. next showed that hfYFP can successfully withstand conditions used in expansion microscopy (proExM), histology, and electron microscopy techniques. Authors also solved crystal structures of hfYFP, mhYFP and FOLD6 proteins and suggested structural basis of their high stability. Using these structures they developed green fluorescent proteins with a large Stokes shift, LSSmGFP and LSSA12, which have superior thermal and

chemical stability and ensure cross-blead-free two-color live cells imaging as compared to available mT-Sapphire and EGFP proteins.

Finally, Benjamin C. Campbell et al. applied the hfYFP and LSSmGFP proteins as fusion tags to enable fluorescence-assisted protein purification under denaturing conditions. Overall, the way of the protein engineering and a set of the stable green and yellow fluorescent proteins developed in this manuscript are interesting for broad readers from different fields. Besides expansion and electron microscopy, hfYFP can be potentially applicable to bimolecular fluorescence complementation, super-resolution imaging techniques (STED nanoscopy) and other advanced microscopy methods and biological applications demanding fluorescent proteins that can preserve its fluorescence under rigid sample preparation conditions, such as tissue clearing. hfYFP and LSSmEGFP proteins can also be used for the development of the stable and expression-enhanced indicators.

Minor points:

1. Main text, line 28. Misprint "...in advances microscopy methods..." should be replaced with "...in advanced microscopy methods..."

Fixed.

2. Main text, lines 133 and 389. In the main text and Table S3 authors compared stabilities of sfGFP and hfYFP in 7M GdnHCl but in Fig2a legend the same comparison is shown for 6.3 M GdnHCl. Please, correct "7M" for "6.3 M".

The same wrong 7M concentration is mentioned below in line 389.

Line 133 corrected: the 3-month protein was in 7 M GdnHCl, whereas the 48 hr denaturing experiment was in 6.3 M GdnHCl. This is now consistent with the figure legend.

3. Main text, line 182. mhYFP is mentioned in line 182 but on Figure 3a hfYFP is shown. Please, correct "mhYFP" for "hfYFP".

Line 182 corrected to hfYFP.

4. Main text, lines 232-233 and Figure 4f and 4g. hfYFP fluorescence is discussed in the text but Fig.4f and 4g are related with mhYFP. Please, correct.

We're sorry for the inconsistency of labeling here: both hfYFP and mhYFP were used in Fig. 4a-b, as shown, but only mhYFP was tested in Fig. 4c-h. Paragraph of lines 227-236 has been corrected to mhYFP. Fig. 4d panel label and figure legend have been corrected to mhYFP.

5. Main text, line 261. Please, correct misprint "... the possibility surface..." to "...the possibility of surface..."

Fixed.

6. Main text, line 267 and Figure S11a. For better understanding of impact of T65S mutation, T65 residue might be included on Figure S11a.

The location of G65 in ~~Fig. S11a-b~~ (Fig. S13a-b) is now indicated.

7. Figure S11b and S11c. Please, define crosses on the Fig S11b,c.

We've defined the crosses as water molecules that are not directly involved in the proton wire.

8. Main text, line 306 and Figure S15. I couldn't find Fig. S15l panel. Please, correct. Also, I suggest that Fig S15k panel corresponds to mitochondria localization and Fig S15j – to endoplasmic reticulum. Please, correct by swapping panel j and k on Figure S15.

These issues have been corrected in main text and Fig. S17.

9. Main text, line 322. Authors cite Fig. S15b to refer to the high HdnHCl stability on the LSSmGFP protein. However, Fig. 15b describes thermal stability of the LSSmGFP protein. Reference to Fig.S15c would be appropriate in case authors correct X-axis label on Fig. S15c from "H2O2 (%/w/v)" to "Time".

The text points to the correct panels, but the figure legends were out of order for ~~Fig. S15b-d~~ (Fig. S17). We've corrected them:

Fig. S17b. GdnHCl stability.

Fig. S17c. H₂O₂ stability.

Fig. S17d. Thermostability.

10. Error-prone libraries, line 923. Could authors explain the meaning of the following phrase: "the reaction was PCR purified"?

Changed to, "and the DNA was isolated using spin columns (Thermo Scientific #K0702) to remove reaction components."

11. Line 1558, Figure S3d. Green line corresponding to "Clover-cc" is not seen in the Figure S3d.

Also, on the Figure S3d I could not ascribe grey line (3rd from the bottom) to any of FPs shown on Figure S3d.

The grey line is Clover-cc. We've changed the color to green so it now matches the figure legend.

12. For convenience of the readers, I would recommend to provide supplementary Figure containing alignment of the amino acids sequences for the major developed proteins, i.e. at least for hfYFP, mhYFP, LSSmGFP and LSSA12.

Great suggestion. We'll add an alignment figure (**Fig. S18**) and highlight sequence differences between the proteins.

Overall, the publication in Nature Methods is appropriate and highly recommended.

Thank you very much for the careful reading and suggestions.

Reviewer #2

Remarks to the Author:

Based on the previously developed mGreenLantern (mGL), the authors developed the well-folded cysteine-free FP hyperfolder YFP (hfYFP) and monomeric mhYFP, and investigated the applications and performances of these FPs in advanced microscopy methods such as proExM and CLEM. Next, they solved the crystal structures of hfYFP, mhYFP and a cysteine-free mGL variant called FOLD6, and engineered two FPs with large Stokes shift (LSSA12 and LSSmGFP) based on these structures. Finally, the authors successfully visualized the different steps of protein purification using the chemical stability of hfYFP and LSSmGFP.

Compared with the existing common FPs (eGFP, eYFP, mClover3, and mNeonGreen, etc.), the chemical stabilities of hfYFP and mhYFP have been improved greatly, but these properties have no significant advantages compared with those of mGL, and some properties are even worse.

It is very rare that any one reagent is the best-in-class along all dimensions. Rather, reagents will be best along one or several dimensions, with the specific experiments guiding reagent choice. This is indeed the case here as well. We do not believe, and did not mean to suggest, that hfYFP and mhYFP are superior to all other FPs along all axes. Rather, hfYFP and mhYFP are best-in-class for some specific applications, which enable new types of experiments. If the reviewer has in mind any specific language from the manuscript that seems overstated, please refer us directly to that and we will amend.

Specifically, we believe that (m)hfYFP:

- Has incredible thermodynamic stability, significantly higher than mGL.
- Performs dramatically better than mGL in EM (we do not claim superiority to mEos4, mEosEM, etc.). mGL was effectively non-functional in our EM conditions.
- Allows simple and robust protein purification. We did not specifically test mGL in this assay, but its cysteines (of which (m)hfYFP has none) will likely complicate the folding and trafficking of fusions.
- Allows robust refolding, which again mGL may struggle with due to cysteine content.
- Has substantially lower pKa (~5.6) than most commonly used FPs (~6-6.5), providing more acid-insensitivity in uses around physiological pH.
- Will serve as templates for biosensor engineering, given the crystal structures and structure-activity relationships.

We also note that we have compared hfYFP to another monomeric cysteine-less avFP, moxGFP, which has been advertised for its performance in the secretory pathway. As summarized in manuscript lines 112-116, hfYFP has twice the molecular brightness, three times the brightness in cells, lower pKa, improved thermodynamic stability, twice the speed of chromophore maturation, slower unfolding, and better refolding, and it also contains no cysteines. Therefore, we expect hfYFP to outperform the most popular cysteine-less “mox” FP in similar use cases while offering many additional advantages.

For example, compared with mGL (Table S2), the molecular brightnesses of hfYFP and mhYFP are about the same as that of mGL, but the cellular brightness is only 1/3 of mGL, and T_m is only slightly increased (92.8 vs 87.2).

That is only one of the many measures of thermodynamic stability that we measured. In many assays, the thermodynamic stability of (m)hfYFP was markedly higher than mGL:

- In Fig. 2a, (m)hfYFP never unfolded in 6.3 M GdnHCl (up to 3 months), whereas mGL unfolded fairly quickly (although more slowly than EGFP and mF4Y). See also Fig. S9 (Fig. S11).
- In Fig. 2b, (m)hfYFP unfolded much more slowly than mGL in 3.6 M GdnSCN.
- In Fig. 2e, isothermal unfolding showed that mGL unfolds much more quickly than (m)hfYFP at 87°C. See also Fig. S7 (Fig. S9); holds for $T \geq 70^\circ\text{C}$.
- In Fig. 2h, (m)hfYFP was much more stable than mGL in 1-20% H₂O₂.
- In Fig. S5 (Fig. S5), (m)hfYFP provided a higher fraction of soluble protein than mGL.
- In Fig. S8 (Fig. S10), (m)hfYFP was more stable than mGL in 2 M NaOH.
- See also Table S3 for quantification of previous measurements.

In specific advanced imaging applications, mhYFP does not possess any single property or performance that cannot be surpassed by the best FPs reported.

We respectfully disagree with the reviewer. The thermodynamic stability of (m)hfYFP is far and away the largest of any FP in the literature. This allowed the targeted removal of cysteine residues, which has led to dramatic destabilization of previous GFP-based fluorophores including sfGFP, as we show. That in turn makes the protein purification pipeline work robustly - we do not think that this would be possible with any other FP.

We have compared eGFP, sfGFP, and moxGFP (cysteineless sfGFP) to hfYFP in a number of experiments and show that hfYFP can tolerate detrimental mutations that render even sfGFP almost entirely nonfluorescent. Notably, the destabilized hyperfolder mutant hfYFP-W57F, even with an 18 amino acid genetic code, still outperforms *wild-type* sfGFP in a variety of stability assays (Fig. S40) (Fig. S6). hfYFP-W57F even retains $T_m = 89.1^\circ\text{C}$, greater than wild-type mGreenLantern's or sfGFP's ($T_m = 87.2$ and 86.4°C , respectively), tolerating mutations that are detrimental and usually prohibitive to avFP fluorescence.

For these reasons, among others, hfYFP can potentially serve as a superior engineering template to sfGFP, or at least it can offer a very appealing option for biosensor construction that was unavailable previously. In this manuscript, we engineered LSSmGFP and LSSA12 from hfYFP in a structure-guided manner, having extensively detailed the mutations important to hfYFP stability—and these LSS-FPs are generally more stable than mT-Sapphire and mAmetrine, retaining many of the hfYFP advantages.

Lastly, we'd like to disclose that we have already produced a variety of biosensors from hfYFP, including redox-sensitive fluorescent proteins that outperform all others in their class in our extensive head-to-head comparisons. Applying the same core redox-sensing mutations into the eGFP or sfGFP backgrounds produced nonfluorescent, misfolded product. It appears likely that hfYFP does indeed offer advantages in biosensor design.

For example, in the application of proExM, hfYFP/mhYFP has no greater advantage than mGL. Conversely, as the authors described, “mGreenLantern, with its 6.0-fold greater fluorescence than eGFP in mammalian cells compared to 2.4-fold for hfYFP, may be best suited for proExM.”

We do not claim that (m)hfYFP is better than mGL at proExM.

Nor does mhYFP outperform mEos4b in CLEM applications, although it survives CLEM sample preparation but its fluorescence retention only matches mEos4b's.

We also do not claim that its CLEM performance is better than mEos4b/mEosEM. The mere existence of a 2nd protein that can tolerate those fixation conditions is noteworthy, however. It is likely that (m)hfYFP will enable 2-color CLEM applications - but that is beyond the scope of this paper.

hfYFP performs better, but it is not a monomer.

hfYFP is not a “pure” monomer (69% normal), no, but it is much more monomeric than many common proteins treated essentially universally as monomers, including Emerald (59%), superfolder GFP (62%), Venus/Citrine (36%), mRuby2/3 (14%), mTagBFP2 (49%), mKate2 (59%), TagRFP (30%), mCardinal (41%), and YPet (62%). mhYFP (79%) is additionally more monomeric than the classic proteins EGFP (76%) and Clover (73%) (Cranfill et al., *Nat. Met.*, 2016).

Since samples imaged with one method (eg, ExM) are difficult to image with another method (eg, CLEM) due to the different handling of samples in different applications or imaging techniques, it is obvious that, to obtain the best imaging results, users will choose mGL for ExM imaging, mEosEM/mEos4b for CLEM imaging, and not mhYFP. Most of the time, what we need is a single event champion, not an all-around champion.

We wholeheartedly agree. We did not intend to imply that (m)hfYFP is an all-around champion - please point us to specific language that the reviewer feels is misleading. Rather, we present (m)hfYFP as a remarkably stable FP - almost certainly the most stable FP ever made - and

shows several best-in-class applications (e.g., protein purification) and several other quite-good-but-not-quite-best-in-class applications. We can foresee many situations in which users will opt for a protein with excellent thermostability (e.g. biosensor engineering, grafting of antibody CDRs into the loops to create “fluorobodies”, hyperthermophile research, fusion for crystallization, etc.). The EM fixation resistance, coupled with the lower pKa, means that (m)hfYFP could work in CLEM in acidic compartments better than mEos4/mEosEM. It could also enable 2-color CLEM alongside one of those proteins. Although we didn’t characterize it here, LSSmGFP might also be CLEM-compatible - as it is almost as stable as hfYFP in our hands - which would dramatically improve CLEM options.

We also believe that, in addition to the utility of (m)hfYFP as tools in their own right, that the engineering process, with much structure determination and site-directed mutagenesis, offers a strong roadmap for researchers wishing to engineer greater thermodynamic stability into proteins. We have tried to make the manuscript as useful as possible along this axis, as well.

Last, in biological applications, the authors did not provide biological new insights or discoveries using the developed FPs.

We would first point out that very few original publications of tools reveal profound new biological insights. That is simply not the purpose of such papers, and indeed would serve to obscure the engineering process. That said, we believe that the new reagents pave the way for such discoveries, by us and others, such as:

1. Superfolder GFP has been extensively studied for its stability and folding properties as well as its use as a platform for biosensor development. We offer three crystal structures that can serve these functions just as sfGFP has, and we demonstrate that hfYFP surpasses sfGFP in stability assays that have been positively correlated with biosensor performance. hfYFP goes well beyond sfGFP, tolerating mutations that render sfGFP nonfluorescent, and we extensively characterized a large library of mutants that will be valuable to developers. Besides, the study offers multiple fully characterized FPs that can function as stand-alone probes. It is reasonable to assume that hfYFP will fill many of the same roles as sfGFP, and probably more due to its unique properties that offer new opportunities—including its ability to withstand 7 M GdnHCl at room temperature for many weeks or months, something that sfGFP could tolerate for only a minute. In preliminary work external to this manuscript, we have shown that only using the (m)hfYFP scaffold were we able to make best-in-class redox biosensors for cell use - none of the other scaffolds were up to the task. We believe that this may extend to other sensor classes.
2. The potential of hfYFP to serve as a solubility tag is very intriguing, and insights like this provide avenues for further research to expand on these findings. We expect that our high-resolution crystal structures will lead to biophysical insights about the nature of chemical resistance of FPs—and perhaps proteins in general—when structural and computational biologists perform molecular dynamics (and other) simulations and other probes on them.
3. Using those structures, we engineered a highly stable LSS-GFP and modified its 488 nm absorbance using structure-guided mutagenesis. We found that just one novel mutation that we

identified was sufficient to correct the 488 nm absorbance problem of mT-Sapphire, and this provides structure-function insight that may apply to FPs of other lineages. Thus, the engineering pipeline and specific mutations identified here may help to improve even distantly related FPs and sensors.

Main concern :

1. The manuscript mainly focused on hfYFP. However, because hfYFP has weak dimerization property, it is not optimal as a fusion tag to label target proteins, especially when the target protein is a membrane protein, has weak dimerization property, or has a high expression level, labeling with hfYFP will affect the localization of the target protein. This is even more important for imaging techniques such as ExM and CLEM that require more precise and accurate localization of target protein. Obviously, the author also realized this problem and developed the monomeric mhYFP. The photochemical properties of mhYFP are the same as those of hfYFP, but mhYFP is a real monomer. Therefore, this manuscript should use mhYFP, rather than hfYFP, as the final tagging FP and compare the properties of mhYFP with other FPs.

We appreciate these points about the value of monomericity, and that is indeed why we produced mhYFP. Although our study dedicates significant time and attention to mhYFP, characterizing it alongside hfYFP in most experiments, we could further emphasize the importance of preferring mhYFP for proExM and CLEM in those sections. We would also point out that monomeric fluorescent proteins are not always preferred. For filling of cells or organelles, less monomeric proteins are often preferred - we would use hfYFP in these applications.

Furthermore, we do in fact compare mhYFP in almost all applications. Specifically:

- Fig. 1 compares cellular brightness.
- Fig. 2 focuses on hfYFP because mhYFP was developed much later. That said, we did measure thermodynamic stability of mhYFP as T_m , which was almost identical to hfYFP.
- Fig. 4 does in fact use mhYFP and not hfYFP for CLEM, so its performance is demonstrated relative to eGFP and mGL there. We can make this clearer in the text.
- Fig. 5 uses hfYFP, but mhYFP should allow such engineering as well. We can make this point explicit in the text.

In addition, oligomeric properties tend to increase the extension coefficient and Mol. bright of FPs. The comparison of these properties in the text table should include mhYFP.

We thank the reviewer for the excellent suggestion. We will include mhYFP in Table 1.

In addition, mhYFP mutants are derived from mGL, which must be added as a control when comparing the properties and performances of different FPs.

mGL is used as a control in many experiments here, and we have characterized its biochemical properties in great detail in the mGL study alongside the same controls including eGFP, eYFP,

mNeonGreen, mClover3, etc., using the same or similar techniques and even reagents. We can make this point clearer in the text.

2. In order to highlight certain chemical properties or application advantages of the developed FPs, the authors biased selection of some FPs for comparison, but did not select the best FPs with specific properties that have been reported. For example, mGL is the previously developed FP that survived GdnHCl treatment and proExM (PNAS, 2020, Benjamin C. Campbell, et al), in which mGL has already been shown to have obvious advantages over eGFP and is currently the best FP for proExM. hfYFP and mhYFP are from mGL, however when the authors compared certain biochemical properties of different FPs, some FPs were selected for comparison with a biased selectivity. In detail, in Figure 1b-1g the FPs selected for comparison were different, and mGL was not selected for comparison.

We did not purposely select FPs for comparison with “bias.” Rather, we tried to select a range of commonly used FPs throughout. Obviously, we could not compare to all FPs for the laborious EM, proExM, and protein-purification experiments - we tried to pick the most reasonable comparisons in those cases. eYFP, eGFP, and superfolder GFP are among the most commonly used FPs in biology - we felt that comparisons to these were perhaps the most useful that we could make. Although we appreciate that mGL is a useful benchmark, it is still a new protein, and biologists are much more familiar with eGFP and eYFP. We use the Fig. 1 panels to direct attention to very specific and clear points about hfYFP, not to compare it against numerous controls - which we have already measured in the mGL study, often in the same assays using the same methods. Nonetheless, this study greatly expands our knowledge about mGL through numerous experiments in which it is used as a standard. We will make more clear in text the numerous comparisons that have already been done in the mGL paper.

Key points of the mentioned panels:

1b) eYFP absorbs 405 nm light, whereas hfYFP and mhYFP do not. We have already shown the spectrum of mGL in the mGL paper and we wished to compare Y203-type avFPs here.

1c) hfYFP matures faster than eYFP. mGL's maturation is measured in the mGL paper using the same methods. We wanted to show that the widely used eYFP has a maturation problem that hfYFP does not.

1d) mGL is included here.

1e) mGL cellular brightness is already shown in the mGL paper against the same controls using the same method.

1f-g) hfYFP has a lower pKa and is less chloride sensitive than eYFP. We know that mGL has a very low pKa, and chloride sensitivity is not a concern in avGFPs or suspected for mGL.

mGL is only compared in Table S2 for biochemical properties and brightness.

mGL was fully characterized in the mGL paper. In the hfYFP paper, we explore its chemical resistance properties in as much detail as hfYFP's. We will do a better job of referencing data published in the mGL paper.

The mF4Y-SR mutant suddenly appeared only in Fig.3e and not in other Fig. 3 figures.

We were curious to see how mF4Y-SR would perform, and since we had the data, decided to include it. However, since the other figures do not include it, we'll remove mF4Y-SR to avoid confusion.

In Fig. 4d-g, mEos4b was not selected for comparison as a control to demonstrate the CLEM imaging performance of mhYFP in cells.

The performance of mEos4b is known and has been reported. We aimed to compare constitutively fluorescent FPs here - that is, an in-class comparison. We also showed that (m)hfYFP resist OsO₄, the primary quenching reagent, at least as well as mEos4b. We do not mean for (m)hfYFP to supplant mEos4b - indeed, it is possible that they would complement it in 2-color applications.

For another example, based on the anti-GdnHCl properties of hfYFP and LSSmGFP, the authors propose that they can be used to achieve fluorescence-assisted benchtop purification of inclusion body proteins under fully denaturing conditions. As a template, mGL has anti-GdnHCl properties. When visualizing the different steps of protein purification using hfYFP and LSSmGFP (Fig. 5), it is necessary to compare with mGL and prove that the performances are better than mGL.

Although mGL exhibits strong resistance to GdnHCl, hfYFP simply did not denature under GdnHCl. That gave us the idea for the purification system.

Also, mAmetrine should be selected as a control in Fig. S15a. etc.

mAmetrine is shown in every panel of ~~Fig. S15~~ (**Fig. S17**) except for ~~S15a~~ Fig. S17a. The excitation spectrum of mAmetrine has been reported elsewhere, and we describe in the text that it is not 488-nm excitable, like LSSmGFP.

Additionally, the authors' claim in the Discussion that "we produced a series of avFP mutants demonstrating superior stability over sfGFP in every assay that we performed." is incorrect. Many figures in the manuscript used eGFP instead of sfGFP as a control (Fig. 1c, Fig. 3a-f, Fig. 4d-g, Fig. 5f). It is more reasonable to use sfGFP as a control rather than eGFP when discussing the stability of FPs and their applications. Taking Fig.5f and Fig.S17 as examples, when used as solubility tags, the aggregation properties of FPs themselves will affect the solubility properties of POIs. eGFP is a weak dimer. mGL and mEGFP are monomers. It is not reasonable to compare hfYFP and LSSmGFP with eGFP rather than mEGFP/sfGFP.

This is a reasonable criticism of this statement. We will tone it down in the text. Thank you for this recommendation.

Minor concern

1. When I first saw "ultrastable" in the title I thought "photostable". I recommend using "chemically stable" instead of "ultrastable".

We will replace the word "ultrastable" with "chemically stable."

2. There is no representative cell image by proExM using different FPs in Fig.3.

For this revision we have performed ExM experiments to visualize LifeAct-mhYFP and will include a representative image in Fig. 3.

3. mEos4b is not the best OsO4-tolerant fluorescent protein. In Fig.4 the authors need to compare mhYFP with mEosEM (Nature Methods, 2020). The performances of these FPs in GMA and Epon-embedded samples also need to be compared.

mEosEM was published in 2020, after we performed the CLEM experiments in late 2018 (there were enormous COVID-induced delays in this project). At the time, mEos4b was the state of the art. Although the performance of our FPs in GMA and Epon would be very interesting to know, we haven't tested those resins and we make no claims about mGL or hfYFP performance in them. And again, we are not claiming that (m)hfYFP is superior to any Eos protein for CLEM - they are different types of FP - one constitutive, one photoconvertible. They are not interchangeable - rather, they might be used alongside.

4. How do LSSmGFP and LSSA12 perform in ExM applications compared to mhYFP?

They are compatible with proExM, but we haven't quantified their retained fluorescence in the way that we did with the other FPs, since we engineered these quite a while after performing the quantitative ExM experiments.

5. The title of the manuscript is "Ultrastable FPs for advanced microscopy". Because super-resolution (SR) imaging is an important component of advanced microscopy methods, as the authors described in their introduction, the authors need to discuss the application of the developed FPs in SR and compare their advantages and disadvantages with existing proteins.

We have demonstrated that hfYFP survives OsO4 and a standard CLEM preparation protocol. One application is that hfYFP can survive the process at all—very few FPs can. The FPs tolerated proExM with fluorescence retention values that we display, and we use eGFP, mClover3, mNG, eYFP, and mGL as controls (Fig. 3a). We can discuss these issues more in the text.

6. The quality of images by advanced microscopies depends on the retained fluorescence of FP. Since different FPs have different molecular and cellular brightness, the retained fluorescence relative to eGFP should be compared in Fig.3 and Fig.4 as in Fig.1e.

We show the molecular and cellular brightness values of each FP in different figures, but we can put together a table that unifies these values and try to come up with an appropriate mathematical formula to bring that all together. It would further emphasize the advantages of mGL and hfYFP in these experiments compared to eGFP, sfGFP, and eYFP, for example.

7. The photostability of mGL is not good. How about hfYFP?

Thank you for this reminder. Photobleaching curves under laser-scanning confocal illumination for hfYFP have been added (**Fig. S8b**).

8. Fig.4 : 1) Fig.4d and 4e: Inconsistent with the description of Line 231: which one is before? Which one is after? 2) hfYFP is used in d and e, but mhYFP is used in statistical graphs f and g; mhYFP is marked in h, but hfYFP is described in the text (Line 234-235). 3) The fluorescence signals of eGFP and mGL in Fig. 4h are very strong, and the significant advantage of mhYF over eGFP and mGL cannot be seen from the figure.

Thank you for catching this, and we apologize for the confusion. These are typos and we have corrected them. mhYFP that was used throughout Fig. 4d-h, not hfYFP.

9. Sometimes eGFP is used in the manuscript, and sometimes EGFP is used, which needs to be unified.

Corrected.

10. mAmetrine should be compared in Fig.5 and Fig.S15. mAmetrine and mGL are a good pair, which shows no bleed-through between the 405 nm and 470 nm channels. And there are many FPs reported for dual-color imaging. The advantages of the mGL/LSSmGFP pair is not significant when it is only used for the general dual-color imaging techniques. The authors should demonstrate the advantages of the mhYFP/LSSmGFP pair in applications based on their chemical stability, for example in ExM imaging.

mAmetrine has a clean excitation spectrum with no peak in the 488 nm window, so we fully expect that it would pair well with mGL. Something we aimed to demonstrate through LSSmGFP is the value of our crystal structures in guiding the engineering of new FPs. This led to LSSmGFP, with good chemical stability and a clean excitation spectrum due to a novel mutation uncovered in our screening (which was also effective at improving mT-Sapphire). It's true that there are various FPs for dual-color imaging approaches, most notably mT-Sapphire and mAmetrine for the green-yellow range. We simply wished to demonstrate the practical consequences of these excitation traces in cells, since the impact on imaging may not be obvious to a broad audience when interpreting only the excitation scans. Fig. 5e shows using

photographs that cross-excitation from the eGFP/mT-Sapphire pair occurs under routine imaging conditions, consistent with the spectral scans.

11. There are two bs in Table S3.

Thank you!

12. It appears that the localization of mhYFP in Fig. 3f is different from those of the other proteins. Does mhYFP labeling affect H2B localization? More localization images of mhYFP need to be shown as in Fig. 1h-1m.

We routinely use hfYFP for nuclear targeting with H2B and have not found it to affect H2B localization. Some of the cells selected for Fig. 3f appear to be displaying overexpression artifacts. We have decided to drop the glyoxal figure and to repeat the PFA and PFA/Glut experiments using HEK293T cells with cytosolic localization. The PFA and PFA/Glut data are displayed in Fig. 3 with representative images.

To confirm that the puncta in Fig. 3f are due to transfection with too much DNA, we transfected HeLa cells with 150 ng or 500 ng DNA and compared them. Representative images of healthy nuclear morphology after transfection (150 ng DNA) with H2B-eGFP, H2B-hfYFP, and H2B-mhYFP, are shown in Fig. 8a and below.

13. The proportion of normal cells for Clover measured in Fig. S6 is 63%, while that in Fig. S15f is 75%. How to explain such a big difference? The proportion of normal cells for Clover is 75% (Fig. S15f), which is defined as weak dimer. However, in Fig. S6 hfYFP-V206K (79%) and mhYFP (80%) were defined as monomers. What is the basis for defining the aggregation state of a FP as weak dimer or monomer?

The OSER assay is highly—some would say notoriously—variable. The scores depend greatly on transfection conditions, experimental timing, focusing and brightness of cells, scoring criteria,

and the passage number of cells. We make our absolute best effort to reproduce the protocol described in our methods section and in Costantini et al., *Traffic*, 2012, and Cranfill et al., *Nat. Met.*, 2016, but there is an expected degree of variability inherent in this morphological assay, which is scored by eye.

Source	mEGFP	mGL	moxGFP	Clover	mTSapphire	tdTomato
Cranfill et al., Nat. Met. , 2016	98.1%	N.D.	N.D.	72.9%	95.5%	57.6%
Campbell et al., PNAS , 2020	86%	86%	N.D.	71%	N.D.	29%
Fig. S6c	N.D.	85%	86%	63%	N.D.	49%
Fig. S15f	90%	N.D.	N.D.	75%	82%	37%

As shown above, the values we obtained in our study for the monomeric proteins were all at least 80%. Consequently, we continue to use the >80% healthy morphology definition to define monomericity as we used it in the mGL paper, based on the performance of the controls in our hands. We consider our experimental data and interpretations to be in alignment with literature reports and within a very reasonable error range.

We noticed that we have interpreted hfYFP-V206K (79%) as monomeric. If we are to apply a strict 80% cutoff for monomericity, then we should instead refer to hfYFP-V206K as a “weak dimer,” and we will make that correction.

We have updated Table 1, Fig. S6e (**Fig. S8e**), and Fig. S15f (**Fig. S17h**), to include our scoring criteria for the OSER assay (same as in Campbell et al., *PNAS*, 2020).

14. In Fig.S2, there is a typo that S147P should be S147R.

Thank you! Corrected.

Reviewer #3

Remarks to the Author:

This manuscript describes the engineering of robustly folding and super stable fluorescent proteins that can withstand harsh chemical conditions and high temperatures. Such proteins would have broad applications in expansion microscopy, correlated light and electron microscopy, and as visualization tags during protein purification. The manuscript is a tour-de-force and presents a comprehensive analysis of structural features that contribute to rapid folding and high stability. The supplementary information is a treasure trove of valuable insights

that would be extremely useful for the broader protein engineering community. The tools themselves (hfYFP, mhYFP, LSSmGFP, LSSA12) appear to present notable strengths over currently available FPs for advanced applications. The experiments are well described. The methods and analysis are thorough and rigorous. The conclusions are well-supported by the data presented. The referencing is appropriate. The manuscript is clear and easy to follow. I think this is a strong contribution and I support publication. There are a few minor issues (noted below) that would improve clarity.

- 1) I don't understand why there is an increase in fluorescence for mGreenLantern and Hyperfolder YFP (Fig 2a and 2c). Can the authors speculate on/explain the origin of this increase?

We're not entirely sure mechanistically why a peak or plateau often shows up in our kinetic unfolding traces. It may represent a transient aggregation effect or a gradual change in solubility as the protein equilibrates with the concentrated guanidinium (7.0 M \rightarrow 6.3 M upon 1:10 dilution by protein stock), displaces water molecules, associates with the protein surface, and disperses the proteins.

We see these initial peaks in other FPs, so the effect is not exclusive to hfYFP and mGL. However, most proteins such as eGFP and sfGFP denature within seconds, sometimes before the microplate can be transferred from the bench to the monochromator (lag time \approx 10 s).

In the figure below, we've plotted sfGFP, mGL, and hfYFP data from Fig. 2a on separate graphs and have included the native and GdnHCl curves that produce the "fraction folded" curves (GdnHCl divided by Native) displayed in Fig. 2a. An initial dip in the Native fluorescence trace is visible for sfGFP. The dip is more exaggerated for the slowly denaturing proteins mGL and hfYFP.

Calculating “fraction folded” by dividing the GdnHCl condition by corresponding wells of native protein (in the same buffer without GdnHCl, at the same pH) corrects for minor potential photobleaching effects that can occur during kinetic loops with short time intervals (as described in Fisher & DeLisa, *PLOS One*, 2008).

If the peaks look odd or confusing, we could instead plot the only the GdnHCl trace (instead of “fraction folded”), but that would make the plot inconsistent with the other GdnHCl figures (e.g., Fig. S11), especially those with shorter time intervals... We’re open to feedback on this.

Looking at the Data from Fig. S11:

Above: hfYFP data from Fig. S11 screening data, here modified to display nonlinear regression (one-phase association). No peak was observed in this experiment. The one-phase association curve overlaps with the data points ($R^2 = 0.9904$ and 0.9824 for hfYFP and mhYFP, respectively). Note that a minor dip is still visible at <0.5 hr here, too.

Above: back to the hfYFP data from Fig. 2a, with data points and non-linear regression shown above. Denaturation occurred more rapidly in this experiment (N.B., for all samples). Nonetheless, the overall trends and conclusions remain consistent across experiments, with hfYFP and mhYFP reaching a stable plateau at $\sim 1.5x$ brighter fluorescence than baseline.

Please let us know your thoughts and whether we should make any changes to the figure.

- 2) Figure 3: (c) and (f) would benefit from inclusion of a scale bar. The quality of the images is poor in my pdf (even when zoomed in on a computer screen) – but the morphology of mhYFP in (f) appears drastically different (it looks punctate?) compared to the others. Is there a localization/morphology difference? This seems important to evaluate for histology purposes. Lastly, given the variability in signal in (a), (d), and (e), these would benefit from statistical analysis.

We had originally used BE(2)-M17 neuroblastoma cells in the Fig. 3b-c experiment. Their nuclei are small, and under the selected imaging settings and magnification, we agree that the image quality is low. Since the small size and pixelation could affect quantitation, we've repeated the experiment using HEK293T cells with FPs expressed in the cytosol rather than the nucleus. To maintain consistency between cell lines, reagents, and equipment, we also repeated the 4% PFA fixation experiment in Fig. 3d.

Methods have been updated accordingly. Representative images are shown, and scale bars have been included. Trends are similar to those in the prior experiments. All FPs are generally less quenched than in the prior experiment—this is probably due to better focusing during the experiment and improved thresholding/masking during analysis. Minor cell-type specific differences are possible.

The puncta observed in some cells in Fig. 3f are overexpression artifacts from transfecting too much DNA (please see our response to a similar comment by Reviewer #2 for more information). We've reviewed the original tile scans and see too many cells with this appearance in the original glyoxal experiment, so we've decided to drop that figure.

We've generated statistics tables for the experiments in **Fig. 3a,b,e** and **Fig. S17e-f** (fixative and ExM experiments) to create **Fig. S12**. We've also added images of additional representative H2B-eGFP, H2B-hfYFP, and H2B-mhYFP transfected HeLa cells in **Fig. 8a**, showing healthy nuclear morphology.

- 3) Figure 4: I think scale bars are present in (d) and (e) but they are so tiny I can't read them. Please add visible, readable scale bars.

Thank you. We've updated the scale bars.

- 4) The manuscript would benefit from some sort of qualified statement about the potential risks of super-stable proteins. There is a lot of accumulated knowledge in the FP community about what FP to use and what FP to avoid for a particular application. These nuances are not always passed on to the broader biological community. For example, the community generally accepts that over long periods of time mCherry can be toxic and can accumulate in lysosomes. The protein appears not to be efficiently broken down in the lysosome and because of mCherry's low

pKa, it retains fluorescence. I have heard discussions at meetings and workshops that perhaps mCherry isn't efficiently degraded because it is too stable. I imagine the same problem could occur for fusions to super stable proteins. This may not be an issue if users are fusing hfYFP to a long lived structural protein to be visualized in CLEM or proExM, but it could be a significant issue on a short-lived signaling protein. It seems wise to include a qualified statement that users should check whether fusion alters the protein stability/half-life.

This is an excellent suggestion. We've added several sentences in the Discussion section.

Decision Letter, second revision:

Dear Ben,

Thank you for submitting your revised manuscript "Chemically stable fluorescent proteins for advanced microscopy" (NMETH-A48110C). We have gone through the last added experiments and are happy in principle to publish the paper in Nature Methods, pending minor revisions to comply with our editorial and formatting guidelines.

Note, we try to avoid the word 'ultra' in our papers, as where does one go from ultra as technology improves? Will you consider changing your title to "highly stable" or similar and updating the text accordingly? We are OK with 'hyperfolder' because we don't usually ask authors to rename their tools and the referees didn't object to that term.

TRANSPARENT PEER REVIEW

Nature Methods offers a transparent peer review option for new original research manuscripts submitted from 17th February 2021. We encourage increased transparency in peer review by publishing the reviewer comments, author rebuttal letters and editorial decision letters if the authors agree. Such peer review material is made available as a supplementary peer review file. Please state in the cover letter 'I wish to participate in transparent peer review' if you want to opt in, or 'I do not wish to participate in transparent peer review' if you don't. Failure to state your preference will result in delays in accepting your manuscript for publication.

Thank you again for your interest in Nature Methods Please do not hesitate to contact me if you have any questions.

Sincerely,
Rita

Rita Strack, Ph.D.
Senior Editor
Nature Methods

ORCID

Final Decision Letter:

Dear Ben,

I am pleased to inform you that your Article, "Chemically stable fluorescent proteins for advanced microscopy", has now been accepted for publication in Nature Methods. Your paper is tentatively scheduled for publication in our November print issue, and will be published online prior to that. The received and accepted dates will be Jan 23, 2022 and Sept 26, 2022. This note is intended to let you know what to expect from us over the next month or so, and to let you know where to address any further questions.

Over the next few weeks, your paper will be copyedited to ensure that it conforms to Nature Methods style. Once your paper is typeset, you will receive an email with a link to choose the appropriate

publishing options for your paper and our Author Services team will be in touch regarding any additional information that may be required.

Your paper will now be copyedited to ensure that it conforms to Nature Methods style. Once proofs are generated, they will be sent to you electronically and you will be asked to send a corrected version within 24 hours. It is extremely important that you let us know now whether you will be difficult to contact over the next month. If this is the case, we ask that you send us the contact information (email, phone and fax) of someone who will be able to check the proofs and deal with any last-minute problems.

If, when you receive your proof, you cannot meet the deadline, please inform us at rjsproduction@springernature.com immediately.

Once your manuscript is typeset and you have completed the appropriate grant of rights, you will receive a link to your electronic proof via email with a request to make any corrections within 48 hours. If, when you receive your proof, you cannot meet this deadline, please inform us at rjsproduction@springernature.com immediately.

Once your paper has been scheduled for online publication, the Nature press office will be in touch to confirm the details.

Content is published online weekly on Mondays and Thursdays, and the embargo is set at 16:00 London time (GMT)/11:00 am US Eastern time (EST) on the day of publication. If you need to know the exact publication date or when the news embargo will be lifted, please contact our press office after you have submitted your proof corrections. Now is the time to inform your Public Relations or Press Office about your paper, as they might be interested in promoting its publication. This will allow them time to prepare an accurate and satisfactory press release. Include your manuscript tracking number NMETH-A48110D and the name of the journal, which they will need when they contact our office.

About one week before your paper is published online, we shall be distributing a press release to news organizations worldwide, which may include details of your work. We are happy for your institution or funding agency to prepare its own press release, but it must mention the embargo date and Nature Methods. Our Press Office will contact you closer to the time of publication, but if you or your Press Office have any inquiries in the meantime, please contact press@nature.com.

If you are active on Twitter, please e-mail me your and your coauthors' Twitter handles so that we may tag you when the paper is published.

Please note that Nature Methods is a Transformative Journal (TJ). Authors may publish their research with us through the traditional subscription access route or make their paper immediately open access through payment of an article-processing charge (APC). Authors will not be required to make a final decision about access to their article until it has been accepted. Find out more about Transformative Journals

Authors may need to take specific actions to achieve compliance with funder and institutional open access mandates. If your research is supported by a funder that requires immediate open access (e.g. according to Plan S principles) then you should select the gold OA route, and we will direct you to the compliant route where possible. For authors selecting the subscription publication route, the journal's standard licensing terms will need to be accepted, including self-archiving policies. Those licensing terms will supersede any other terms that the author or any third party may assert apply to any version of the manuscript.

To assist our authors in disseminating their research to the broader community, our SharedIt initiative provides you with a unique shareable link that will allow anyone (with or without a subscription) to read the published article. Recipients of the link with a subscription will also be able to download and print the PDF. As soon as your article is published, you will receive an automated email with your shareable link.

Please note that you and your coauthors may order reprints and single copies of the issue containing your article through Nature Portfolio's reprint website, which is located at <http://www.nature.com/reprints/author-reprints.html>. If there are any questions about reprints please send an email to author-reprints@nature.com and someone will assist you.

Best regards,
Rita